# ON THE SUBOPTIMALITY OF SEMI-MARKOV DECISION PROCESS IN HIERARCHICAL REINFORCEMENT LEARNING

## ABSTRACT

Hierarchical Reinforcement Learning (HRL) demonstrates highly efficient exploration in long-horizon decision-making problems with sparse rewards via the Semi-Markov Decision Process (SMDP). However, we observe a structural limitation of SMDP in HRL: once calling a subtask, the agent is locked into a fixed course of action, losing the flexibility to adapt to other higher-value subtasks, which is a critical barrier in optimality. To address this issue, we first decompose this suboptimality into execution suboptimality and policy suboptimality, and then propose corresponding algorithmic improvement frameworks. On the theoretical side, we reveal a fundamental design flaw in HRL where SMDP is simultaneously adopted in both the target and behavior policies. To overcome this flaw, we introduce the concepts of task tree and execution tree to decouple them, reducing the problem to a tradeoff between exploration and exploitation over policy execution modes. By constructing a unified value function and a generalized hierarchical Bellman equation, we achieve a multi-level value formalization. Upon this, we further propose Hierarchical Policy Improvement Theorem and Optimal Execution Theorem. These results theoretically prove the existence of two types of suboptimality and provide guarantees for the proposed improvement frameworks. Controlled experiments across diverse environments consistently validate both the correctness of our theory and the effectiveness of the proposed improvements.[1]

## 1 INTRODUCTION

The agent using Hierarchical Reinforcement Learning (HRL) based on Semi-Markov Decision Process (SMDP) possesses the ability to learn and make decisions at multiple time scales (Sutton et al., 1999; Andre & Russell, 2002; Barto & Mahadevan, 2003; Riemer et al., 2018; 2020). The agent can effectively utilize the repetitive structure within Markov Decision Processes (MDP) by organizing a hierarchical structure between tasks. This gives HRL higher statistical efficiency when dealing with long-term decision problems in sparse reward environments (Wen et al., 2020). Compared with traditional single-level RL algorithms difficult to learn such problems effectively, HRL algorithms often perform excellently (Levy et al., 2019; Zhang et al., 2020; Pateria et al., 2021).

Just as the exploration-exploitation tradeoff exists in RL, HRL, while enabling more efficient exploration, also introduces various suboptimality issues (Nachum et al., 2019; Sutton et al., 1999). Intuitively, the agent in SMDP gives up deciding new subtasks in many timesteps, thereby fully calling the dynamic transitions generated by the subtasks and leveraging the repetitive structure for efficient exploration during the training phase. However, the ability of the agent to select better subtasks is also given up in these timesteps, even in the deployment phase without exploration, which brings about suboptimality in terms of exploitation. This paper formally analyzes the suboptimality issue arising from the policy's execution mode.

**Contributions:** 1) We identify two execution modes in HRL: *Markov Execution (ME)* and *Semi-Markov Execution (SME)*. The suboptimality caused by SME in SMDP can be decomposed into *policy suboptimality* during the training phase and *execution suboptimality* during the deployment

---

[1]The associated source code is available in the supplementary.

Figure 1: Differences between MDP and SMDP: calling actions vs. calling subtasks. The superscript of $\tau$ denotes its hierarchical level, while the subscript denotes its called index $\hat{t}$. The subscript of actions $a$ indicates the timestep index $t$.

phase. 2) We introduce the concepts of *task tree* and *execution tree*, formally decouple the optimization objectives and execution modes of tasks in more-than-two-level HRL. Based on this, we establish the corresponding *unified value functions* and *Bellman equations*, propose *Hierarchical Policy Improvement Theorem* and *Optimal Execution Theorem*, and extend the HRL framework to address both types of suboptimality. 3) We theoretically prove these two types of suboptimality and design three algorithmic *improvement frameworks* to validate these theories experimentally.

Fundamentally, we **formally state** the following arguments (We illustrate SME and ME in Fig. 2):

1) The agent in HRL should interact with the environment using SME during the training phase to ensure exploration. 2) The policy optimization objective should be set to the optimal policy under ME. 3) During the deployment phase, ME should be used to maximize the expected return.

## 2 BACKGROUND

### 2.1 MARKOV DECISION PROCESSES

A **task** $\tau$ is described as a Markov Decision Process (MDP) defined by the six-tuple $(\mathcal{S}, \mathcal{A}, T, R_\tau, \alpha_\tau, \beta_\tau)$. This MDP comprises a state space $\mathcal{S}$, a action space $\mathcal{A}$, a transition kernel $T : \mathcal{S} \times \mathcal{A} \to \Delta(\mathcal{S})$ (where $\Delta(\mathcal{S})$ denotes probability distributions over $\mathcal{S}$), a reward function $R_\tau : \mathcal{S} \times \mathcal{A} \to \mathbb{R}$, a task initial-state distribution $\alpha_\tau \in \Delta(\mathcal{S})$, and a termination function $\beta_\tau : \mathcal{S} \to [0, 1]$ which means the probability of exiting task $\tau$ upon visiting a state $s$. $1 - \beta_\tau$ plays a role analogous to the discount factor $\gamma_\tau$.

We define one unit of timestep in which the agent takes an action and interacts with the environment as a *frames*, and denote the current frame index as $t$. When the agent interacts with the environment by calling the policy $\pi_\tau : \mathcal{S} \to \Delta\mathcal{A}$, its value function is defined as $V_\pi(s|\tau) = \mathbb{E}_{\pi_\tau, s_0 = s} \left[ \sum_{t=0}^\infty \left( R_\tau(s_t, a_t) \prod_{k=0}^{t-1} (1 - \beta_\tau(s_k)) \right) \right]$, where $\pi$ is the set of policies for all tasks,[2] $\forall \tau, \pi_\tau \in \pi$. The optimization objective of $\pi_\tau$ is to maximize $V_\pi(\cdot|\tau)$, and different $\tau$ will have different optimization objectives due to different $\beta_\tau$. The expected return of task $\tau$ is $J(\pi|\tau) = \mathbb{E}_{s \sim \alpha_\tau} V_\pi(s|\tau)$.

### 2.2 SEMI-MARKOV DECISION PROCESSES

Tasks not only affect the optimization objective of their policies, but also induce dynamic transitions in the environment when executed by the agent, similar to the effects of actions. Therefore, as illustrated in Fig. 1, in HRL it is possible to call other tasks multiple times to accomplish $\tau$ instead of directly calling actions. We refer to this stochastic process with variable-duration transitions as a Semi-Markov Decision Process (SMDP), and the tasks that can be called under $\tau$ are called the **subtasks** of $\tau$.

In the formalization, we replace $\mathcal{A}$ in the MDP tuple with the subtask set $\mathcal{T}'_\tau$ of $\tau$, and define the SMDP of task $\tau$ as $(\mathcal{S}, \mathcal{T}'_\tau, \bar{T}_\tau, \bar{R}_\tau, \alpha_\tau, \beta_\tau)$. Here, $\bar{T}_\tau : \mathcal{S} \times \mathcal{T}'_\tau \to \Delta\mathcal{S}$ is the subtask transition function, and $\bar{R}_\tau : \mathcal{S} \times \mathcal{T}'_\tau \to \mathbb{R}$ is the subtask reward function.

---

[2]We will extend and redefine the concepts of certain traditional symbols. Notations are in Appendix A

Since tasks at different hierarchical levels operate on different time scales, we use $\hat{t}$ to record the number of subtasks that have been called by a given task. We refer to the policy by which $\tau$ calls its subtasks as the node policy $\hat{\pi}_\tau : \mathcal{S} \to \Delta\mathcal{T}'_\tau$, and we denote the mapping for $\tau$ that starts from $\hat{\pi}_\tau$ and keeps selecting subtasks until an action is eventually chosen, while keeping the original notation, as the policy $\pi_\tau : \mathcal{S} \to \Delta\mathcal{A}$. The value function that the node policy $\hat{\pi}_\tau$ needs to maximize can be simply formalized as $V_\pi(s|\tau) = \mathbb{E}_{\tau' \sim \hat{\pi}_\tau, s_0 = s} \left[ \sum_{\hat{t}=0}^{\infty} \left( R_\tau(s_{\hat{t}}, \tau'_{\hat{t}}) \prod_{k=0}^{\hat{t}-1}(1 - \beta_\tau(s_k)) \right) \right]$. [3] In an MDP, the agent (standard RL) calls primitive actions that last exactly one time step. In an SMDP, the agent (HRL at a higher level) calls subtasks that last for a random number of time steps.

### 2.3 PROBLEM SETUP

In this paper, we divide the algorithm into two phases: the training phase and the deployment phase.

**Training phase:** The agent interacts with the environment to learn appropriate policies.

**Deployment phase:** The agent needs to call the learned policies in a reasonable manner.

We examine the final performance exhibited by the agent when interacting with the environment during the deployment phase while executing policies in a certain manner.[4] The main motivation for this setup is that during training, the agent needs to explore the environment, and policy execution involves an exploration mechanism. However, during deployment, more emphasis should be placed on the optimal performance that the learned policies can achieve in terms of exploitation. We find that this exploration-exploitation tradeoff takes on a new form in SMDP-based HRL.

## 3 PROBLEM FORMULATION

In this section, we analyze the impact of Semi-Markov Decision Processes on the agent during training and deployment through a simple case study. We point out two distinctly different execution modes when calling subtasks in HRL.

**Definition 3.1** (**Markov Execution, ME**). *Task $\tau$ makes a decision and calls a new subtask $\tau'$ at every frame, meaning that each called $\tau'$ generates only a single-frame transition. We refer to this execution mode, in which each transition lasts only one frame, as task $\tau$ calling $\hat{\pi}_\tau$ in a ME manner. When all tasks of the agent call their node policies in the ME manner, we refer to the agent as calling $\pi$ in the ME manner. The value function under ME is formalized as* $V_\pi^{ME}(s|\tau) = \mathbb{E}_{\tau' \sim \hat{\pi}_\tau, s_0 = s} \left[ \sum_{\hat{t}=0}^{\infty} \left( R_\tau(s_{\hat{t}}, \tau'_{\hat{t}}) \prod_{k=0}^{\hat{t}-1}(1 - \beta_\tau(s_k)) \right) \mid \tau' \text{ is executed in single-frame} \right]$.

**Definition 3.2** (**Semi-Markov Execution, SME**). *After task $\tau$ decides and calls a subtask $\tau'$, $\tau'$ is executed in full until it terminates due to termination function $\beta_{\tau'}$. We refer to this execution mode, in which each transition lasts for the entire execution of the subtask, as task $\tau$ calling $\hat{\pi}_\tau$ in a SME manner. When all tasks of the agent call their node policies in the SME manner, we refer to the agent as calling $\pi$ in the SME manner. The value function under SME is formalized as* $V_\pi^{SME}(s|\tau) = \mathbb{E}_{\tau' \sim \hat{\pi}_\tau, s_0 = s} \left[ \sum_{\hat{t}=0}^{\infty} \left( R_\tau(s_{\hat{t}}, \tau'_{\hat{t}}) \prod_{k=0}^{\hat{t}-1}(1 - \beta_\tau(s_k)) \right) \mid \tau' \text{ is executed in full} \right]$.

As shown in Fig. 2, ME ignores the original termination functions of subtasks, whereas SME strictly follows the termination functions of subtasks. SME used in HRL can lead to suboptimality. In this paper, we decompose this suboptimality into *execution suboptimality* and *policy suboptimality*, which are illustrated using the *GridWorld* shown in Fig. 3(a):

### 3.1 EXECUTION SUBOPTIMALITY

Fig. 3(b) shows the trajectories of an agent with a two-level policy set $\pi$ under SME and ME. Given the termination functions and node policies of all tasks within the agent, ***using different execution modes results in different trajectories***. That is, for any $\pi$, there exist the expected return under SME $J^{\text{SME}}(\pi|\tau)$, and the expected return under ME $J^{\text{ME}}(\pi|\tau)$.

---

[3] The value calculation here ignores the case where a subtask is terminated due to the termination of its higher-level task during its execution.

[4] This setup is more reasonable in RL, for example, by using $\epsilon$-greedy during training and greedy during deployment, or softmax during training and argmax during deployment.

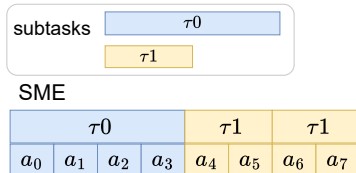

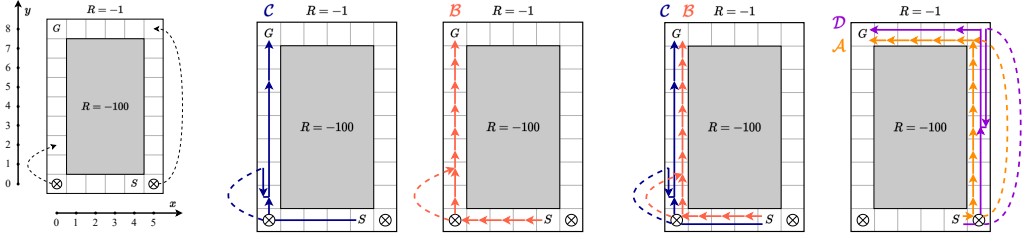

Figure 2: Interaction differences between SME and ME: calling subtasks in full vs. calling subtasks in one timestep. Task $\tau$ has a 4-timestep subtask $\tau0$ and a 2-timestep subtask $\tau1$. Both $\tau0$ and $\tau1$ directly call actions. The subtask $\tau0$ and the actions it decides are represented in blue, while $\tau1$ and its actions are represented in yellow.

(a) The GridWorld.   (b) Execution results of SME and ME.   (c) Policy differences between $\hat{\pi}_{hi}^-$ and $\hat{\pi}_{hi}^*$.

Figure 3: Trajectories of the agent in the *GridWorld* under different policies and execution modes. The agent needs to reach $G$ at $(0,8)$ from $S$ at $(4,0)$ in the minimum number of frames, receiving a reward of $-1$ per frame. After reaching the position marked $\otimes$, there is a 50% chance of an additional state transition along the dashed line. The trajectories of the SMDP-optimal policy $\hat{\pi}_{hi}^-$ under SME and ME are shown in blue and red, with their expected returns denoted by $J_{\mathcal{B}}$ and $J_{\mathcal{C}}$ respectively. The trajectories of the MDP-optimal policy $\hat{\pi}_{hi}^*$ under SME and ME are shown in purple and orange, with their expected returns denoted by $J_{\mathcal{D}}$ and $J_{\mathcal{A}}$ respectively. The detailed descriptions of the policies and execution modes, as well as the related numerical computations, are provided in the Appendix B

To reach $(0,8)$ as quickly as possible, up is the optimal action at state $(0,2)$. For the agent policy set in Fig. 3(b), it is clear that ME achieves a higher expected return than SME: $J^{\text{ME}}(\pi|G) = -11 > -12 = J^{\text{SME}}(\pi|G)$. During the execution of subtask to reach $(0,1)$, the subtask with the highest expected return changed (due to a random event), but SME could not promptly stop the unfinished suboptimal subtask, resulting in suboptimality. We refer to this kind of suboptimality during the deployment phase as **execution suboptimality**.

### 3.2 POLICY SUBOPTIMALITY

Fig. 3(c) shows the trajectories of the optimal policy under the two different execution modes, SME and ME: $J^{\text{ME}}(\pi^*|G) = -8 > -11 = J^{\text{ME}}(\pi^-|G)$, $J^{\text{SME}}(\pi^-|G) = -12 > -14 = J^{\text{SME}}(\pi^*|G)$. Given the termination functions of all subtasks within an agent, ***different execution modes can lead to different optimal node policies.***This paper refers to the difference in policy set caused by the execution mode during the training phase as **policy suboptimality**. We provide the following definitions of policy optimality under these two specific execution modes:

**Definition 3.3** (MDP-optimal). *For a task $\tau$, given the node policies of all its descendant tasks, the node policy $\hat{\pi}_\tau$ that maximizes $V_\pi^{ME}(\cdot|\tau)$ is called the MDP optimal node policy of task $\tau$, denoted as $\hat{\pi}_\tau^*$. That is, $\hat{\pi}_\tau^* = \arg\max_{\hat{\pi}_\tau} V_\pi^{ME}(\cdot|\tau)$. When all node policies in $\pi$ are MDP optimal node policies, we call $\pi$ an* MDP-optimal policy set, *denoted as $\pi^*$.*

**Definition 3.4** (SMDP-optimal). *For a task $\tau$, given the node policies and termination functions of all its descendant tasks, the policy $\hat{\pi}_\tau$ that maximizes $V_\pi^{SME}(\cdot|\tau)$ is called the SMDP optimal node policy of task $\tau$, denoted as $\hat{\pi}_\tau^-$. That is, $\hat{\pi}_\tau^- = \arg\max_{\hat{\pi}_\tau} V_\pi^{SME}(\cdot|\tau)$. When all node policies in $\pi$ are SMDP optimal node policies, we call $\pi$ an* SMDP-optimal policy set, *denoted as $\pi^-$.*

## 4 SMDP SUBOPTIMALITY THEORY

### 4.1 LIMITATIONS OF SME

We now formally present the narrow SMDP suboptimality theory under the two-level HRL setting, which provides theoretical guarantees for the limitations of SME.

**Theorem 4.1** (**Narrow SMDP Suboptimality Theorem**). *In a two-level HRL problem where the environment satisfies the Markov property, the four expected returns generated by the MDP-optimal policy set $\pi^*$ and the SMDP-optimal policy set $\pi^-$ under ME and SME are defined in Table 1. Then the following ordering holds:*

$$J_{\mathcal{A}} \geq J_{\mathcal{B}} \geq J_{\mathcal{C}} \geq J_{\mathcal{D}}.$$

Furthermore, this paper further proposes a **Generalized SMDP Suboptimality Theorem** under more-than-two-level and arbitrary execution modes, which constitutes the core theoretical contribution of this work. We provide detailed formulations, descriptions and proofs in the Appendix C, where the narrow SMDP suboptimality theory is also proven as a special case.

Table 1: Optimality comparison table.

| Expected return | SME | ME |
|---|---|---|
| SMDP-optimal policies | $J_{\mathcal{C}}$ | $J_{\mathcal{B}}$ |
| MDP-optimal policies | $J_{\mathcal{D}}$ | $J_{\mathcal{A}}$ |

*Proof Sketch for Theorem 4.1.* **Conceptual Clarification:** This paper notes that each task $\tau$ has two termination functions with different roles. One controls the optimization objective of $\hat{\pi}_{\tau}$, namely the **termination function** $\beta_{\tau}$ introduced in Section 2.2. The other controls the duration of $\tau$'s transitions when it is called as a subtask during interaction, defined as the **interrupt function** $\hat{\beta}_{\tau}$. In this context, we observe that when using ME, $\hat{\beta}_{\tau} \equiv 1$, while when using SME, $\hat{\beta}_{\tau} \equiv \beta_{\tau}$. In our proof, we further introduce the **task tree** and the **execution tree** to decouple the binding of $\hat{\beta}_{\tau}$ and $\beta_{\tau}$ in existing HRL, which respectively describe the complex task optimization objectives and task execution modes across multiple hierarchical levels.

In simple terms, for each $\tau$, the task tree corresponds to the task decomposition, which determines the termination conditions of $\tau$ as well as the boundary cases when returning to different upper-level tasks. The execution tree, on the other hand, acts as the policy scheduler, specifying how $\tau$ calls its lower-level descendant tasks. By defining these two types of trees, we can precisely formalize the complex situations in more-than-two-level hierarchical settings where the termination or interruption of any upper-level task of $\tau$ will cause $\tau$ itself to be terminated or interrupted.

**Unified Formalization:** The value computation under different execution modes cannot be directly described by the Bellman equation, since action selection depends not only on the current frame's state but also on the task. Moreover, at each frame during interaction, the hierarchical level of a task is jointly influenced by $\hat{\pi}$ and $\hat{\beta}$. For instance, in a two-level HRL, under ME, decisions are made from $\hat{\pi}_{hi}$ downward at every frame, whereas under SME, downward decisions from $\hat{\pi}_{hi}$ occur only when a subtask ends, and the remaining frames are decided by $\hat{\pi}_{lo}$. To address this, we describe the generalized interaction process of HRL using four stages: **Preparation stage:** Determine which task makes downward decisions based on $\hat{\beta}$. **Decision stage:** The task determined in the previous stage makes downward decisions according to $\hat{\pi}$ until an action $a$ is selected. **Interaction stage:** Each task receives rewards according to the agent's transitions. **Termination stage:** Tasks in execution determine termination based on $\beta$, including boundary conditions for termination.

Based on these four stages, we propose a **Unified Value Function for HRL(UVFH)**. Using this function, we establish the **Generalized Hierarchical Bellman Equation (GHBE)** and the **Generalized Hierarchical Bellman Optimality Equation (GHBOE)**, which describe the dynamic value transitions across the four stages and serve as the foundation for subsequent theoretical analysis. Here "generalized" refers to arbitrary execution modes and task hierarchy configurations.

**Theoretical Proofs:** We prove that the iterative operators of the two equations above are Contraction Mappings, thereby providing theoretical guarantees for the convergence of their solutions. On this basis, we further prove the **Generalized Hierarchical Policy Improvement Theorem**, which ensures that under the ME and SME execution settings, the optimal node policies induced by the

GHBOE correspond to the MDP optimal node policies and the SMDP optimal node policies, respectively. Consequently, we establish that $J_{\mathcal{A}} \geq J_{\mathcal{B}}$ and $J_{\mathcal{C}} \geq J_{\mathcal{D}}$.

Next, we prove the **Hierarchical Execution Improvement Theorem** for execution modes controlled by $\hat{\beta}$. Using this theorem, together with certain subtask conditions (satisfied by two-level HRL), we prove the **Optimal Execution Theorem**. This theorem guarantees that, under some conditions, the expected return of any execution mode is no greater than that under ME, i.e., $J_{\mathcal{B}} \geq J_{\mathcal{C}}$. In particular, we also provide counterexamples for multi-level HRL settings where the conditions are not satisfied for descendant task, demonstrating that the optimality of ME is not evident.

$\square$

In the process of proving Theorem 4.1, we derived the following key theorems. The relevant concepts and their precise formal definitions can be found in Appendix C.

**Theorem 4.2 (Hierarchical Policy Improvement Theorem).** *Given two execution trees $e$ and $e'$, suppose that the only difference between them lies in the node policy functions $\hat{\pi}_{n(h)}$ and $\hat{\pi}'_{n(h)}$ at state $s$. If $\mathbb{E}_{\tau_{n(h+1)} \sim \hat{\pi}_{n(h)}(s)} V_{e|n(h')}(s, \tau_{n(h+1)}) \leq \mathbb{E}_{\tau_{n(h+1)} \sim \hat{\pi}'_{n(h)}(s)} V_{e|n(h')}(s, \tau_{n(h+1)})$, then for all $s$, it holds that $V_{e|n(h')}(s, \tau_{n(h)}) \leq V_{e'|n(h')}(s, \tau_{n(h)})$, where $h \geq h'$.*

**Theorem 4.3 (Hierarchical Execution Improvement Theorem).** *Given execution trees $e$ and $e'$, suppose that only the decision interruption functions $\hat{\beta}_{n(h)}$ and $\hat{\beta}'_{n(h)}$ differ at $s'$. Let decision transition mechanism $\hat{\gamma}$ and $\hat{\gamma}'$ be computed from $\hat{\beta}$ and $\hat{\beta}'$, respectively. If for all $n(l)$ it holds that $\sum_{k=h'}^{l} \hat{\gamma}_{n(k)}(s') V_{e|n(h')}(s', \tau_{n(k)} | \tau_{n(h')}) \leq \sum_{k=h'}^{l} \hat{\gamma}'_{n(k)}(s') V_{e|n(h')}(s', \tau_{n(k)} | \tau_{n(h')})$, then for all $s$ we have $V_{e|n(h)}(s, \tau_{n(h)}) \leq V_{e'|n(h)}(s, \tau_{n(h)})$, where $l \geq h \geq h'$, and $l$ denotes the level of the task directly calling an action in the previous frame.*

**Theorem 4.4 (Optimal Execution Theorem).** *Under Claim C.1, if the node policy $\hat{\pi}_{n(h)}$ is the optimal policy for the execution induced by any interruption subtree $\mathbf{B}|n(h)$, then $V_{\Pi|n(h)}^{\mathbf{B}|n(h)} \leq V_{\Pi|n(h)}^{ME}$, i.e., the ME is the optimal execution method.*

Through the SMDP suboptimality theory, we reveal the expected return suboptimality of SME during both the training and deployment phases. To avoid policy suboptimality, the target policy should use ME to approaches $J_{\mathcal{A}}$ instead of being limited by $J_{\mathcal{B}}$. To avoid execution suboptimality, the deployment policy shoud use ME to approaches $J_{\mathcal{B}}$(or $J_{\mathcal{A}}$) instead of being limited by $J_{\mathcal{C}}$(or $J_{\mathcal{D}}$).

### 4.2 Limitations of ME

Unlike SME, under ME, $\hat{\beta}$ and $\beta$ are set differently, which may result in frequent interruptions of a task by higher-level tasks during its execution. During task optimization, deviations between the target policy and the behavior policy may occur, as indicated by the red marks in Fig. 4. This makes it difficult for the boundary conditions controlled by $\beta$ at task termination to be properly back-propagated along the interaction trajectory during dynamic programming.

Therefore, if the agent directly interacts with the environment using ME during training, the node policies of all subtasks will face severe off-policy issues, making efficient training difficult. To ensure stable sampling and learnability of lower-level subtasks during training, the behavior policy should still use SME.

### 4.3 The exploration–exploitation tradeoff induced by execution modes

Given that we have already adopted a hierarchical architecture, what extra suboptimality is introduced purely by the SME, and how can we remove it by using a more suitable execution mode? Based on the theoretical analysis above, the problem can be reduced to an exploration-exploitation

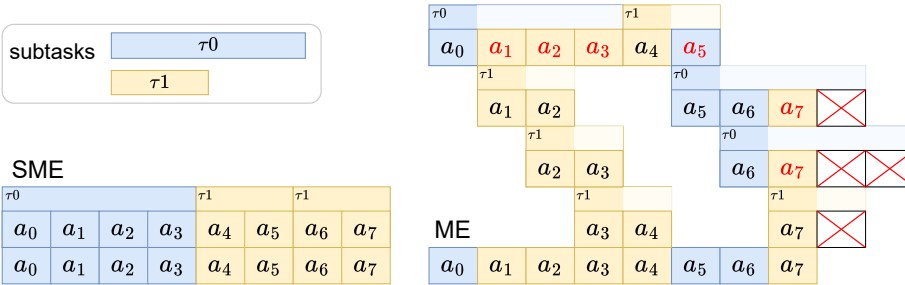

Figure 4: Subtask off-policy issue under ME. Following the example in Figure 1, the bottom row of the action sequence represents the agent's interaction trajectory with the environment, where blue and yellow colors are used to distinguish whether an action is decided by the node policy of subtask $\tau 0$ or $\tau 1$ respectively. Within each subtask box, the action sequence corresponds to the truncated trajectory used for training that subtask. Whenever the truncated trajectory contains data that is not decided by the node policy of the corresponding subtask, or when trajectory data is missing, these actions are highlighted in red, indicating the occurrence of an off-policy issue.

tradeoff caused by the execution mode in HRL. Overall, the common HRL design in which both target and behavior policies use SME is exactly the fundamental design flaw.

**During the training phase:** Directly using ME to learn the MDP-optimal policy set results in low subtask sampling efficiency, leading to an exploration problem; directly using SME to learn the SMDP-optimal policy set results in policy suboptimality, leading to an exploitation problem.

**During the deployment phase:** The optimal execution mode, ME, should be used to ensure exploitation, rather than SME.

Therefore, a reasonable HRL algorithm should balance exploration and exploitation, interacting with the environment via SME to generate data while learning the MDP-optimal policy set in some way.

## 5 IMPROVEMENT FRAMEWORKS AND ALGORITHMS

Based on Section 4, this paper designs three theoretically guaranteed frameworks for improving HRL algorithms by decoupling the **task optimization objective**, the **execution mode of the target policy set**, and the **execution mode of the behavior policy set**: the *Execution Suboptimality Improvement Framework (ESIF)*, the *Two-Stage Policy Suboptimality Improvement Framework (PSIF-2S)*, and the *One-Stage Policy Suboptimality Improvement Framework (PSIF-1S)*. Existing HRL algorithms can be improved through these frameworks to balance the exploration-exploitation tradeoff and mitigate the SMDP suboptimality of the original algorithms to varying degrees.

### 5.1 FRAMEWORKS

**ESIF:** This framework does not modify the training process of the algorithm and directly uses the original algorithm to learn $\pi^-$ (i.e., by sampling data through SME to solve the SMDP-optimal policy set). During the deployment phase, using ME can achieve the expected return $J_\mathcal{B}$. This framework is analogous to using $\epsilon$-greedy SARSA (Rummery & Niranjan, 1994) to train the agent in RL, followed by greedy execution during deployment.

**PSIF-2S:** Each task node in the task hierarchy has two training stages. **Stage 1 interaction:** When all Stage-1 tasks $\tau$ are called, they are executed until termination according to $\beta_\tau$, i.e., they are called by their higher-level tasks using SME. **Stage 2 interaction:** When all Stage-2 tasks $\tau$ are called, they are executed for only a single frame, i.e., they are called by their higher-level tasks using ME. Initially, action nodes, as special task nodes, start in Stage 2, while all other task nodes start in Stage 1. During the training phase, all tasks solve for the optimal node policies under the current execution mode according to the GHBOE. If all subtasks of $\tau$ are in Stage 2, then $\hat{\pi}_\tau$ will converge to $\hat{\pi}_\tau^*$ according to the GHBOE. Once $\hat{\pi}_\tau$ has converged to $\hat{\pi}_\tau^*$, $\tau$ enters Stage 2, and $\hat{\pi}_\tau$

completes training and its mapping is fixed. Training stops once all task nodes in the task hierarchy have entered Stage 2. During the deployment phase, using ME can achieve the expected return $J_{\mathcal{A}}$.

**PSIF-1S:** Each task node in the task hierarchy is called using SME, but all nodes off-policy improve their node policies according to the GHBOE under the ME execution mode, with each node policy converging to $\hat{\pi}_\tau^*$. During the deployment phase, using ME can achieve the expected return $J_{\mathcal{A}}$. This is analogous to using Q-Learning (Watkins & Dayan, 1992) to train the agent in RL, followed by greedy execution during deployment.

We provide the pseudocode and improvement instances for the three frameworks in Appendix D.

## 5.2 Algorithm improvement instances

The OC algorithm is a representative algorithm of the Options Framework (Stolle & Precup, 2002; Riemer et al., 2018; Brunskill & Li, 2014; Harutyunyan et al., 2019), suitable for verifying theoretical correctness.

**ESIF framework:** During deployment, options can be selected greedily at each frame according to the higher-level value function.

**PSIF-2S framework:** Stage 1: All Stage-1 tasks are executed using SME to collect trajectory data. The value updates of both high-level and low-level node policies follow the original algorithm:

$$V(s_t, \omega_t) \leftarrow R(s_t, a_t) + \gamma\big(\beta_{\omega_t}(s_{t+1})\max_{\omega} V(s_{t+1}, \omega) + (1 - \beta_{\omega_t}(s_{t+1}))V(s_{t+1}, \omega_t)\big)$$

$$V(s_t, \omega_t, a_t) \leftarrow R(s_t, a_t) + \gamma\big(\beta_{\omega_t}(s_{t+1})\max_{\omega} V(s_{t+1}, \omega) + (1 - \beta_{\omega_t}(s_{t+1}))V(s_{t+1}, \omega_t)\big)$$

where $\omega$ denotes an option subtask, and the left arrow indicates that the left-hand side is trained under its input distribution sampling, using the right-hand side as the supervision target via Mean Squared Error. $V(s, \omega)$ is used to induce the greedy high-level node policy, to provide gradients for the termination function of the low-level policy, and can also serve as a baseline for low-level policy gradient updates (note that the original OC algorithm did not subtract a baseline).

Stage 2: Tasks are executed using ME. The lower-level node policy and termination function are fixed, and the upper-level value update is modified according to the GHBOE under ME:

$$V(s_t, \omega_t) \leftarrow R(s_t, a_t) + \gamma \max_{\omega} V(s_{t+1}, \omega)$$

**PSIF-1S framework:** The agent interacts with SME, but the higher-level values are updated off-policy according to the GHBOE under ME:

$$V_{hi}(s_t, \omega_t) \leftarrow R(s_t, a_t) + \gamma \max_{\omega} V_{hi}(s_{t+1}, \omega)$$

Since lower-level actions are called for only a single frame, the lower-level value update under ME does not go off-policy. However, because options are called by the higher-level SME policy, the lower-level value updates are:

$$V_{lo}(s_t, \omega_t) \leftarrow R(s_t, a_t) + \gamma\big(\beta_{\omega_t}(s_{t+1})\, V_{lo}(s_{t+1}, \arg\max_{\omega} V_{hi}(s_{t+1}, \omega))$$
$$+ \big(1 - \beta_{\omega_t}(s_{t+1})\big)V_{lo}(s_{t+1}, \omega_t)\big)$$
$$V_{lo}(s_t, \omega_t, a_t) \leftarrow R(s_t, a_t) + \gamma\big(\beta_{\omega_t}(s_{t+1})\, V_{lo}(s_{t+1}, \arg\max_{\omega} V_{hi}(s_{t+1}, \omega))$$
$$+ \big(1 - \beta_{\omega_t}(s_{t+1})\big)V_{lo}(s_{t+1}, \omega_t)\big)$$

Here, $V_{hi}(s, \omega)$ induces the higher-level greedy node policy, while $V_{lo}(s, \omega)$ is used for lower-level termination gradient updates and as the baseline for lower-level node policy gradients.

## 6 Experiments validation and analysis

In the grid-world environment *Fourrooms*, we employ the option-based Option Critic (OC) algorithm (Bacon et al., 2017) as a representative of option frameworks with solid theoretical guarantees, to empirically validate the suboptimality theory of SMDPs. In the robotic navigation environment *AntReacher*, we adopt the HAC algorithm (Levy et al., 2019) as a representative of mainstream

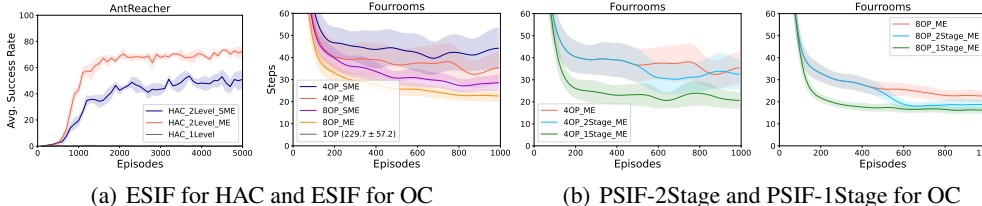

(a) ESIF for HAC and ESIF for OC    (b) PSIF-2Stage and PSIF-1Stage for OC

Figure 5: Results of the improvement frameworks. (a) Verification of execution suboptimality. (b) Verification of policy suboptimality. The shaded regions represent one standard deviation over results obtained with multiple random seeds. Higher success rates and fewer steps are better.

goal-based frameworks, verifying the execution suboptimality problem of the feudal framework and achieving performance improvements. Details of the environments, additional experiments, and visualizations are provided in Appendix E.

**Improvement of ESIF and Verification of Execution Suboptimality.** In the right side of Fig. 5(a), the execution suboptimality of the option-based baseline algorithm OC was validated on the *Fourrooms* problem, while on the left side of Fig. 5(a), the execution suboptimality of the goal-based baseline algorithm HAC was validated on the stochastic version of *AntReacher*, where this suboptimality led to a significant performance gap of over 20%. This experiment verifies both $J_{\mathcal{B}} \geq J_{\mathcal{C}}$ and the effectiveness of ESIF.

**Ablation Study.** To further illustrate the difficulty of these tasks, we also conducted single-level experimental validation using corresponding flat RL algorithms, which failed to be effectively trained on these long-horizon tasks. Therefore, it is infeasible to directly obtain the MDP-optimal policy $\pi^*$ using standard MDP-based RL algorithms. [5]

**Improvement of PSIF and Verification of Policy Suboptimality.** Fig. 5(b) shows the comparison of the OC algorithm under different settings of 4 and 8 options, where the OC baseline, OC-PSIF-2S, and OC-PSIF-1S are deployed using ME. In OC-PSIF-2S, subtasks enter the second stage after 500 episodes. This experiment validates the policy suboptimality of the OC baseline in the *Fourrooms* problem: the MDP-optimal policies $\pi^*$ obtained via the two improvement frameworks achieve higher expected returns $J_{\mathcal{A}}$ under ME deployment compared with the SMDP-optimal policies $\pi^-$ and their expected returns $J_{\mathcal{B}}$ under ME. For 8 options, OC-PSIF-2S completes the task in over 30% fewer steps than the OC baseline, and for 4 options, the reduction exceeds 50%, indicating that the OC baseline suffers from stronger SMDP suboptimality when fewer options.

**Analysis of Differences Between the Two PSIF Variants.** We observe that PSIF-1S achieves better final performance than PSIF-2S upon deployment. This is because in PSIF-2S, the low-level node policies are fixed during the second-stage training. However, when $\hat{\pi}_{hi}$ updates from $\hat{\pi}_{hi}^-$ to $\hat{\pi}_{hi}^*$ upon entering the second stage, the state distribution of the agent $s$ changes, and the frequency of subtask $g$ called also changes. This causes a more severe $(s, g)$ distribution shift for the low-level policy $\hat{\pi}_{lo}(s, g)$, degrading the performance of the learned low-level policies and limiting the achievable final performance of the high-level tasks. This issue does not occur in the PSIF-1S framework.

We define the set of already-trained subtasks among the subtasks of $\tau$ as the *subtask support* of $\tau$. Since employing different policies for $\tau$ results in different distributions over its subtasks during training, the obtained subtask support also differs accordingly. Comparing the performance curves of PSIF-1S and PSIF-2S in Fig. 5(b), the expected return $J_{\mathcal{A}}$ obtained with $\hat{\pi}_{hi}^*$ subtask support is higher than that obtained with $\hat{\pi}_{hi}^-$ subtask support. The advantages of PSIF-1S over PSIF-2S are: 1) It does not require manual determination of whether a node policy has reached $\pi_\tau^*$, i.e., when a task enters the second stage. 2) PSIF-1S learns the subtask support of $\pi_\tau^*$ without being constrained by the subtask support of $\pi_\tau^-$.

**Limitations.** ESIF can only address execution suboptimality and is ineffective for policy suboptimality. In PSIF-2S, during second-stage training, node policies that have already entered the second stage suffer from state-task distribution shift. PSIF-1S, on the other hand, faces an off-policy prob-

---

[5]For the single-layer RL baseline ablation experiments that were truncated due to zero success rate and excessively long episode lengths, we display them only as numerical values in the legend so as not to affect the clarity of the main curves.

lem during training due to the mismatch between the target execution mode (using ME) and the behavior execution mode (using SME).

This paper revises the execution modes and training objectives for HRL, but efficiently computing these training objectives for differentiated algorithms remains an open problem. For instance, in the goal-based PSIF-2S method, the challenge lies in subtask dependencies: during training, subtask frequencies are concentrated on those commonly called by the SME-optimal node policy, leading to insufficient learning of subtasks required by the ME-optimal node policy. PSIF-1S needs to address the off-policy problem of SME data. Furthermore, extending SMDP suboptimality improvement frameworks to policy-gradient-based HRL algorithms is a promising direction for future research.

## 7 RELATED WORK

**Optimality in SMDP.** Wen et al. (2020) provide optimality guarantees for planning in HRL, but the number of subtasks required to guarantee optimality grows exponentially and becomes impractical. Nachum et al. (2019) studies the impact of subtask space representation on suboptimality, which represents another type of suboptimality in HRL. Sutton et al. (1999) also reveals the existence of suboptimality in SMDP under the option framework, but only identifies *execution suboptimality* while overlooking *policy suboptimality*. In contrast, this paper further proves the *Optimal Execution Theorem*, which formally establishes the value-optimal execution mode and provides theoretical guarantees for general hierarchical policy improvement. We also provide a generalized multi-level proof and counterexamples for SMDP suboptimality, reducing the problem to the exploration–exploitation tradeoff in HRL. Recent work Drappo et al. (2023) analyzed regret bounds in finite-horizon SMDPs, showing how option duration shapes performance. Meanwhile, HiT-MDP (Li et al., 2023) reformulated option-induced processes as equivalent MDPs to stabilize optimization. These studies reveal the tradeoffs of temporal abstraction but not take into account that SME and ME should differ in the behavior policy, target policy, and deployment policy.

**Exploration in SMDP.** Potential algorithms in HRL that can be adapted to our improvement frameworks cover a wide range of research directions, including autonomously creating option policies from data (Harutyunyan et al., 2019; Riemer et al., 2020; Klissarov & Precup, 2021; Zhang et al., 2021; Lin et al., 2024; Nayyar & Srivastava, 2025), skill discovery (Menache et al., 2002; Bagaria & Konidaris, 2020; Eysenbach et al., 2019; Tang et al., 2018a), multi-task or multi-agent learning (Igl et al., 2020; Liu et al., 2021; Yang et al., 2019; Tang et al., 2018b), and goal-space learning (Andre & Russell, 2002; Ghosh et al., 2018; Zhang et al., 2020). These algorithms consistently demonstrate that HRL can significantly improve efficiency in long-horizon decision-making problems, and Wen et al. (2020); Robert et al. (2023) also provide theoretical guarantees on the sample efficiency of goal-based HRL. However, most of these algorithms are based on the SMDP framework and therefore inherently suffer from the SMDP suboptimality issues studied in this work.

**Value formalization in HRL.** Schaul et al. (2015) proposed the concept of a unified value function, and Bacon et al. (2017) share similar ideas, but these approaches cannot represent multi-level policy values under arbitrary execution modes.

## 8 CONCLUDING REMARKS

**Summary:** 1) **Abstraction:** The Task Tree and the Execution Tree extend the HRL framework. This formalism enables us to decouple hierarchical task decomposition from hierarchical policy scheduling, thereby facilitating a more principled design of HRL algorithms. 2) **Formalization:** UVFH and GHBE provide unified analytical tools for HRL, which can represent and compute multi-level values under arbitrary execution modes. 3) **Theorems:** The suboptimality theory of SMDP establishes the inherent suboptimality of existing HRL algorithms, while the Hierarchical Policy Improvement Theorem and the Optimal Execution Theorem guarantee the effectiveness of our improvements. 4) **Frameworks:** We propose three improvement frameworks that decouple the execution modes of the target policy and the behavior policy, thereby achieving a more balanced exploration-exploitation tradeoff. Experiments demonstrate consistent performance gains.

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

# A NOTATIONS

Table 2: Notations used in this paper.

| Symbol | Description |
|---|---|
| $\mathcal{S}$ | State space |
| $\mathcal{A}$ | Action space |
| $T : \mathcal{S} \times \mathcal{A} \to \Delta(\mathcal{S})$ | Transition kernel of MDP |
| $R_\tau : \mathcal{S} \times \mathcal{A} \to \mathbb{R}$ | Per-frame reward function of task $\tau$ |
| $\hat{R}_\tau : \mathcal{S} \times \mathcal{T}'_\tau \to \mathbb{R}$ | Exit reward function of task $\tau$ |
| $\alpha_\tau \in \Delta(\mathcal{S})$ | Initial state distribution of task $\tau$ |
| $\beta_\tau : \mathcal{S} \to [0, 1]$ | Termination function of task $\tau$ (in execution tree) |
| $\mathcal{T}'_\tau$ | Subtask set of task $\tau$ |
| $\bar{T}_\tau : \mathcal{S} \times \mathcal{T}'_\tau \to \Delta(\mathcal{S})$ | Subtask transition function |
| $\bar{R}_\tau : \mathcal{S} \times \mathcal{T}'_\tau \to \mathbb{R}$ | Subtask reward function |
| $t$ | Frames number |
| $\hat{t}$ | Subtasks been called number |
| $\pi_\tau : S \to \Delta(\mathcal{A})$ | Policy of task $\tau$ (calls actions across full level) |
| $\hat{\pi}_\tau : S \to \Delta(\mathcal{T}'_\tau)$ | Node policy of task $\tau$ (calls subtasks) |
| $\pi$ | Policy set |
| $V_\pi(s\|\tau)$ | Value function of policy set $\pi$ under task $\tau$ |
| $J(\pi\|\tau)$ | Expected return of policy set $\pi$ under task $\tau$ |
| $\hat{\beta}_\tau : \mathcal{S} \to [0, 1]$ | Interrupt function of task $\tau$ (in execution tree) |
| $\pi^*$ | MDP-optimal policy set (all $\hat{\pi}_\tau$ optimal under ME) |
| $\pi^-$ | SMDP-optimal policy set (all $\hat{\pi}_\tau$ optimal under SME) |
| $J_{\mathcal{A}}, J_{\mathcal{B}}, J_{\mathcal{C}}, J_{\mathcal{D}}$ | Expected returns under ME/SME with $\pi^*$ or $\pi^-$ |
| $\gamma, \hat{\gamma}$ | Discount factors for task termination / decision interruption |
| $N(i)$ | Set of node indices at depth $i$ in a tree |
| $n(i)$ | Node index at depth $i$ in a tree |
| $\cdot\|n(i)$ | Subtree rooted at $\cdot_{n(i)}$ |
| $\tau_{n(i)}$ | Task node at index $n(i)$ |
| $e_{n(i)}$ | Execution node at index $n(i)$ |
| $\Pi$ | Policy tree in the execution tree |
| $\mathbf{B}$ | Interruption tree in the execution tree |
| $\hat{s} : \left(s, \tau_{n(i)}\right)$ | Extended state |
| $V_e(\hat{s}\|\tau) : V_\Pi^{\mathbf{B}}(\hat{s}\|\tau)$ | Value function of execution tree $e$ under task $\tau$ (UVFH) |
| $\mathcal{F}_{n(h)}^{e,\tau}$ | The GHBE operator for executing $\tau_{n(h)}$ using $e\|n(h)$ |

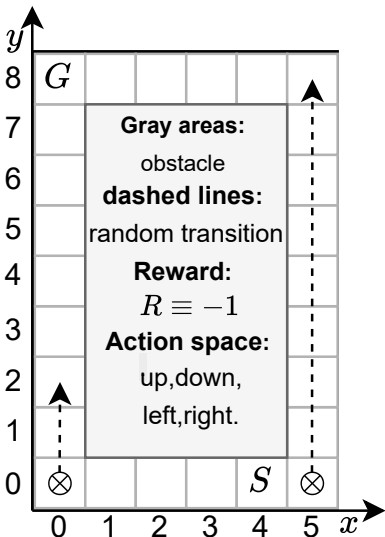

Figure 6: GridWorld.

## B  NAIVE CASE STUDY: GRIDWORLD

This is a $6 \times 8$ GridWorld, where the central $4 \times 6$ region is set as an obstacle. Under this configuration, the agent can only reach the goal through the left or right corridors. The action space of the agent is $\{\text{up}, \text{down}, \text{left}, \text{right}\}$, which allows the agent to move one cell in the corresponding direction. When the agent reaches the $\otimes$ region on the map, it has a $50\%$ probability of undergoing an additional state transition along the dashed line. During interaction with the environment, the agent starts from the starting point $S$ and receives a reward of $-1$ for each transition until it reaches the goal $G$.

Within the HRL framework, the agent in this example completes sub-tasks by calling actions, and accomplishes the overall task of reaching $G$ by calling these sub-tasks.

In this example, the sub-task space $G'$ is defined as:

$G' = \{g \in S \mid \text{sub-goal}^6 \ g \text{ is a state whose } L_1 \text{ distance from the current state is exactly } 5\}$,

where the $L_1$ distance is defined as

$$L_1(s_1, s_2) = |x_1 - x_2| + |y_1 - y_2|.$$

For instance, at the starting point $(4, 0)$, the only two selectable sub-goals are $(0, 1)$ and $(5, 4)$. In fact, under this setting, there are always exactly two selectable sub-goals from any state, which greatly facilitates our clear analysis of the problem.

In each sub-task, the agent will learn a policy that reaches the sub-goal as quickly as possible: starting from the initiation of the sub-task, the agent receives a reward of $-1$ for each transition until reaching the sub-goal $g$. By maximizing the cumulative reward, the optimal policy for each sub-task can be obtained. Since the shortest paths to each sub-goal in this example are straightforward, we omit the detailed explanation here.

### B.1  EXECUTION SUBOPTIMALITY

The figure illustrates the trajectories of an agent under SME and ME execution modes with a two-level policy set $\pi$. The high-level policy is denoted as $\hat{\pi}_{hi}(g \mid s, G)$, which selects an appropriate

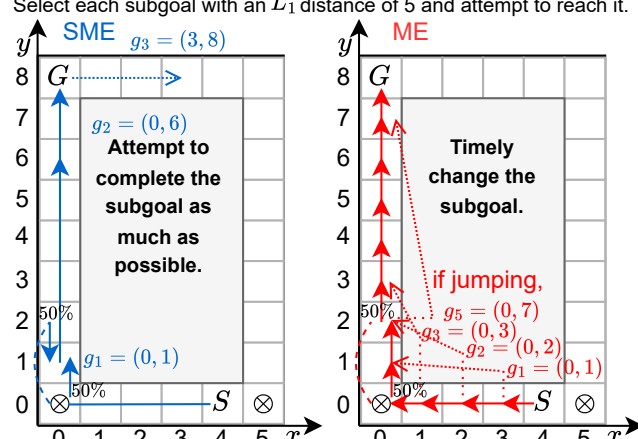

Figure 7: Execution Suboptimality.

Table 3: The mapping of $\hat{\pi}_{hi}$ and its value estimation under ME.

| $s$ | (4,0) | (3,0) | (2,0) | (1,0) | (0,0) | (0,1) |
|---|---|---|---|---|---|---|
| $g$ | (0,1) | (0,2) | (0,3) | (0,4) | (0,5) | (0,6) |
| $V(s)$ | -11 | -10 | -9 | -8 | -8 | -7 |
| $s$ | (0,2) | (0,3) | (0,4) | (0,5) | (0,6) | (0,7) |
| $g$ | (0,7) | (0,8) | (1,8) | (2,8) | (3,8) | (4,8) |
| $V(s)$ | -6 | -5 | -4 | -3 | -2 | -1 |

subgoal $g$ from the subtask space based on the current state $s$ in order to reach $G$ as quickly as possible. The low-level policy is denoted as $\hat{\pi}_{lo}(a \mid s, g)$, which selects an appropriate action $a \in \{\text{up}, \text{down}, \text{left}, \text{right}\}$ based on the current state $s$ to reach $g$ as quickly as possible.

The mapping of $\hat{\pi}_{hi}$ and its value estimation under ME are as Table 3 (only listing the left-side path):

The trajectory of the agent using SME is marked by the blue arrows on the left side of Fig. 7. $\hat{\pi}_{hi}$ will sequentially select three subgoals $(0,1), (0,6), (3,8)$ and call $\hat{\pi}_{lo}$ to complete these subgoals, eventually reaching $G$.

The trajectory of the agent using ME is marked by the orange arrows on the right side of Fig. 7. At each frame, it selects a new optimal subgoal $g_t$ through the high-level policy $\hat{\pi}_{hi}$, such as $(0,1), (0,2), \ldots$, and executes the optimal action $a_t$ for $g_t$ through $\hat{\pi}_{lo}$ for one frame.

If the agent does not encounter the stochastic transition event at $\otimes$, then both SME and ME agents will require 12 frames to reach $G$, yielding a return of $-12$.

If the agent encounters a random transition event at $\otimes$ and is additionally teleported to $(0,2)$, the two execution modes will diverge: the agent using SME will continue executing the unfinished subtask $(0,1)$ and choose the optimal action down under this subtask; in contrast, the agent using ME will, at $(0,2)$, reselect a new optimal subtask $(0,7)$ via $\hat{\pi}_{hi}$ and choose the optimal action up under this subtask. In this case, the agent with SME still requires 12 frames to reach $G$, while the agent with ME only requires 10 frames to reach $G$.

Thus,

$$J^{ME}(\pi|G) = -12 \times 0.5 - 10 \times 0.5 = -11 > -12 = -12 \times 0.5 - 12 \times 0.5 = J^{SME}(\pi|G).$$

In the table,

$$V(s = (1,0)) = -1 + 0.5 \times V(s' = (0,0)) + 0.5 \times V(s' = (0,2)) = -1 - 0.5 \times 8 - 0.5 \times 6 = -8.$$

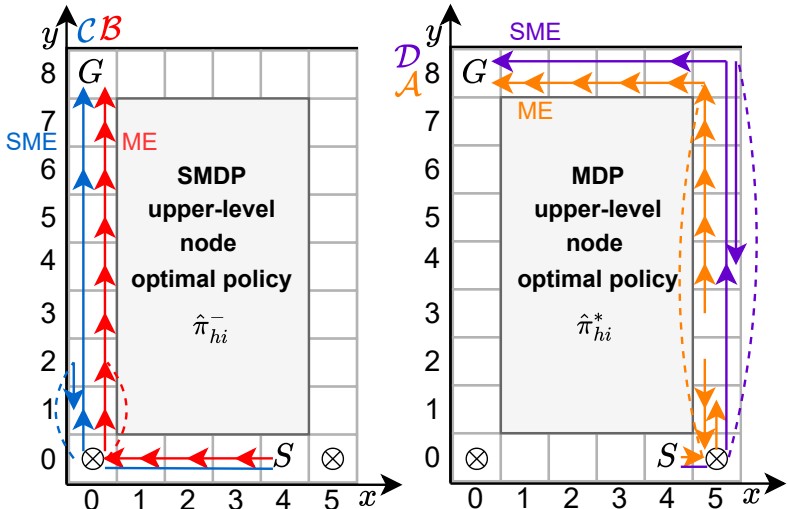

Figure 8: Policy Suboptimality.

Table 4: The mapping of $\hat{\pi}_{hi}^*$ and its value estimation under ME (right-hand path).

| $s$ | (4,0) | (5,0) | (5,1) | (5,2) | (5,3) | (5,4) | (5,5) |
|---|---|---|---|---|---|---|---|
| $g$ | (5,4) | (5,5) | (1,0) | (2,0) | (5,8) | (4,8) | (3,8) |
| $V(s)$ | -8 | -9 | -8 | -9 | -10 | -9 | -8 |
| $s$ | (5,6) | (5,7) | (5,8) | (4,8) | (3,8) | (2,8) | (1,8) |
| $g$ | (2,8) | (1,8) | (0,8) | (0,7) | (0,6) | (0,5) | (0,4) |
| $V(s)$ | -7 | -6 | -5 | -4 | -3 | -2 | -1 |

## B.2 POLICY SUBOPTIMALITY

Now consider an agent with a two-level policy set $\pi^*$, the mapping of $\hat{\pi}_{hi}^*$ and the value estimation under ME are as Table 4 (right-hand path) and Table 5 (left-hand path):

The agent's trajectory under SME is indicated by the purple arrows on the right of Fig. 8. It sequentially selects three sub-tasks $(5,4), (4,8), (0,7)$ and executes them using $\hat{\pi}_{lo}^*$, eventually reaching $G$. The trajectory under ME is indicated by the yellow arrows on the right of Fig. 8; at each frame, the agent selects a new sub-task $g_t$: $(5,4), (5,5)$ or $(0,8), \ldots$ via the high-level policy $\hat{\pi}_{hi}^*$, and executes a single-frame optimal action $a_t$ under $\hat{\pi}_{lo}^*$.

The expected returns of the two trajectories are denoted as $J_{\mathcal{D}} = J^{SME}(\pi^*|G)$ and $J_{\mathcal{A}} = J^{ME}(\pi^*|G)$. The original policy set $\pi$ from Appendix B.1 is denoted as $\pi^-$, with its expected returns under SME and ME denoted as $J_{\mathcal{C}} = J^{SME}(\pi^-|G)$ and $J_{\mathcal{B}} = J^{ME}(\pi^-|G)$, with trajectories shown on the left side of Fig. 8.

The calculation of $J^{SME}(\pi^*|G)$ is relatively straightforward:

$$J_{\mathcal{D}} = J^{SME}(\pi^*|G) = -14 \times 0.5 - 14 \times 0.5 = -14.$$

The expected return under ME is more complex, as the agent may repeatedly leverage stochastic transitions to reach the goal faster:

$$J_{\mathcal{A}} = J^{ME}(\pi^*|G) = -1 - \text{Time}[(5,0) \sim (5,8)] - 5 = -6 - \sum_{i=0}^{\infty}(0.5)^{i+1}(2 \times i) = -6 - 2 = -8.$$

In this example, under SME, the agent's $\hat{\pi}_{hi}$ can only make decisions at the five positions $(4,0), (0,1), (0,6), (5,4), (4,8)$. Among the $2^5 = 32$ possible node policies, only $\hat{\pi}_{hi}^-$ and $\hat{\pi}_{hi}^*$

Table 5: The mapping of $\hat{\pi}_{hi}^*$ and its value estimation under ME (left-hand path).

| $s$ | (4,0) | (3,0) | (2,0) | (1,0) | (0,0) | (0,1) |
|---|---|---|---|---|---|---|
| $g$ | (5,4) | (5,3) | (0,3) | (0,4) | (0,5) | (0,6) |
| $V(s)$ | -8 | -9 | -9 | -8 | -8 | -7 |
| $s$ | (0,2) | (0,3) | (0,4) | (0,5) | (0,6) | (0,7) |
| $g$ | (0,7) | (0,8) | (1,8) | (2,8) | (3,8) | (4,8) |
| $V(s)$ | -6 | -5 | -4 | -3 | -2 | -1 |

considered in this paper have finite expected returns. It is thus easy to verify that $\hat{\pi}_{hi}^-$ is the node policy that maximizes the expected return from $S$ to $G$ under SME[7], i.e., $\hat{\pi}_{hi}^-$ is the SMDP optimal node policy as defined in Definition 3.3.

It can be verified that the value under ME using $\pi^*$ satisfies the Bellman optimality equation, which proves that $\pi_{hi}^*$ is the optimal policy in the single-level RL case. Therefore, $\hat{\pi}_{hi}^*$ is also the node policy that achieves the highest expected return under ME, i.e., $\hat{\pi}_{hi}^*$ is the MDP optimal node policy as defined in Definition 3.4.

When the agent calls $\pi^-$ under SME, we have $J_{\mathcal{C}} = -12$, and for any policy $\pi$ with a different high-level node strategy, $J^{SME}(\pi|G) \leq J_{\mathcal{C}}$, e.g., $J_{\mathcal{D}} = J^{SME}(\pi^*|G) = -14 < J_{\mathcal{C}}$.

When the agent calls $\pi^*$ under ME, we have $J_{\mathcal{A}} = -8$, and for any policy $\pi$ with a different high-level node strategy, $J^{ME}(\pi|G) \leq J_{\mathcal{A}}$, e.g., $J_{\mathcal{B}} = J^{ME}(\pi^-|G) = -11 < J_{\mathcal{A}}$.

## C  FORMALIZATION AND THEORETICAL PROOF

### C.1  TASK TREE AND EXECUTION TREE

This work observes that each task $\tau$ has two distinct termination functions. One controls the optimization objective of $\hat{\pi}_\tau$, i.e., the **termination function** $\beta_\tau$ defined in Section 2.2. The other controls the duration for which $\tau$ executes when it is called as a subtask during interaction, and is defined as the **interrupt function** $\hat{\beta}_\tau$.

In HRL, both for the execution of the behavioral policy and the target policy, the number of frames a subtask is actually executed is simply tied to the subtask's own duration, i.e., $\hat{\beta}_\tau \equiv \beta_\tau$. From the analysis in Appendix B, we recognize that this is unreasonable.

To address this, we propose the **task tree** and **execution tree**, which independently describe, respectively, the complex task optimization objectives and task calling mechanisms in multi-level hierarchies. The most crucial aspect is the precise formalization and proper design of hierarchical boundary conditions when a task terminates.

We standardize the notation for node indices used in the tree structures in this paper asDefinition C.1:

**Definition C.1 (Node indices in the tree).** *Let $N$ be the set of all node indices in the tree, and $N(i) \subset N, i \geq 0$ denote the set of indices of all nodes at depth $i$ in the tree. We use the variable $n(i) \in N(i)$ as the index of a node at depth $i$, $n(i-1)$ as the index of its parent node, and $n(i+1)$ as the index of one of its child nodes. Given a path $L$ on the tree, $n(i,L) \in N(i) \cap L$ denotes the index of the node at depth $i$ on path $L$. The set of leaf node indices is denoted by $N_{leaf}$. In this paper, if the variable $x$ represents a tree, we use $x_{n(i)}$ to denote the node corresponding to index $n(i)$, and $x|n(i)$ to denote the subtree rooted at $x_{n(i)}$.*

#### C.1.1  TASK TREE

In this paper, we use the concept of a task tree to formally describe the hierarchical nested calling relationships among multiple tasks in HRL, and to specify the optimization objective for each task.

**Definition C.2.** *Task tree We extend the task $\tau$ in RL into a task tree $\tau$. A task tree $\tau$ is a tree structure composed of task nodes $\tau_{n(i)}, n(i) \in N$. It describes the calling relationships among*

---

[7]We do not further elaborate on the SMDP optimality of $\hat{\pi}_{hi}^-$ at other states, as the proof follows in a similar manner.

*tasks and the nested relationships among the optimization objectives of the node policies of each task.*

**Definition C.3** (**Task Node**). [8] *A task node $\tau_{n(i)}$ is defined as a triple $(\mathcal{T}', \bar{\beta}, R)_{n(i)}$ consisting of:*

- $\mathcal{T}'$**: *The set or space of child task nodes.*** *The parent-child relationships between task nodes form the tree structure of the task tree.*

- $\beta$**: *The task termination function.*** $\beta_{n(i)}(\cdot) : \mathcal{S} \to [0,1]$ *represents the probability that $\tau_{n(i)}$ generates a task termination event when the agent reaches state $s'$. This is an extension of the termination function in Section 2.1 of the main text. As will be shown in **??**, the termination of a task at each frame is jointly determined by its own and all its ancestor task nodes' termination events, which defines the temporal scope of reward accumulation.*

- $R$**: *The per-frame reward function.*** $R_{n(i)}(\cdot, \tau_{leaf}) : \mathcal{S} \to \mathbb{R}$ *provides a per-frame reward whenever the agent takes an action $a$ at state $s$ during the interaction process of $\tau_{n(i)}$.*

- $\hat{R}$**: *The exit reward function.*** $\hat{R}_{n(i)}(\cdot, \tau_{n(j)}) : \mathcal{S} \to \mathbb{R}, \; -1 \le j < i$, *provides a boundary reward for the entire process when $\tau_{n(i)}$ exits to $\tau_{n(j)}$ due to its own or an ancestor node's termination event.*

**Definition C.4** (**Task termination event**). *For a task node $\tau_{n(i)}$, if a task termination event is triggered by itself or any of its ancestor nodes $\tau_{n(j)}$, $0 \le j \le i$, then the termination event of the highest-level ancestor $\tau_{n(j_{\min})}$ takes effect. In this case, the task node $\tau_{n(i)}$ is terminated and exits to its ancestor node $\tau_{n(j_{\min}-1)}$. This mechanism ensures that when an ancestor task is terminated, all of its descendant tasks are also terminated, thereby maintaining the nested structure among tasks.*

The task tree functions in the following two phases: 1) **Interaction phase per frame:** After the agent executes an action $a$, each task node $\tau_{n(i)}$ will individually obtain a per-frame reward for this task through its per-frame reward function $R_{n(i)}(s, a)$. 2) **Termination phase per frame:** All task nodes in the task tree will independently trigger task termination events according to their termination functions. Based on the triggered termination events, Definition B.4 determines whether each task is terminated. If a task $\tau_{n(i)}$ is terminated, the accumulation of per-frame rewards ends, and an exit reward $R_{n(i)}(s', \tau_{n(j_{\min}-1)})$ is provided by the exit reward function. Otherwise, the task $\tau_{n(i)}$ continues.

The task tree defines the optimization objective of each task node's policy, which is to maximize the expected return $\mathbb{E}_{s_0=s}\left[ \sum_{t=0}^{T-1} R_{n(r)}(s_t, a_t) + \hat{R}_{n(r)}(s_T, \tau_{n(j_{\min}-1)}) \right]$ starting from any state $s$, where $T$ is the number of frames when the task is terminated. Unlike standard RL, for tasks in HRL we must also consider the hierarchical structure at the task's termination. Specifically, when different termination events occur, they should correspond to different task boundary conditions, as in the option framework.

**Example: Option framework.** The root task has a countable set of subtasks (options). The termination function of the root task is given by the environment. The per-frame reward of the root task is defined as $R_{n(0)}(s, \tau_{leaf}) = R(s, a)$, which is the environmental reward, and its exit reward is $R_{n(0)}(s, \tau_{n(-1)}) \doteq 0$. For each option, its subtasks are the actions. The termination function of an option is learned via the termination gradient theorem. The per-frame reward of the option is $R_{n(1)}(s, \tau_{leaf}) = R(s, a)$, and its exit rewards are $R_{n(1)}(s, \tau_{n(-1)}) \doteq 0$ and $R_{n(1)}(s, \tau_{n(0)}) \doteq V(s)$, where $V(s)$ denotes the state value function of the root task. By setting the exit reward to the value function of the upper-level task, the lower-level policy is optimized to improve the upper-level task.

---

[8]There are two special types of nodes in the task tree:

- **External task** $\tau_{out} = \tau_{n(-1)} : (\mathcal{T}'_{out}, /, /, /)$, where $\mathcal{T}'_{out} = \{\tau_{n(0)}\}$. This node does not require a policy optimization objective and is used solely to call the root task and wait for its termination.

- **Action** $\tau_{leaf} : (/, \beta_{leaf}, /, /)$, where $\beta_{leaf} \equiv 1$. This node also does not require a policy optimization objective and is used to produce a single-frame state transition.

**Example: Goal-based framework.** In the Goal-based framework, the root task has a sub-task space (sub-goal space). The termination function of the root task is given by the environment. The per-frame reward of the root task is $R_{n(0)}(s, \tau_{leaf}) = R(s, a)$, and its exit reward is $R_{n(0)}(s, \tau_{n(-1)}) \doteq 0$. For each goal, its subtasks are the actions. The termination function of a goal is fixed as $\bar{\beta}(s') \doteq \mathbb{I}[d(s', g) < d_0]$, where $d$ is a distance metric and $d_0$ is a hyperparameter related to neighborhood determination. The reward tree of a goal is provided by itself: its per-frame reward $R_{n(1)}(s, \tau_{leaf})$ is defined by the goal, and its exit rewards are $R_{n(1)}(s, \tau_{n(-1)}) \doteq 0$ and $R_{n(1)}(s, \tau_{n(0)}) \doteq 0$. By setting the exit reward to zero, the lower-level tasks become feudal, i.e., independent from the upper-level value.

### C.1.2 EXECUTION TREE

This paper introduces the concept of an execution tree to describe how an agent in HRL executes its hierarchical policies to interact with the environment. We observe a fundamental difference between HRL and standard RL: during the execution process of HRL, in addition to each task node having its own node policy function, an extra *execution mechanism* is required to properly organize and call these node policies at every frame. This is the key point discussed in Section 3.1 of the main text.

**The differences in execution mechanisms can significantly affect the agent's expected return, its exploration capability, and the data efficiency of each node policy during the learning process.**

Therefore, the choice of execution mechanisms for the agent's behavior policy during training, the target policy during training, and the policy during deployment are all crucial research questions that have been largely overlooked.

**Definition C.5** (**Execution Tree**). *An execution tree $e$ is a tree structure composed of execution nodes $e_{n(i)}, n(i) \in N$, used to control the agent's level-wise decision-making to call actions and complete various tasks in the task tree. Except for the two special nodes in the task tree, each execution node $e_{n(i)}$ corresponds one-to-one with a task node $\tau_{n(i)}$. The execution subtree $e|n(i)$ is used to execute the corresponding task $\tau_{n(i)}$.*

**Definition C.6** (**Execution Node**). *An execution node $e_{n(i)}$ is composed of a tuple $(\hat{\pi}, \hat{\beta})_{n(i)}$:*

- *$\hat{\pi}$: **Node policy function.** $\hat{\pi}_{n(i)}(\cdot) : \mathcal{S} \to \Delta\mathcal{T}'_{n(i)}$ (or equivalently $\hat{\pi}_{n(i)}(\cdot|s) : \mathcal{T}'_{n(i)} \to [0, 1]$), which allows task $\tau_{n(i)}$ to select a subtask when at state $s$.*

- *$\hat{\beta}$: **Decision interruption function.** $\hat{\beta}_{n(i)}(\cdot) : \mathcal{S} \to [0, 1]$ denotes the probability that $e_{n(i)}$ generates a decision interruption event after the agent reaches state $s'$.*

**Definition C.7** (**Decision Interruption Event**). *For an execution node $e_{n(i)}$, if a decision interruption event occurs at $e_{n(i)}$ itself or any of its ancestors $e_{n(j)}, 0 \leq j \leq i$, then the interruption from the highest-level ancestor $e_{n(j_{\min})}$ takes effect. The decision authority of execution node $e_{n(i)}$ is then transferred to the ancestor node $e_{n(j_{\min}-1)}$.*

**Definition C.8** (**Policy Tree**). *The tree structure in the execution tree $e$ formed by the node policy functions is referred to as the policy tree $\Pi$.*

**Definition C.9** (**Interruption Tree**). *The tree structure in the execution tree $e$ formed by the decision interruption functions is referred to as the interruption tree $\mathbf{B}$.*

When the agent interacts with the environment using the execution tree, only the execution nodes that currently hold **decision authority** can call their node policies to make decisions. Taking the interaction of executing task $\tau_{n(h)}$ using the execution subtree $e|n(h)$ as an example, the execution tree operates in the following two phases: 1) **Preparation phase of each frame:** At frame 0, $e_{n(h)}$ directly obtains decision authority. In subsequent frames, the decision authority initially resides with the execution node $e_{n(l)}$ that made the decision in the previous frame. All execution nodes within the subtree $e|n(h)$, *except the root $e_{n(h)}$*, then generate decision interruption events according to their interruption functions. The decision authority is transferred based on Definition B.7. If $e_{n(l)}$ is interrupted, the authority is passed to its ancestor $e_{n(j_{min}-1)}$; otherwise, it remains with $e_{n(l)}$. 2) **Decision phase of each frame:** The execution node holding the decision authority selects a subtask according to its node policy function and continuously passes the decision authority to the

execution node corresponding to the selected subtask. If there is no corresponding execution node (e.g., for actions), the authority is not transferred. This process continues until an action is selected and executed.

At this point, we have decoupled the original termination function $\beta$ under the definitions of the task tree and execution tree: $\beta$ now controls the intrinsic optimization objective of the task, while $\hat{\beta}$ controls the calling of node policies during execution.

In the task tree, the optimal subtask set for facilitating task learning can be found by optimizing $\beta$ and $R$. In existing algorithms, some $\beta$ functions are fixed; for example, in the Goal-based framework, each task (goal) has a fixed $\beta$. Others are learnable; for instance, in the OC method, each task (option) has a $\beta$ optimized via the termination gradient theorem.

In the execution tree, the optimization of $\hat{\beta}$ must balance the exploration and exploitation of task $\tau_{n(h)}$ during both behavior policy and target policy calling. On one hand, it should consider interacting with the environment under a certain execution mode to achieve high sampling efficiency (exploration). On the other hand, it should consider planning an optimal policy tree under the given execution mode using existing data to ensure maximal expected return (exploitation)[9].

Only by decoupling the intrinsic optimization objective of the task, the execution mode of the behavior policy, and the execution mode of the target policy can the HRL field further address and study the exploration-exploitation tradeoff during the learning process.

For the sake of formal convenience in the proofs of subsequent theorems, we extend and decouple the discount factor $\gamma$ in the same manner as the termination function $\beta$. Unlike $\beta$ and $\hat{\beta}$, which are bound to each node, $\gamma$ and $\hat{\gamma}$ are defined to directly describe task transitions or decision transitions during the interaction process.

**Definition C.10** (**Task Transition Mechanism** $\gamma$). *Let $\bar{\gamma}_{n(i)}^{n(h)}$ denote the probability that the currently executing task $\tau_{n(h)}$ exits to its ancestor $\tau_{n(i)}, -1 \leq i \leq h$ during each frame's termination phase based on the task termination events.*

$\gamma_{n(i)}^{n(h)}$ can be computed from $\beta$: when exiting to $\tau_{n(i)}, -1 \leq i < h$, $\gamma_{n(i)}^{n(h)}$ is the probability that along the path from $\tau_{n(h)}$ to the root task $\tau_{n(0)}$, $\tau_{n(i+1)}$ triggers a termination event while all its ancestors do not trigger any termination events, i.e.,

$$\gamma_{n(i)}^{n(h)} = \beta_{n(i+1)} \prod_{k=0}^{i} (1 - \beta_{n(k)}).$$

When $\tau_{n(h)}$ does not exit,

$$\gamma_{n(h)}^{n(h)} = 1 - \sum_{k=0}^{h-1} \gamma_{n(k)}^{n(h)}.$$

In this paper, when task $\tau_{n(h)}$ is clear from context, the superscript of $\gamma_{n(i)}^{n(h)}$ is omitted for simplicity, and we write it as $\gamma_{n(i)}$.

**Definition C.11** (**Decision transition mechanism** $\hat{\gamma}$). *Let $\hat{\gamma}_{n(i)}^{n(h)}$ denote the probability that, when executing the task $\tau_{n(h)}$ using the execution subtree $e|n(h)$, the decision authority at the preparation stage of each frame transfers from the execution node $e_{n(l)}$ that made the final decision in the previous frame to $e_{n(i)}$.*

$\hat{\gamma}_{n(i)}^{n(h)}$ can be computed from $\hat{\beta}$: when the decision authority transfers to $e_{n(i)}, h \leq i < l$, $\hat{\gamma}_{n(i)}^{n(h)}$ is the probability that, along the path from $e_{n(l)}$ to $e_{n(h)}$, $e_{n(i+1)}$ triggers a decision interruption while

---

[9]Analysis shows that there is room for improvement in the design of OC here. The termination gradient theorem should be used for "exploitation" to learn $\hat{\beta}$ in order to maximize expected return, rather than to modify the subtasks' $\beta$. When $\beta$ approaches 0, subtasks rarely terminate, causing the subtask learning difficulty to approach that of the original task. Conversely, when $\beta$ approaches 1, subtasks terminate almost every frame, preventing the higher-level task from effectively leveraging the subtask for efficient exploration.

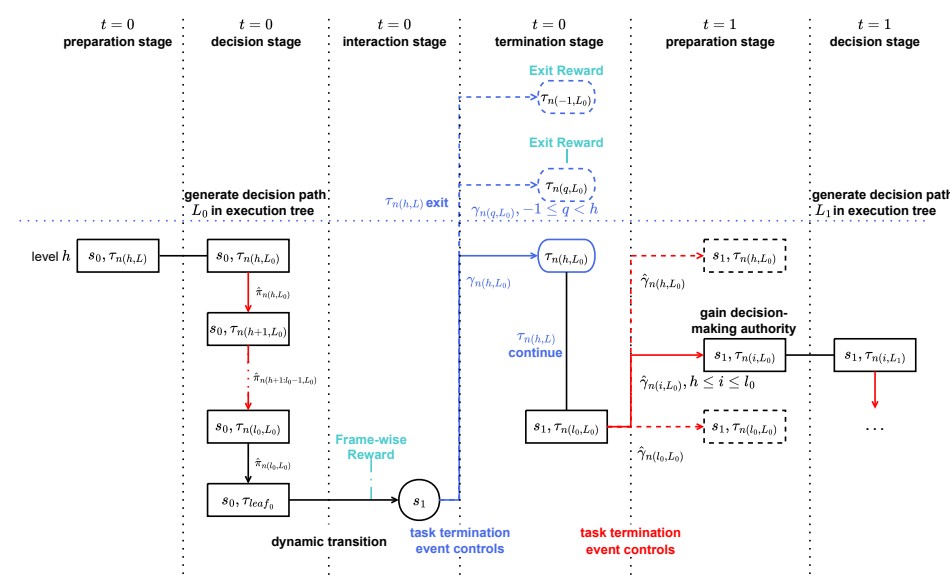

Figure 9: SME interaction process.

none of its ancestors do, i.e.,

$$\hat{\gamma}_{n(i)}^{n(h)} = \hat{\beta}_{n(i+1)} \prod_{k=h+1}^{i} (1 - \hat{\beta}_{n(k)}).$$

When the authority does not transfer,

$$\hat{\gamma}_{n(l)}^{n(h)} = 1 - \sum_{k=h}^{l-1} \hat{\gamma}_{n(k)}^{n(h)}.$$

When the task $\tau_{n(h)}$ is clear from context, we omit the superscript and write $\hat{\gamma}_{n(i)}^{n(h)}$ simply as $\hat{\gamma}_{n(i)}$.

### C.1.3 TWO SPECIAL EXECUTION MODES: SME AND ME

Existing hierarchical reinforcement learning algorithms often execute their policy trees using the SME mode, without decoupling the task termination functions. The transfer of decision authority is controlled by task termination events, and the decision interruption events are bound to the task termination events. In the computation of the decision transition mechanism $\hat{\gamma}$, it holds that $\hat{\beta} \doteq \beta$.

If all node policies within a policy subtree $\Pi|n(h)$ are called under SME, it is referred to as SME called the policy subtree $\Pi|n(h)$. The interaction process when an agent uses SME to execute task $\tau_{n(h)}$ with the policy subtree $\Pi|n(h)$ is illustrated in Fig. 9:

Since under SME the transfer of decision authority is controlled by task termination events, during the SME call of a policy subtree $\Pi|_{n(h')}$ with $h' < h$ to execute task $\tau_{n(h)}$, if a task termination event causes the decision authority to exit the execution subtree $e|_{n(h)}$, then the task $\tau_{n(h)}$ will also be terminated. Therefore, for the entire execution of task $\tau_{n(h)}$, it is equivalent to calling the policy subtree $\Pi|_{n(h)}$ under SME. Consequently, when the agent interacts with the environment using the SME call of the policy subtree $\Pi|_{n(h')}$ as the behavior policy, all descendant tasks $\tau_{n(h)}$ can obtain the complete trajectory data corresponding to executing $\tau_{n(h)}$ under an SME call of $\Pi|_{n(h)}$.

After decoupling the termination functions, the agent's decision interruption events and task termination events become independent. When the agent executes task $\tau_{n(h)}$ by calling the policy subtree $\Pi|_{n(h)}$ under ME, we have $\gamma_{n(h)} \equiv 1$.

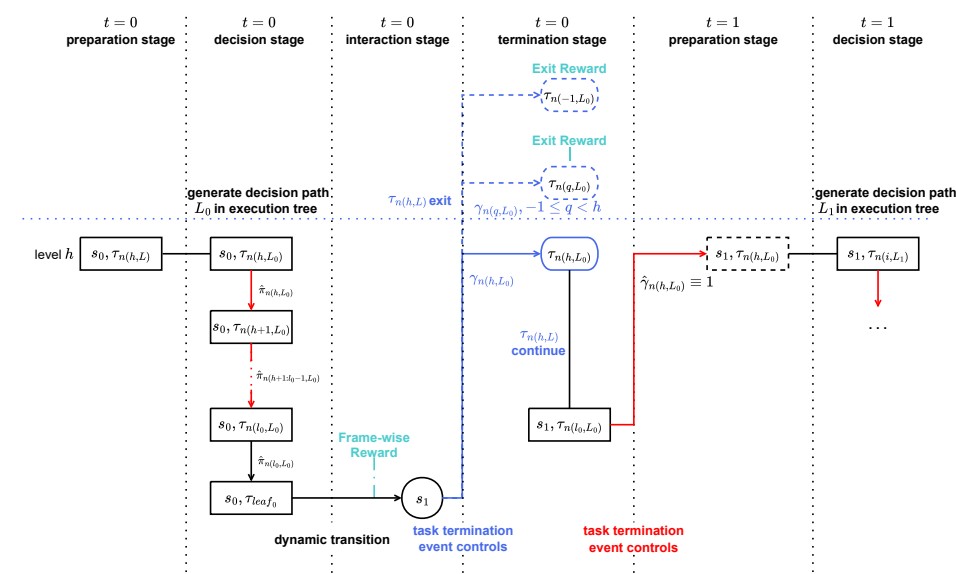

Figure 10: ME interaction process.

If all node policies within the policy subtree $\Pi|_{n(h)}$ are called under ME, the subtree is referred to as a ME called policy subtree $\Pi|_{n(h)}$. The interaction process when the agent executes task $\tau_{n(h)}$ using $\Pi|_{n(h)}$ under ME is illustrated in Fig. 10:

Unlike in SME, decision interrupt events are independent of task termination events in ME. During the execution of task $\tau_{n(h)}$ with the ME called policy subtree $\Pi|_{n(h')}, h' < h$, if the decision authority is transferred out of the execution subtree $e|_{n(h)}$ in the preparation phase due to a decision interrupt event while the task $\tau_{n(h)}$, controlled by task termination events, is not terminated, and in the next frame the decision authority is not returned to $e_{n(h)}$, as Fig. 11, this will cause an off-policy discrepancy between the target policy (decisions made downward from $e_{n(h)}$) and the behavior policy (decisions made downward from the execution node $e'_{n(h)}$ that holds the decision authority).

## C.2 HIERARCHICAL BELLMAN EQUATION

### C.2.1 UNIFIED VALUE FUNCTION IN HRL

We need to establish a consistent representation such that the value of any policy can be expressed under any execution mechanism. To this end, we construct a *unified value function* applicable to HRL under arbitrary execution modes, which describes the expected return across different settings.

**Definition C.12 (Unified Value Function for HRL, UVFH).** *The unified value function in HRL is denoted as $V_e(\hat{s}|\tau)$, where $\hat{s}$ is the extended state of the agent at the decision stage, and $\tau$ is the task node whose value we aim to evaluate.*

- ***Extended state*** $\hat{s} \dot{=} (s, \tau_{n(i)})$: *represents the agent being at state $s$ during the decision stage and the execution node $e_{n(i)}$ having the decision authority (or having selected an action).*

- ***task*** $\tau \dot{=} \tau_{n(h)}$: *specifies the scope of per-frame reward accumulation, the per-frame reward values, and the exit reward values.*

- ***execution tree*** $e$: *is the execution tree used by the agent.*

- ***unified value function*** *is defined as*

$$V_{\Pi}^{\mathbf{B}}(\hat{s}|\tau) = V_e(\hat{s}|\tau) = \mathbb{E}_{s_0=s, T\sim\tau, e}\left[\sum_{t=0}^{T-1} R_{n(r)}(s_t, a_t) + R_{n(h)}(s_T, \tau_{n(j_{min}-1)})\right],$$

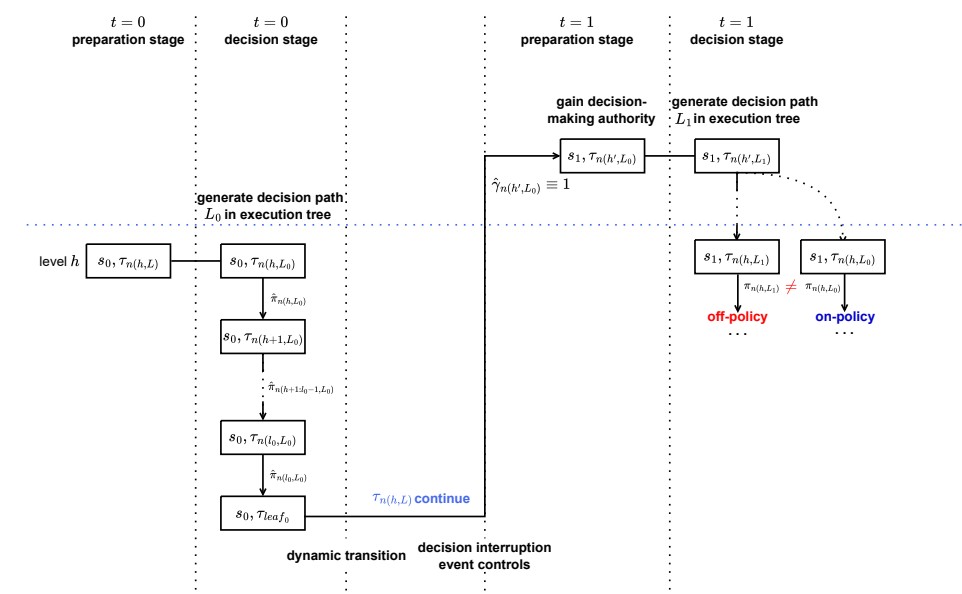

Figure 11: Off-policy issue in subtask node policy learning under ME.

> *which represents the cumulative sum of all rewards, including per-frame and exit rewards, when the agent uses execution tree $e$, starts from $\hat{s}$, and executes until task $\tau$ terminates.*

In particular, given a strategy tree $\Pi$, when the execution modes are SME or ME, the UVFH for these two special execution trees are denoted as $V_{\Pi}^{SME}$ and $V_{\Pi}^{ME}$, respectively.

The state value and state-action value in RL are unified into the value of the extended state $\hat{s}$ at the decision stage. This transforms the non-Markovian process in HRL, given $\tau$ and $e$, into an extended Markov process that is easier to analyze. Meanwhile, both the Goal-based framework and the Option framework in HRL are unified under the task node $\tau$.

### C.2.2 GENERALIZED HIERARCHICAL BELLMAN EQUATION

For each task node in the task tree, there exists a set of generalized Bellman equations for value estimation. When the agent executes a task $\tau_{n(h)}$, $h \geq h'$, using any given execution subtree $e|n(h')$ of $e$, we can iteratively compute the HRL unified value through the generalized hierarchical Bellman equation given $\tau_{n(h)}$ and $e|n(h')$, i.e., value evaluation:

**Definition C.13 (Generalized Hierarchical Bellman Equation, GHBE).** *Decision stage:*

$$V_{e|n(h')}(s, \tau_{n(i)}|\tau_{n(h)}) = \mathbb{E}_{\tau_{n(i+1)} \sim \hat{\pi}_{n(i)}} \left[ V_{e|n(h')}(s, \tau_{n(i+1)}|\tau_{n(h)}) \right], \quad \forall n(i)$$

*Interaction stage:*

$$V_{e|n(h')}(s, \tau_{leaf}|\tau_{n(h)}) = R_{n(h)}(s, a) + \mathbb{E}_{s' \sim T(s,a)} \left[ \bar{V}_{e|n(h')}(s', \tau_{leaf}|\tau_{n(h)}) \right]$$

*Termination stage:*

$$\bar{V}_{e|n(h')}(s', \tau_{leaf}|\tau_{n(h)}) = \hat{V}_{e|n(h')}(s', \tau_{leaf}|\tau_{n(h)})\gamma_{n(h)}(s') + \sum_{k=-1}^{h-1} \gamma_{n(k)}(s')\hat{R}_{n(h)}(s', \tau_{n(k)})$$

*Preparation stage:*

$$\hat{V}_{e|n(h')}(s', \tau_{leaf}|\tau_{n(h)}) = \sum_{k=h'}^{l} \hat{\gamma}_{n(k)}(s')V_{e|n(h')}(s', \tau_{n(k)}|\tau_{n(h)})$$

$\bar{V}$ and $\hat{V}$ are intermediate variables. In the termination stage equation, the first term on the right-hand side is the bootstrapped value when $\tau_{n(h)}$ has not yet terminated, and the second term is the exit reward provided by $\hat{R}_{n(h)}$ when $\tau_{n(h)}$ terminates and transfers to another task. In the preparation stage equation, the right-hand side represents the bootstrapped value when decision authority jumps occur due to decision interruption events.

In particular, since the execution subtree $e|n(h)$ will be used to execute task $\tau_{n(h)}$, we need to optimize the node policy $\hat{\pi}_{n(h)}$ given all decision interruption functions in $e|n(h)$ and all node policies except $\hat{\pi}_{n(h)}$, such that the expected return in $\tau_{n(h)}$

$$J(e|\tau_{n(h)}) = \mathbb{E}_{s \sim \alpha_{n(h)}} \left[ V_{e|n(h)}(s, \tau_{n(h)}|\tau_{n(h)}) \right]$$

is maximized.

If we want to optimize the node policy $\hat{\pi}_{n(h)}$ in $e|n(h')$, given all decision interruption functions and all node policies except $\hat{\pi}_{n(h)}$ for $h \geq h'$, such that the expected return in $\tau_{n(h')}$

$$J(e|\tau_{n(h')}) = \mathbb{E}_{s \sim \alpha_{n(h')}} \left[ V_{e|n(h')}(s, \tau_{n(h')}|\tau_{n(h')}) \right]$$

is maximized, this can be achieved via the Generalized Hierarchical Bellman Optimality Equation (GHBOE) for the optimization of $\hat{\pi}_{n(h)}$ given $\tau_{n(h')}$ and $e|n(h')$. We refer to this as the GHBOE for $\hat{\pi}_{n(h)}$. That is, in HRL, the optimization objective of a node policy can be set as maximizing the expected return of an ancestor task node $\tau_{n(h')}$, including its own task node. We denote the results of GHBOE iterations by adding superscripts to $V$, $\bar{V}$, and $\hat{V}$ as $V^*$, $\bar{V}^*$, $\hat{V}^*$ to indicate the computed values from the GHBOE rather than the value evaluation under a specific execution tree.

**Definition C.14** (**GHBOE for $\hat{\pi}_{n(h)}$**). *decision stage:*

$$V_{e|n(h')}^*(s, \tau_{n(h)}|\tau_{n(h')}) = \max_{\tau_{n(h+1)}} V_{e|n(h')}^*(s, \tau_{n(h+1)}|\tau_{n(h')})$$

$$V_{e|n(h')}^*(s, \tau_{n(i)}|\tau_{n(h')}) = \mathbb{E}_{\tau_{n(i+1)} \sim \hat{\pi}_{n(i)}} \left[ V_{e|n(h')}^*(s, \tau_{n(i+1)}|\tau_{n(h')}) \right], \quad \forall n(i) \neq n(h)$$

*interaction stage:*

$$V_{e|n(h')}^*(s, \tau_{leaf}|\tau_{n(h')}) = R_{n(h')}(s, a) + \mathbb{E}_{s' \sim T(s,a)} \left[ \bar{V}_{e|n(h')}^*(s', \tau_{leaf}|\tau_{n(h')}) \right]$$

*termination stage:*

$$\bar{V}_{e|n(h')}^*(s', \tau_{leaf}|\tau_{n(h')}) = \hat{V}_{e|n(h')}^*(s', \tau_{leaf}|\tau_{n(h')})\gamma_{n(h')}(s') + \sum_{k=-1}^{h-1} \gamma_{n(k)}(s')\hat{R}_{n(h')}(s', \tau_{n(k)})$$

*preparation stage:*

$$\hat{V}_{e|n(h')}^*(s', \tau_{leaf}|\tau_{n(h')}) = \sum_{k=h'}^{l} \hat{\gamma}_{n(k)}(s')V_{e|n(h')}^*(s', \tau_{n(k)}|\tau_{n(h')})$$

Next, we introduce the special cases of GHBE and GHBOE under the two particular execution modes, SME and ME. The difference in execution modes only affects the preparation stage of the equations.

For SME, we have:

$$\hat{V}_{\Pi|n(h')}^{SME}(s', \tau_{leaf}|\tau_{n(h')}) = \sum_{k=h'}^{l} \gamma_{n(k)}(s')V_{\Pi|n(h')}^{SME}(s', \tau_{n(k)}|\tau_{n(h')}),$$

and the corresponding equation sets are denoted as HBE-SMDP and HBOE-SMDP.

For ME, we have:

$$\hat{V}_{\Pi|n(h')}^{ME}(s', \tau_{leaf}|\tau_{n(h')}) = V_{\Pi|n(h')}^{ME}(s', \tau_{n(h')}|\tau_{n(h')}),$$

and the corresponding equation sets are denoted as HBE-MDP and HBOE-MDP.

As an extension of Section 4.1, given a task tree $\tau$ and a policy subtree $\Pi|n(h)$ with all node policies except the root node $\hat{\pi}_{n(h)}$ fixed, we define $\Pi^-$ as the policy tree derived from the solution of $\hat{\pi}_{n(h)}$'s HBOE-SMDP. The expected return under SME in $\tau_{n(h)}$ is denoted as $J_{\mathcal{C}}$, and under ME as $J_{\mathcal{B}}$. Similarly, given a policy subtree $\Pi|n(h)$ with all node policies except the root node $\hat{\pi}_{n(h)}$ fixed, we define $\Pi^*$ as the policy tree derived from the solution of $\tau_{n(h)}$'s HBOE-MDP. The expected return under SME in $\tau_{n(h)}$ is denoted as $J_{\mathcal{D}}$, and under ME as $J_{\mathcal{A}}$. We will prove the convergence of these solutions and the optimality of the corresponding policies.

### C.3 GENERALIZED SMDP SUBOPTIMALITY THEORY

**Theorem C.1** (**Generalized SMDP Suboptimality Theory**). *In HRL problems where the environment satisfies the Markov property, given a task tree $\tau$ and a policy tree $\pi$ with all node policies except $\hat{\pi}_{n(h)}$ fixed, the policy $\hat{\pi}_{n(h)}^-$ derived from the HBOE-SMDP of $\hat{\pi}_{n(h)}$ is the SMDP optimal node policy for $\tau_{n(h)}$, while the policy $\hat{\pi}_{n(h)}^*$ derived from the HBOE-MDP of $\hat{\pi}_{n(h)}$ is the MDP optimal node policy for $\tau_{n(h)}$. Moreover, as shown in Table 5.1, the four expected returns under SME and ME satisfy $J_{\mathcal{A}} \geq J_{\mathcal{B}}$ and $J_{\mathcal{C}} \geq J_{\mathcal{D}}$. Under Claim C.1, we also have $J_{\mathcal{B}} \geq J_{\mathcal{C}}$.*

**Claim C.1** (**Task Subtree Compatibility Condition**[10]). *A task subtree $\tau|n(h)$ rooted at task $\tau_{n(h)}$ either has only two levels, or all of its descendant task nodes $\tau_{n(h')}, h' > h$ have their child task sets $\mathcal{T}'_{\tau_{n(h')}}$ satisfy $\mathcal{T}'_{\tau_{n(h')}} \subseteq \mathcal{T}'_{\tau_{n(h)}}$.*

Based on the formal tools provided in Appendix C.1 and Appendix C.2, we now formally begin the proof of Theorem C.1.

**Lemma C.1.** *When $\forall s, n(h) \in N(h)$ satisfy $1 - \gamma_{n(h)}(s) \geq \epsilon > 0$, the GHBE for executing $\tau_{n(h)}$ using the execution subtree $e|n(h)$ has a unique fixed point. If $\max_{\tau,s} |\hat{R}_{n(h)}(s,\tau)| < M$ and $\max_{a,s} |R_{n(h)}(s,a)| < M$, then this fixed point $V^*$ satisfies $\max_{s,\tau} |V^*(s,\tau)| \leq \frac{2M}{\epsilon}$.*

**Proof:** The GHBE operator for executing $\tau_{n(h)}$ using the execution subtree $e|n(h)$ can be written as:

$$\begin{aligned} \mathcal{F}_{n(h)}^{e,\tau} V(s, \tau_{n(i)}) = & \mathbb{E}_{a \sim \pi_{n(i)}(s)}\Big[ R_{n(h)}(s,a) \\ & + \mathbb{E}_{s' \sim T(s,a)}\Big[ \gamma_{n(h)} \sum_{k=h}^{l} \hat{\gamma}_{n(k)}(s') V(s', \tau_{n(k)}) \\ & + \sum_{k=-1}^{h-1} \gamma_{n(k)}(s') \hat{R}_{n(h)}(s', \tau_{n(k)}) \Big]\Big], \quad \forall n(i) \in N|n(h) \end{aligned}$$

First, we prove that $\mathcal{F}_{n(h)}^{e,\tau}$ is a contraction mapping. Let $V : \mathcal{S} \times \mathcal{T} \to \mathbb{R}$ be a bounded function defined on the state space $\mathcal{S}$ and task space $\mathcal{T}$ (the union of all tasks in the task tree, which is complete). Denote the set of such functions by $B(\mathcal{S}, \mathcal{T})$. Define the metric $d$ on $B$ as:

$$\forall V_1, V_2 \in B(\mathcal{S}, \mathcal{T}), \quad d(V_1, V_2) = \max_{s \in \mathcal{S}, \tau \in \mathcal{T}} |V_1(s, \tau) - V_2(s, \tau)|.$$

Then we have $V_1(s, \tau) - V_2(s, \tau) \leq d(V_1, V_2)$, i.e., $V_1(s, \tau) \leq V_2(s, \tau) + d(V_1, V_2)$.

---

[10]Theorem 4.1 and existing mainstream HRL algorithms are both two-level, and thus clearly satisfy Claim C.1.

When the conditions of Theorem 5.2 hold, for all $e$ and for all $s \in \mathcal{S}, \tau_{n(i)} \in \mathcal{T}$,

$$\mathcal{F}_{n(h)}^{e,\tau} V_1(s, \tau_{n(i)}) = \mathbb{E}_{a \sim \pi_{n(i)}(s)} \Big[ R_{n(h)}(s, a)$$

$$+ \mathbb{E}_{s' \sim T(s,a)} \Big[ \gamma_{n(h)} \sum_{k=h}^{l} \hat{\gamma}_{n(k)}(s') V_1(s', \tau_{n(k)})$$

$$+ \sum_{k=-1}^{h-1} \gamma_{n(k)}(s') \hat{R}_{n(h)}(s', \tau_{n(k)}) \Big] \Big]$$

$$\leq \mathcal{F}_{n(h)}^{e,\tau} V_2(s, \tau_{n(i)}) + (1-\epsilon) d(V_1, V_2)$$

From this, we have for all $s \in \mathcal{S}, \tau_{n(i)} \in \mathcal{T}$,

$$\mathcal{F}_{n(h)}^{e,\tau} V_1(s, \tau_{n(i)}) - \mathcal{F}_{n(h)}^{e,\tau} V_2(s, \tau_{n(i)}) \leq (1-\epsilon) d(V_1, V_2).$$

Similarly, for all $s \in \mathcal{S}, \tau_{n(i)} \in \mathcal{T}$,

$$\mathcal{F}_{n(h)}^{e,\tau} V_2(s, \tau_{n(i)}) - \mathcal{F}_{n(h)}^{e,\tau} V_1(s, \tau_{n(i)}) \leq (1-\epsilon) d(V_1, V_2).$$

Hence, for all $s \in \mathcal{S}, \tau_{n(i)} \in \mathcal{T}$,

$$\left| \mathcal{F}_{n(h)}^{e,\tau} V_1(s, \tau_{n(i)}) - \mathcal{F}_{n(h)}^{e,\tau} V_2(s, \tau_{n(i)}) \right| \leq (1-\epsilon) d(V_1, V_2).$$

Therefore, for all $V_1, V_2 \in B$,

$$d(\mathcal{F}_{n(h)}^{e,\tau} V_1, \mathcal{F}_{n(h)}^{e,\tau} V_2) \leq (1-\epsilon) d(V_1, V_2).$$

Since $\epsilon > 0$, the operator $\mathcal{F}_{n(h)}^{e,\tau}$ is a contraction mapping for any given $e$. By the Banach fixed-point theorem, $\mathcal{F}_{n(h)}^{e,\tau}$ has a unique fixed point, denoted by $V^*$.

Next, we show the bound of $V^*$. For all $s, \tau_{n(i)}$,

$$\mathcal{F}_{n(h)}^{e,\tau} V^*(s, \tau_{n(i)}) = \mathbb{E}_{a \sim \pi_{n(i)}(s)} \Big[ R_{n(h)}(s, a)$$

$$+ \mathbb{E}_{s' \sim T(s,a)} \Big[ \gamma_{n(h)} \sum_{k=h}^{l} \hat{\gamma}_{n(k)}(s') V^*(s', \tau_{n(k)})$$

$$+ \sum_{k=-1}^{h-1} \gamma_{n(k)}(s') \hat{R}_{n(h)}(s', \tau_{n(k)}) \Big] \Big]$$

$$\leq M + \gamma_{n(h)} \max_{s,\tau} |V^*(s, \tau)| + M$$

$$\leq 2M + (1-\epsilon) \gamma_{n(h)} \max_{s,\tau} |V^*(s, \tau)|.$$

Thus, we have

$$\max_{s,\tau} |V^*(s, \tau)| \leq 2M + (1-\epsilon) \gamma_{n(h)} \max_{s,\tau} |V^*(s, \tau)|,$$

which implies

$$\max_{s,\tau} |V^*(s, \tau)| \leq \frac{2M}{\epsilon}.$$

Lemma C.1 is thus proved. $\qquad \square$

**Corollary C.1.** *Under the conditions of Lemma C.1, for any $e$ and any initial value $V_0$, performing successive iterations using the operator $\mathcal{F}_{n(h)}^{e,\tau}$ will result in convergence: $\lim_{n \to \infty} V_n = V$, where $V$ is the unique fixed point of $\mathcal{F}_{n(h)}^{e,\tau}$, i.e., the unified HRL value for the given execution tree $e$, $V_{e|n(h)}(\cdot, \cdot | \tau_{n(h)}) = V$.*

**Proof:** From the proof of Lemma C.1 we have shown that $\mathcal{F}_{n(h)}^{e,\tau}$ is a contraction mapping; hence the successive (Gauss) iterations converge to the unique fixed point, and this fixed point satisfies the GHBE for executing $\tau_{n(h)}$ with execution subtree $e|n(h)$. By Definition C.13, this fixed point is exactly $V_{e|n(h)}(\cdot, \cdot | \tau_{n(h)})$. $\qquad \square$

### C.3.1 STEP 1: HIERARCHICAL POLICY IMPROVEMENT THEOREM

In this section, we first establish the *Hierarchical Policy Improvement Theorem* for multi-level HRL. We then prove that when a node policy is iteratively improved until convergence, the resulting solution satisfies the GHBOE of that node policy under a given interruption tree $\mathbf{B}$. Throughout this section, the discussion is based on a fixed task $\tau_{n(h')}$ with its associated rewards and boundary conditions. Hence, for notational simplicity, we omit the conditioning part in the HRL unified value function.

**Theorem C.2 (Hierarchical Policy Improvement Theorem).** *Given two execution trees $e$ and $e'$, suppose that the only difference between them lies in the node policy functions $\hat{\pi}_{n(h)}$ and $\hat{\pi}'_{n(h)}$ at state $s$. If $\mathbb{E}_{\tau_{n(h+1)} \sim \hat{\pi}_{n(h)}(s)} V_{e|n(h')}(s, \tau_{n(h+1)}) \leq \mathbb{E}_{\tau_{n(h+1)} \sim \hat{\pi}'_{n(h)}(s)} V_{e|n(h')}(s, \tau_{n(h+1)})$, then for all $s$, it holds that $V_{e|n(h')}(s, \tau_{n(h)}) \leq V_{e'|n(h')}(s, \tau_{n(h)})$, where $h \geq h'$.*

**Proof.** We first show that

$$\mathcal{F}^{e,\tau}_{n(h')} V_{e|n(h')} \quad \leq \quad \mathcal{F}^{e',\tau}_{n(h')} V_{e|n(h')}.$$

If $e_{n(\hat{h})}, \hat{h} \geq h'$ does not belong to $e_{n(h)}$ itself nor its ancestors, then the execution subtrees $e|n(\hat{h})$ and $e'|n(\hat{h})$ are identical. Hence,

$$\mathcal{F}^{e,\tau}_{n(h')} V_{e|n(h')}(s, \tau_{n(\hat{h})}) \;=\; \mathcal{F}^{e',\tau}_{n(h')} V_{e|n(h')}(s, \tau_{n(\hat{h})}).$$

Otherwise, we have

$$\mathcal{F}^{e',\tau}_{n(h')} V_{e|n(h')}(s, \tau_{n(\hat{h})}) - \mathcal{F}^{e,\tau}_{n(h')} V_{e|n(h')}(s, \tau_{n(\hat{h})})$$
$$= \mathbb{P}[n(\hat{h}) \to n(h)]\Big( \mathbb{E}_{\tau_{n(h+1)} \sim \hat{\pi}'_{n(h)}(s)} V_{e|n(h')}(s, \tau_{n(h+1)})$$
$$- \mathbb{E}_{\tau_{n(h+1)} \sim \hat{\pi}_{n(h)}(s)} V_{e|n(h')}(s, \tau_{n(h+1)}) \Big)$$
$$\geq 0,$$

where $\mathbb{P}[n(\hat{h}) \to n(h)]$ denotes the probability of eventually selecting $\tau_{n(h)}$ when starting from $\hat{\pi}_{n(\hat{h})}$ and making decisions down the subtree.

The first equality follows directly from the decision stage equation of the GHBOE. The second inequality is guaranteed by the condition of Theorem C.2.

Therefore, we conclude that

$$\mathcal{F}^{e,\tau}_{n(h')} V_{e|n(h')} \quad \leq \quad \mathcal{F}^{e',\tau}_{n(h')} V_{e|n(h')}.$$

Since

$$V_{e|n(h')} = \mathcal{F}^{e,\tau}_{n(h')} V_{e|n(h')} \quad \leq \quad \mathcal{F}^{e',\tau}_{n(h')} V_{e|n(h')} \quad \leq \quad (\mathcal{F}^{e',\tau}_{n(h')})^k V_{e|n(h')}$$

for all $k$, by taking the limit we obtain

$$V_{e|n(h')} \quad \leq \quad \lim_{k \to \infty} (\mathcal{F}^{e',\tau}_{n(h')})^k V_{e|n(h')} = V_{e'|n(h')}.$$

This completes the proof of Theorem C.2. $\qquad\square$

**Corollary C.2 (SME Hierarchical Policy Improvement Theorem).** *Given two policy trees $\Pi$ and $\Pi'$, where the node policy functions $\hat{\pi}_{n(h)}$ and $\hat{\pi}'_{n(h)}$ differ only at state $s$, if $\mathbb{E}_{\tau_{n(h+1)} \sim \hat{\pi}_{n(h)}(s)} V^{SME}_{\Pi|n(h')}(s, \tau_{n(h+1)}) \leq \mathbb{E}_{\tau_{n(h+1)} \sim \hat{\pi}'_{n(h)}(s)} V^{SME}_{\Pi|n(h')}(s, \tau_{n(h+1)})$, then for all states $s$, it holds that $V^{SME}_{\Pi|n(h')}(s, \tau_{n(h)}) \leq V^{SME}_{\Pi'|n(h')}(s, \tau_{n(h)})$.*

**Corollary C.3 (ME Hierarchical Policy Improvement Theorem).** *Given two policy trees $\Pi$ and $\Pi'$, where the node policy functions $\pi_{n(h)}$ and $\pi'_{n(h)}$ differ only at state $s$, if $\mathbb{E}_{\tau_{n(h+1)} \sim \pi_{n(h)}(s)} V^{ME}_{\Pi|n(h')}(s, \tau_{n(h+1)}) \leq \mathbb{E}_{\tau_{n(h+1)} \sim \pi'_{n(h)}(s)} V^{ME}_{\Pi|n(h')}(s, \tau_{n(h+1)})$, then for all states $s$, it holds that $V^{ME}_{\Pi|n(h')}(s, \tau_{n(h)}) \leq V^{ME}_{\Pi'|n(h')}(s, \tau_{n(h)})$.*

Next, we prove the existence and uniqueness of the GHBOE solution for $\hat{\pi}_{n(h)}$ (**hierarchical policy iteration**).

**Existence of the GHBOE solution for $\hat{\pi}_{n(h)}$.** Given an arbitrary initial execution tree $e^0 = e$, perform iterative updates that change only the node policy $\hat{\pi}_{n(h)}$ each round. For the $k$-th iteration let the execution tree be $e^k = \{\Pi^k, \mathbf{B}\}$, and define

$$\hat{\pi}^k_{n(h)}(\tau_{n(h+1)} \mid s) = \arg \max_{\tau_{n(h+1)}} V_{e^{k-1}|n(h')}(s, \tau_{n(h+1)}),$$

where $V_{e^{k-1}|n(h')}$ is the solution of the GHBE given $\tau_{n(h')}$ and $e^{k-1}|n(h')$.

By the Hierarchical Policy Improvement Theorem ( Theorem C.2), the sequence $V_{e^k|n(h')}$ is monotonically nondecreasing in $k$. By Lemma C.1, $\{V_{e^k|n(h')}\}_k$ is uniformly bounded for all $k$. Hence the iterative process converges, and it converges to a unique limit. Denote

$$V_{e^*|n(h')} \doteq \lim_{k \to \infty} V_{e^k|n(h')}.$$

Let the execution tree derived from this limit be $e^*$, where

$$\hat{\pi}^*_{n(h)}(\tau_{n(h+1)} \mid s) = \arg \max_{\tau_{n(h+1)}} V_{e^*|n(h')}(s, \tau_{n(h+1)}),$$

and all other components of $e^*$ coincide with those of $e$. Since the iteration converges, the value evaluation of $e^*$ is $V_{e^*|n(h')}$, and further GHBOE updates on $e^*$ leave it unchanged. It is straightforward to verify that $V_{e^*|n(h')}$ satisfies the GHBOE for $\hat{\pi}_{n(h)}$. Therefore a solution to the GHBOE for $\hat{\pi}_{n(h)}$ exists. $\qquad\square$

**Uniqueness of the GHBOE solution for $\hat{\pi}_{n(h)}$.** Suppose $V^*_{e|n(h')}$ is any solution in the solution set of the GHBOE for $\hat{\pi}_{n(h)}$. Construct an execution tree $\bar{e}^*$ from $e$ and $V^*_{e|n(h')}$ by setting

$$\hat{\bar{\pi}}^*_{n(h)}(\tau_{n(h+1)} \mid s) = \arg \max_{\tau_{n(h+1)}} V^*_{e|n(h')}(s, \tau_{n(h+1)}),$$

while keeping the rest of $\bar{e}^*$ identical to $e$. If we take $e^0 = \bar{e}^*$ and run the same GHBOE iteration, then by construction the iterates satisfy $V_{e^k} \equiv V^*_{e|n(h')}$ for all $k$. But from the previous existence argument the iteration from any initial $e^0$ converges to the unique limit $V_{e^*|n(h')}$. Therefore $V^*_{e|n(h')} = V_{e^*|n(h')}$. As this holds for any solution $V^*_{e|n(h')}$, the GHBOE solution is unique. Consequently, the GHBOE for $\hat{\pi}_{n(h)}$ has a unique solution $V_{e^*|n(h')}$. $\qquad\square$

As a special case of the GHBOE solution $V_{e^*|n(h')}$, the solutions of HBOE-SMDP and HBOE-MDP also exist and are unique; we denote them by $V^{SME}_{\Pi^-|n(h')}$ and $V^{ME}_{\Pi^*|n(h')}$, respectively.

Therefore, given a policy tree $\Pi$ with all node policies except $\hat{\pi}_{n(h)}$ fixed, let $\pi^0 = \pi$. The solutions of the SMDP hierarchical policy iteration and the MDP hierarchical policy iteration satisfy

$$V^{SME}_{\Pi^-|n(h')} \geq V^{SME}_{\Pi|n(h')}, \qquad V^{ME}_{\Pi^*|n(h')} \geq V^{ME}_{\Pi|n(h')}.$$

The corresponding improved node policies are defined by

$$\begin{cases} \hat{\pi}^-_{n(h)}(\tau_{n(h+1)} \mid s) = \arg \max_{\tau_{n(h+1)}} V^{SME}_{\Pi^-|n(h')}(s, \tau_{n(h+1)}), \\ \hat{\pi}^-_{n(\hat{h})}(\tau_{n(h+1)} \mid s) = \hat{\pi}_{n(\hat{h})}(\tau_{n(h+1)} \mid s), \quad \forall n(\hat{h}) \neq n(h), \end{cases}$$

and

$$\begin{cases} \hat{\pi}^*_{n(h)}(\tau_{n(h+1)} \mid s) = \arg \max_{\tau_{n(h+1)}} V^{ME}_{\Pi^*|n(h')}(s, \tau_{n(h+1)}), \\ \hat{\pi}^*_{n(\hat{h})}(\tau_{n(h+1)} \mid s) = \hat{\pi}_{n(\hat{h})}(\tau_{n(h+1)} \mid s), \quad \forall n(\hat{h}) \neq n(h). \end{cases}$$

Hence $\hat{\pi}^-_{n(h)}$ and $\hat{\pi}^*_{n(h)}$ are the SMDP and MDP optimal node policies for $\tau_{n(h)}$, respectively.

Taking $h' = h$, we further obtain the chain of inequalities on expected returns:

$$J_{\mathcal{C}}(\tau_{n(h)}) = J^{SME}(\Pi^- \mid \tau_{n(h)}) = \mathbb{E}_{s \sim \alpha_{n(h)}}\left[V^{SME}_{\Pi^-|n(h)}(s, \tau_{n(h)} \mid \tau_{n(h)})\right]$$

$$\geq \mathbb{E}_{s \sim \alpha_{n(h)}}\left[V^{SME}_{\Pi^*|n(h)}(s, \tau_{n(h)} \mid \tau_{n(h)})\right] = J^{SME}(\Pi^* \mid \tau_{n(h)}) = J_{\mathcal{D}}(\tau_{n(h)}),$$

$$J_{\mathcal{A}}(\tau_{n(h)}) = J^{ME}(\Pi^* \mid \tau_{n(h)}) = \mathbb{E}_{s \sim \alpha_{n(h)}}\left[V^{ME}_{\Pi^*|n(h)}(s, \tau_{n(h)} \mid \tau_{n(h)})\right]$$

$$\geq \mathbb{E}_{s \sim \alpha_{n(h)}}\left[V^{ME}_{\Pi^-|n(h)}(s, \tau_{n(h)} \mid \tau_{n(h)})\right] = J^{ME}(\Pi^- \mid \tau_{n(h)}) = J_{\mathcal{B}}(\tau_{n(h)}).$$

Thus we have established the inequalities in Theorem C.1:

$$J_{\mathcal{A}} \geq J_{\mathcal{B}}, \qquad J_{\mathcal{C}} \geq J_{\mathcal{D}}.$$

$\square$

### C.3.2 STEP 2: HIERARCHICAL EXECUTION IMPROVEMENT THEOREM AND OPTIMAL EXECUTION THEOREM

This section proves the Hierarchical Execution Improvement Theorem in multi-level HRL. Based on this theorem, we further establish the Optimal Execution Theorem under the Task Subtree Compatibility Condition (Claim C.1). All discussions in this section are based on a fixed task $\tau_{n(h')}$ with its corresponding reward and boundary conditions; therefore, for notational simplicity, we omit the conditioning on $\tau_{n(h')}$ in the HRL unified value function.

**Theorem C.3** (**Hierarchical Execution Improvement Theorem**). *Given execution trees $e$ and $e'$, suppose that only the decision interruption functions $\hat{\beta}_{n(h)}$ and $\hat{\beta}'_{n(h)}$ differ at $s'$. Let $\hat{\gamma}$ and $\hat{\gamma}'$ be computed from $\hat{\beta}$ and $\hat{\beta}'$, respectively. If for all $n(l)$ it holds that $\sum_{k=h'}^{l} \hat{\gamma}_{n(k)}(s')V_{e|n(h')}(s', \tau_{n(k)}|\tau_{n(h')}) \leq \sum_{k=h'}^{l} \hat{\gamma}'_{n(k)}(s')V_{e|n(h')}(s', \tau_{n(k)}|\tau_{n(h')})$, then for all $s$ we have $V_{e|n(h)}(s, \tau_{n(h)}) \leq V_{e'|n(h)}(s, \tau_{n(h)})$, where $l \geq h \geq h'$, and $l$ denotes the level of the task directly calling an action in the previous frame.*

**Proof.** First, we show that

$$\mathcal{F}_{n(h')}^{e,\tau} V_{e|n(h')} \leq \mathcal{F}_{n(h')}^{e',\tau} V_{e|n(h')}.$$

Since for all $\tau_{leaf}, n(l)$,

$$\mathcal{F}_{n(h')}^{e',\tau} V_{e|n(h')}(s, \tau_{leaf}) - \mathcal{F}_{n(h')}^{e,\tau} V_{e|n(h')}(s, \tau_{leaf})$$

$$= T(s'|s,a)\gamma_{n(h')}(s') \left[ \sum_{k=h'}^{l} \hat{\gamma}'_{n(k)}(s')V_{e|n(h')}(s', \tau_{n(k)}) - \sum_{k=h'}^{l} \hat{\gamma}_{n(k)}(s')V_{e|n(h')}(s', \tau_{n(k)}) \right]$$

$$\geq 0,$$

by the same reasoning as in the proof of Theorem C.2, it follows that for all $\tau_{n(\hat{h})}, \hat{h} \geq h'$,

$$\mathcal{F}_{n(h')}^{e',\tau} V_{e|n(h')}(s, \tau_{n(\hat{h})}) - \mathcal{F}_{n(h')}^{e,\tau} V_{e|n(h')}(s, \tau_{n(\hat{h})}) \geq 0.$$

Hence,

$$\mathcal{F}_{n(h)}^{e,\tau} V_{e|n(h)} \leq \mathcal{F}_{n(h)}^{e',\tau} V_{e|n(h)}.$$

Similarly, applying the contraction property and taking the limit gives

$$V_{e|n(h')} \leq \lim_{k\to\infty} (\mathcal{F}_{n(h')}^{e',\tau})^k V_{e|n(h')} = V_{e'|n(h')}.$$

This completes the proof of Theorem C.3. $\square$

**Theorem C.4** (**Optimal Execution Theorem**). *Under Claim C.1, if the node policy $\hat{\pi}_{n(h)}$ is the optimal policy for the execution induced by any interruption subtree $\mathbf{B}|n(h)$, then $V_{\Pi|n(h)}^{\mathbf{B}|n(h)} \leq V_{\Pi|n(h)}^{ME}$, i.e., the ME is the optimal execution method.*

**Proof.** for all $s, \hat{h} > h$ We have

$$V_{\Pi|n(h)}^{\mathbf{B}|n(h)}(s, \tau_{n(h)}) = \max_{\tau_{n(h+1)} \in \mathcal{T}'_{\tau_{n(h)}}} V_{\Pi|n(h)}^{\mathbf{B}|n(h)}(s, \tau_{n(h+1)})$$

$$\geq \max_{\tau_{n(h+1)} \in \mathcal{T}'_{\tau_{n(\hat{h})}}} V_{\Pi|n(h)}^{\mathbf{B}|n(h)}(s, \tau_{n(h+1)})$$

$$= \max_{\tau_{n(\hat{h}+1)} \in \mathcal{T}'_{\tau_{n(\hat{h})}}} V_{\Pi|n(h)}^{\mathbf{B}|n(h)}(s, \tau_{n(\hat{h}+1)})$$

$$\geq \mathbb{E}_{\tau_{n(\hat{h}+1)} \sim \hat{\pi}_{n(\hat{h})}} V_{\Pi|n(h)}^{\mathbf{B}|n(h)}(s, \tau_{n(\hat{h}+1)})$$

$$= V_{\Pi|n(h)}^{\mathbf{B}|n(h)}(s, \tau_{n(\hat{h})}),$$

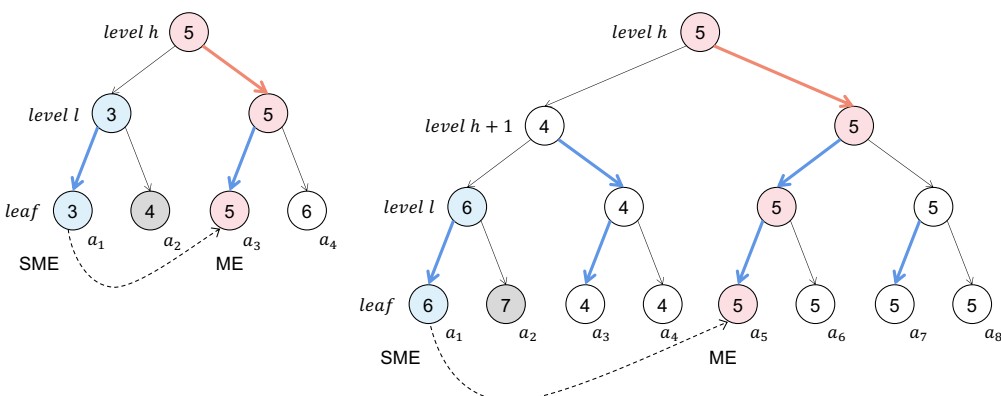

Figure 12: Counterexample: The red and blue colored arrows represent the SMDP-optimal policies for each task node, while the numbers inside the circles indicate the state-task values at the current frame when SME executes $\tau_{n(h)}$. In this example, no task termination events occur. Gray nodes denote the actions taken by the agent in the previous frame; the blue nodes form the decision path of the current frame if SME is used, and the red nodes form the decision path of the current frame if ME is used.

where the first equality follows from the optimality of $\hat{\pi}_{n(h)}$, and the first inequality is guaranteed by Claim C.1.

Thus, for all $n(l)$ we have

$$\sum_{k=h}^{l} \hat{\gamma}_{n(k)}(s') V_{\Pi|n(h)}^{\mathbf{B}|n(h)}(s', \tau_{n(k)}|\tau_{n(h)}) \leq V_{\Pi|n(h)}^{\mathbf{B}|n(h)}(s', \tau_{n(h)}|\tau_{n(h)}).$$

By Theorem C.3, it follows that for all $s$

$$V_{\Pi|n(h)}^{\mathbf{B}|n(h)}(s, \tau_{n(h)}) \leq V_{\Pi|n(h)}^{SME}(s, \tau_{n(h)}).$$

Hence, Theorem C.4 is proved. □

Furthermore, we have

$$
\begin{aligned}
J_{\mathcal{B}}(\tau_{n(h)}) &= J^{ME}(\Pi^-|\tau_{n(h)}) \\
&= \mathbb{E}_{s \sim \alpha_{n(h)}}\left[ V_{\Pi^-|n(h)}^{ME}(s, \tau_{n(h)}|\tau_{n(h)}) \right] \\
&\geq \mathbb{E}_{s \sim \alpha_{n(h)}}\left[ V_{\Pi^-|n(h)}^{SME}(s, \tau_{n(h)}|\tau_{n(h)}) \right] \\
&= J^{SME}(\Pi^-|\tau_{n(h)}) \\
&= J_{\mathcal{C}}(\tau_{n(h)}),
\end{aligned}
$$

which proves that $J_{\mathcal{B}} \geq J_{\mathcal{C}}$ in Theorem C.1. □

Hence, Theorem C.1 is fully proved. □

Claim C.1 is crucial. In multi-level HRL, if Claim C.1 is not satisfied, there exist **counterexamples** where $J_{\mathcal{B}} < J_{\mathcal{C}}$.

We can observe that in the left figure, the values of the red nodes are higher than those of the blue nodes, indicating that using ME to improve the execution method can yield a value increase. However, in the right figure, the values of the red nodes are lower than those of the blue nodes, so applying ME for execution improvement fails. This phenomenon arises because subtasks have their own independent reward accumulation mechanisms, and the decisions made by the subtask's optimal node policy can be suboptimal for the current task $\tau_{n(h)}$.

We note that the node with value 6 at level-$l$ is not selected by the optimal policy of its parent node; it is precisely such special parent tasks that conflict with $\tau_{n(h)}$ that produce this counterexample.

By adjusting the decision path, this situation can be avoided, and Claim C.1 serves as a sufficient condition.

## D  FRAMEWORK IMPROVEMENT AND ALGORITHMIC DESIGN PRINCIPLES

| Frame-work | Execution Mode (De-ployment) | Execution Mode of Be-havior Policy (Training) | Execution Mode of Target Policy (Training) | Training Require-ment | Opti-mality | Limitation |
|---|---|---|---|---|---|---|
| HRL Base-line | SME | All node policies called by SME | All node policies called by SME | On-policy | $J_{\mathcal{C}}$ | Execution and policy subop-timality exist |
| ESIF | ME | All node policies called by SME | All node policies called by SME | On-policy | $J_{\mathcal{B}}$ | Policy subop-timality exists |
| PSIF-2S | ME | Stage-1 node policies called by SME, Stage-2 node policies called by ME | Stage-1 node policies called by SME, Stage-2 node policies called by ME | On-policy | $J_{\mathcal{A}}$ | Subtask sup-port problem exists |
| PSIF-1S | ME | All node policies called by SME | For each optimi-zed node policy, its descendants called by ME, others by SME | Off-policy | $J_{\mathcal{A}}$ | Off-policy is-sue exists |

Table 6: Comparison of HRL Improvement Frameworks

The comparison of the three improved frameworks is presented in Table 6.

### D.1  IMPROVED FRAMEWORK PSEUDOCODE

#### D.1.1  ESIF PSEUDOCODE

---
**Algorithm 1** ESIF Deployment Pseudocode
---
1: **Deployment:** Deploy subtrees of $\Pi^-$ rooted at $\hat{\pi}^-_{n(h)}$ using ME
2: Initialize $s_0$, current task $\bar{\tau} = \tau_{n(h)}$, decision authority task $\tau_0 = \tau_{n(h)}$
3: **for** frame $t = 0, 1, ...$ until $\bar{\tau} = \tau_{out}$ **do**
4:     $\tau = \tau_t$
5:     **while** $\tau \notin \{\tau_{leaf}\}$ **do**
6:         $\tau = \pi_\tau(s_t)$
7:         **if** $\tau \in \{\tau_{leaf}\}$ **then**
8:             $a_t = \tau$
9:         **end if**
10:     **end while**
11:     Execute action $a_t$ and observe next state $s_{t+1}$
12:     Sample task exit $\bar{\tau} = \tau_{n(j)}$ according to $\gamma$
13:     **Return decision authority to root:** $\tau_t = \tau_{n(h)}$
14: **end for**
---

**Algorithm 2** ESIF Training Pseudocode

---

**Require:** HRL algorithm $L$, MDP environment $E$

1: **Training:** Interact with $E$ using the original SME execution mode of $L$ and learn $\Pi^-$

2: Initialize policy tree $\Pi$   $\triangleright$ Initialize each node policy $\hat{\pi}_{n(h)}, n(h) \in N$ and all state-task values
   $V(s, \tau_{n(i)}|\tau_{n(h')})$ for all $s \in \mathcal{S}, n(i) \in N$. $\tau_{n(h')}$ is the task to be optimized by $\hat{\pi}_{n(h)}$

3: **for** episode $m = 0, 1, \dots$ **do**

4:     Initialize state $s_0$, current task $\bar{\tau} = \tau_{n(0)}$, decision authority task $\tau_0 = \tau_{n(0)}$, decision path
       $L = \{\tau_{n(0)}\}$

5:     **for** frame $t = 0, 1, \dots$ until $\tau_t = \tau_{out}$ **do**                              $\triangleright$ Decision phase

6:         $\tau = \tau_t$

7:         **while** $\tau \notin \{\tau_{leaf}\}$ **do**

8:             $\tau = \hat{\pi}_\tau(s_t)$

9:             **if** $\tau \in \{\tau_{leaf}\}$ **then**

10:                 $a_t = \tau$

11:             **else**

12:                 Add $\hat{\pi}_\tau(s_t)$ to $L$

13:             **end if**

14:         **end while**

15:         Execute action $a_t$ and observe next state $s_{t+1}$                       $\triangleright$ Interaction phase

16:         **for** task $\tau_{n(i)} \in L$ **do**

17:             $r_{t+1}(\tau_{n(i)}) = R_{n(i)}(s_t, a_t)$

18:         **end for**

19:         Sample task exit $\bar{\tau} = \tau_{n(j)}$ according to $\gamma$                   $\triangleright$ Termination phase

20:         Return decision authority via SME: $\tau_{t+1} = \bar{\tau}$                    $\triangleright$ Preparation phase

21:         **for** task $\tau_{n(i)} \in L$ **do**                                                $\triangleright$ Data collection

22:             buffer$(n(i)) \leftarrow \{s_t, L, a_t, s_{t+1}, r_{t+1}|\tau_{n(i)}\}$

23:         **end for**

24:     **end for**

25: **end for**

26: **Learning:**

27: **for** node $n(h) \in N$ **do**

28:     **for** transition $\{s_t, L, a_t, s_{t+1}, r_{t+1}|\tau_{n(h')}\} \in$ buffer$(n(h'))$ **do**

29:         Resample task exit along $L$ according to current $\gamma$ to $\tau_{n(j)}$

30:         Update state-task value via HBE-SMDP:

$$V(s_t, a_t|\tau_{n(h')}) \leftarrow r_{t+1} + \mathbb{I}_{-1 \leq j < h'}\hat{R}_{n(h')}(s_t, \tau_{n(j)}) + \mathbb{I}_{h' \leq j}V(s_t, \tau_{n(j)}|\tau_{n(h')})$$

31:         **for** task $\tau_{n(i)} \in L$ **do**

32:             $V(s_t, \tau_{n(i)}|\tau_{n(h')}) \leftarrow \mathbb{E}_{\tau_{n(i+1)} \sim \hat{\pi}_{n(i)}(s_t)}[V(s_t, \tau_{n(i+1)}|\tau_{n(h')})]$

33:         **end for**

34:     **end for**

35:     **Policy update:**

36:     $\hat{\pi}_{n(h)} \leftarrow \arg\max_{\tau_{n(h+1)} \in \mathcal{T}'_{n(h)}} V(\cdot, \tau_{n(h+1)}|\tau_{n(h')})$

37: **end for**

38: **Output:** SMDP-optimal policy tree $\Pi^-$ (each node policy induced by state-task values)

---

### D.1.2   PSIF-2S PSEUDOCODE

---

**Algorithm 3** PSIF-2S Pseudocode

---

**Require:** HRL algorithm $L$, MDP environment $E$

1: **Training:** Interact with $E$ using the original SME execution mode of $L$ and learn $\Pi^-$
2: **Interaction:** Stage-1 tasks are called via SME, Stage-2 tasks are called via ME
3: **Initialize policy tree** $\Pi$ ▷ Initialize each node policy $\hat{\pi}_{n(h)}, n(h) \in N$ and all state-task values
   $V(s, \tau_{n(i)}|\tau_{n(h')})$ for all $s \in \mathcal{S}, n(i) \in N$. $\tau_{n(h')}$ is the task to be optimized by $\hat{\pi}_{n(h)}$
4: **for** episode $m = 0, 1, ...$ **do**
5:      Initialize state $s_0$, current task $\bar{\tau} = \tau_{n(0)}$, decision authority task $\tau_0 = \tau_{n(0)}$, decision path
   $L = \{\tau_{n(0)}\}$
6:      Initialize action nodes for Stage-2 training, task nodes for Stage-1 training
7:      **for** frame $t = 0, 1, ...$ until $\tau_t = \tau_{out}$ **do**            ▷ Decision phase
8:          $\tau = \tau_t$
9:          **while** $\tau \notin \{\tau_{leaf}\}$ **do**
10:             $\tau = \hat{\pi}_\tau(s_t)$
11:             **if** $\tau \in \{\tau_{leaf}\}$ **then**
12:                 $a_t = \tau$
13:             **else**
14:                 Add $\hat{\pi}_\tau(s_t)$ to $L$
15:             **end if**
16:          **end while**
17:          Execute action $a_t$ and observe next state $s_{t+1}$         ▷ Interaction phase
18:          **for** task $\tau_{n(i)} \in L$ **do**
19:             $r_{t+1}(\tau_{n(i)}) = R_{n(i)}(s_t, a_t)$
20:          **end for**
21:          Sample task exit $\bar{\tau} = \tau_{n(j)}$ according to $\gamma$         ▷ Termination phase
22:          Return decision authority: $\tau_{t+1} = \bar{\tau}$     ▷ Preparation phase: If the next task is Stage-2,
   continue transferring decision authority upwards until it reaches a Stage-1 task
23:          **while** $\tau_{t+1}$ is Stage-2 **do**
24:             $\tau_{t+1} = \tau_{t+1}.$father
25:          **end while**
26:          **for** task $\tau_{n(i)} \in L$ **do**                    ▷ Data collection
27:             buffer$(n(i)) \leftarrow \{s_t, L, a_t, s_{t+1}, r_{t+1}|\tau_{n(i)}\}$
28:          **end for**
29:      **end for**
30: **end for**
31: **Learning:**
32: **for** node $n(h) \in N$ **do**
33:      **for** transition $\{s_t, L, a_t, s_{t+1}, r_{t+1}|\tau_{n(h')}\} \in$ buffer$(n(h'))$ **do**
34:          Resample task exit along $L$ according to $\gamma$ to $\tau_{n(j)}$
35:          **while** $\tau_{n(j)}$ is Stage-2 **do**
36:             $\tau_{n(j)} = \tau_{n(j)}.$father
37:          **end while**    ▷ The target policy and the behavior policy use **the same execution mode.**
38:          Update state-task value via GHBE:

$$V(s_t, a_t|\tau_{n(h')}) \leftarrow r_{t+1} + \mathbb{I}_{-1 \leq j < h'}\hat{R}_{n(h')}(s_t, \tau_{n(j)}) + \mathbb{I}_{h' \leq j}V(s_t, \tau_{n(j)}|\tau_{n(h')})$$

39:          **for** task $\tau_{n(i)} \in L$ **do**
40:             $V(s_t, \tau_{n(i)}|\tau_{n(h')}) \leftarrow \mathbb{E}_{\tau_{n(i+1)} \sim \hat{\pi}_{n(i)}(s_t)}[V(s_t, \tau_{n(i+1)}|\tau_{n(h')})]$
41:          **end for**
42:      **end for**
43:      **if** $\hat{\pi}_{n(h)} \rightarrow \hat{\pi}^*_{n(h)}$ **then**
44:          **Mark** $\tau_{n(h)}$ **as Stage-2**
45:      **end if**
46: **end for**
47: **Policy update:** same as ESIF
48: **Output:** MDP-optimal policy tree $\Pi^*$ (each node policy induced by state-task values)
49: **Deployment:** same as ESIF

---

As task nodes gradually enter Stage-2 from the bottom up, the decision authority transfer of the behavior and target policies will gradually change, and the execution mode will progressively shift from SME to ME.

### D.1.3   PSIF-1S PSEUDOCODE

---

**Algorithm 4** PSIF-1S pseudocode

---

**Require:** HRL algorithm $L$, MDP environment $E$
1: **Training:** Interact with $E$ using the original SME execution mode of $L$ and learn $\Pi^*$
2: **Interaction:** Same as ESIF           ▷ Behavior policy execution: SME
3: **Learning:**   ▷ Value update via GHBE, and target policy execution: where $\hat{\pi}_{n(h)}$'s descendant tasks are executed via ME, others via SME
4: **for** node $n(h) \in N$ **do**
5:   **for** transition $\{s_t, L, a_t, s_{t+1}, r_{t+1} | \tau_{n(h')}\} \in \text{buffer}(n(h'))$ **do**
6:    Resample task exit along $L$ according to $\gamma$ to $\tau_{n(j)}$
7:    **if** task $\tau_{n(j)}$ is a descendant of $\tau_{n(h)}$ **then**
8:     $\tau_{n(j)} = \tau_{n(h)}$
9:    **end if**    ▷ If the task is a descendant of $\tau_{n(h)}$, transfer decision authority to $\hat{\pi}_{n(h)}$
10:    Update state-task value via GHBE:

$$V(s_t, a_t | \tau_{n(h')}) \leftarrow r_{t+1} + \mathbb{I}_{-1 \leq j < h'} \hat{R}_{n(h')}(s_t, \tau_{n(j)}) + \mathbb{I}_{h' \leq j} V(s_t, \tau_{n(j)} | \tau_{n(h')})$$

11:    **for** task $\tau_{n(i)} \in L$ **do**
12:     $V(s_t, \tau_{n(i)} | \tau_{n(h')}) \leftarrow \mathbb{E}_{\tau_{n(i+1)} \sim \hat{\pi}_{n(i)}(s_t)}[V(s_t, \tau_{n(i+1)} | \tau_{n(h')})]$
13:    **end for**
14:   **end for**
15: **end for**
16: **Policy update:** Same as ESIF
17: **Output:** MDP-optimal policy tree $\Pi^*$ (each node policy induced by state-task values)
18: **Deployment:** Same as ESIF

---

The behavior policy is executed using SME, whereas during the optimization of the target policy, tasks at higher levels called by SME execute their descendant tasks via ME. This design helps mitigate the off-policy issue, as node policies outside the subtree only influence the initial state distribution of the task, which does not affect the optimality of the target policy optimization.

### D.2   EXAMPLE OF ALGORITHMIC IMPROVEMENT

### D.2.1   OPTION FRAMEWORK ALGORITHM

Appendix C.1 presents the task tree setup under the options framework, where $\tau_{n(0)}$ is the root task, $\tau_{n(1)}$ is an option, $\tau_{leaf}$ is an action, and $R(s, a)$ is the per-step reward provided by the environment. Accordingly, the original algorithm corresponds to the HBOE-SMDP as follows:

For the root task $\hat{\pi}_{n(0)}$:

$$V(s, \tau_{n(0)} | \tau_{n(0)}) = \mathbb{E}_{\tau_{leaf} \sim \hat{\pi}_{n(0)}(s)}[V(s, \tau_{leaf} | \tau_{n(0)})],$$

$$\hat{\pi}_{n(0)}(s) = \arg\max_{\tau_{n(1)}} V(s, \tau_{n(1)} | \tau_{n(0)}),$$

$$V(s, \tau_{n(1)} | \tau_{n(0)}) = \mathbb{E}_{\tau_{leaf} \sim \hat{\pi}_{n(1)}(s)}[V(s, \tau_{leaf} | \tau_{n(0)})],$$

$$V(s, \tau_{leaf} | \tau_{n(0)}) = R(s, a) + \mathbb{E}_{s' \sim T(s,a)}\left[\hat{\gamma}_{n(1)}(s') V(s', \tau_{n(1)} | \tau_{n(0)}) + \hat{\gamma}_{n(0)}(s') V(s', \tau_{n(0)} | \tau_{n(0)})\right].$$

For the option $\hat{\pi}_{n(1)}$:

$$V(s, \tau_{n(0)} | \tau_{n(1)}) = \mathbb{E}_{\tau_{leaf} \sim \hat{\pi}_{n(0)}(s)}[V(s, \tau_{leaf} | \tau_{n(1)})],$$

$$V(s, \tau_{n(1)} | \tau_{n(1)}) = \mathbb{E}_{\tau_{leaf} \sim \hat{\pi}_{n(1)}(s)}[V(s, \tau_{leaf} | \tau_{n(1)})],$$

$$\hat{\pi}_{n(1)}(s) = \arg\max_{\tau_{n(1)}} V(s, \tau_{n(1)}|\tau_{n(1)}),$$

$$V(s, \tau_{leaf}|\tau_{n(1)}) = R(s,a) + \mathbb{E}_{s' \sim T(s,a)}\left[\hat{\gamma}_{n(1)}(s')V(s', \tau_{n(1)}|\tau_{n(1)}) + \gamma_{n(0)}(s')V(s', \tau_{n(0)}|\tau_{n(1)})\right].$$

It can be observed that during value estimation, because $\hat{\gamma} = \gamma$, the two sets of value operators between the root task and the option are identical. Therefore, a single value function can be shared between them.

In the PSIF-2S framework, once all options have entered the second stage, the node policies of all options stop updating, and the options are called by the ME. The GHBOE at this stage is as follows:

For the root task $\hat{\pi}_{n(0)}$:

$$V(s, \tau_{n(0)}|\tau_{n(0)}) = \mathbb{E}_{\tau_{leaf} \sim \hat{\pi}_{n(0)}(s)}[V(s, \tau_{leaf}|\tau_{n(0)})],$$

$$\hat{\pi}_{n(0)}(s) = \arg\max_{\tau_{n(1)}} V(s, \tau_{n(1)}|\tau_{n(0)}),$$

$$V(s, \tau_{n(1)}|\tau_{n(0)}) = \mathbb{E}_{\tau_{leaf} \sim \hat{\pi}_{n(1)}(s)}[V(s, \tau_{leaf}|\tau_{n(0)})],$$

$$V(s, \tau_{leaf}|\tau_{n(0)}) = R(s,a) + \mathbb{E}_{s' \sim T(s,a)}\left[\hat{\gamma}_{n(0)}(s')V(s', \tau_{n(0)}|\tau_{n(0)})\right].$$

Through iterative updates, $\hat{\pi}_{n(0)}$ will gradually converge from $\hat{\pi}_{n(0)}^-$ to $\hat{\pi}_{n(0)}^*$.

In the PSIF-1S framework, unlike the original options framework, the execution modes of the target policies estimated by the root task and the options differ. Therefore, two separate value functions are maintained. The corresponding GHBOE is as follows:

For the root task $\hat{\pi}_{n(0)}$:

$$V(s, \tau_{n(0)}|\tau_{n(0)}) = \mathbb{E}_{\tau_{leaf} \sim \hat{\pi}_{n(0)}(s)}[V(s, \tau_{leaf}|\tau_{n(0)})],$$

$$\hat{\pi}_{n(0)}(s) = \arg\max_{\tau_{n(1)}} V(s, \tau_{n(1)}|\tau_{n(0)}),$$

$$V(s, \tau_{n(1)}|\tau_{n(0)}) = \mathbb{E}_{\tau_{leaf} \sim \hat{\pi}_{n(1)}(s)}[V(s, \tau_{leaf}|\tau_{n(0)})],$$

$$V(s, \tau_{leaf}|\tau_{n(0)}) = R(s,a) + \mathbb{E}_{s' \sim T(s,a)}\left[\hat{\gamma}_{n(0)}(s')V(s', \tau_{n(0)}|\tau_{n(0)})\right].$$

For the option $\hat{\pi}_{n(1)}$:

$$V(s, \tau_{n(0)}|\tau_{n(1)}) = \mathbb{E}_{\tau_{leaf} \sim \hat{\pi}_{n(0)}(s)}[V(s, \tau_{leaf}|\tau_{n(1)})],$$

$$V(s, \tau_{n(1)}|\tau_{n(1)}) = \mathbb{E}_{\tau_{leaf} \sim \hat{\pi}_{n(1)}(s)}[V(s, \tau_{leaf}|\tau_{n(1)})],$$

$$\hat{\pi}_{n(1)}(s) = \arg\max_{\tau_{n(1)}} V(s, \tau_{n(1)}|\tau_{n(1)}),$$

$$V(s, \tau_{leaf}|\tau_{n(1)}) = R(s,a) + \mathbb{E}_{s' \sim T(s,a)}\left[\hat{\gamma}_{n(1)}(s')V(s', \tau_{n(1)}|\tau_{n(1)}) + \gamma_{n(0)}(s')V(s', \tau_{n(0)}|\tau_{n(1)})\right].$$

This corresponds to the OC-PSIF-1S improved algorithm in the main text. $V(\cdot, \cdot|\tau_{n(0)})$ corresponds to $V_{hi}$, while $V(\cdot, \cdot|\tau_{n(1)})$ corresponds to $V_{lo}$.

During the lower-level policy value estimation, neither the greedy root task policy induced by $V(s, \tau_{n(1)}|\tau_{n(1)})$ is used to select actions, nor is $V(s, \tau_{n(1)}|\tau_{n(0)})$ used for value estimation. Instead, the greedy root task policy $\hat{\pi}_{n(0)}$ is induced by $V(s, \tau_{n(1)}|\tau_{n(0)})$, and value estimation uses $V(s, \tau_{n(1)}|\tau_{n(1)})$. The former ensures that the actions needed for the update bootstrap term match the actions actually taken during sampling, eliminating off-policy bias. The latter ensures that value estimation aligns with the execution mode to be computed (options are called by SME rather than by ME as in $V(s, \tau_{n(1)}|\tau_{n(0)})$), guaranteeing that the value obtained under the behavior policy is preserved.

$V(s, \tau_{n(1)}|\tau_{n(0)})$ is used to induce the greedy node policy for the root task, while $V(s, \tau_{n(1)}|\tau_{n(1)})$ is used for the termination gradient of the option and can also serve as a baseline for the option node policy gradient update:

$$\Delta\vartheta \sim -\alpha_\vartheta \frac{\partial \beta_\omega(s)}{\partial \vartheta}\left(V(s, \tau_{n(1)}|\tau_{n(1)}) - V(s, \tau_{n(0)}|\tau_{n(1)})\right),$$

$$\Delta\theta \sim \alpha_\theta \frac{\partial \pi_\omega(a|s)}{\partial \theta}\left(V(s, a|\tau_{n(1)}) - V(s, \tau_{n(1)}|\tau_{n(1)})\right).$$

### D.2.2 GOAL-BASED FRAMEWORK ALGORITHM

Appendix C.1 presents the task tree setup for the target framework, where $\tau_{n(0)}$ belongs to the target set $G$, $\tau_{n(1)}$ belongs to the subgoal set $g$, and $\tau_{leaf}$ represents actions. The original algorithm corresponds to the HBOE-SMDP as follows:

For the root task $\hat{\pi}_{n(0)}$:

$$V(s, \tau_{n(0)}|\tau_{n(0)}) = \mathbb{E}_{\tau_{leaf} \sim \hat{\pi}_{n(0)}(s)}[V(s, \tau_{leaf}|\tau_{n(0)})],$$

$$\hat{\pi}_{n(0)}(s) = \arg\max_{\tau_{n(1)}} V(s, \tau_{n(1)}|\tau_{n(0)}),$$

$$V(s, \tau_{n(1)}|\tau_{n(0)}) = \mathbb{E}_{\tau_{leaf} \sim \hat{\pi}_{n(1)}(s)}[V(s, \tau_{leaf}|\tau_{n(0)})],$$

$$V(s, \tau_{leaf}|\tau_{n(0)}) = R_{n(0)}(s,a) + \mathbb{E}_{s' \sim T(s,a)}\left[\hat{\gamma}_{n(1)}(s')V(s', \tau_{n(1)}|\tau_{n(0)}) + \hat{\gamma}_{n(0)}(s')V(s', \tau_{n(0)}|\tau_{n(0)})\right].$$

For the option $\hat{\pi}_{n(1)}$:

$$V(s, \tau_{n(0)}|\tau_{n(1)}) = \mathbb{E}_{\tau_{leaf} \sim \hat{\pi}_{n(0)}(s)}[V(s, \tau_{leaf}|\tau_{n(1)})],\,^{[11]}$$

$$V(s, \tau_{n(1)}|\tau_{n(1)}) = \mathbb{E}_{\tau_{leaf} \sim \hat{\pi}_{n(1)}(s)}[V(s, \tau_{leaf}|\tau_{n(1)})],$$

$$\hat{\pi}_{n(1)}(s) = \arg\max_{\tau_{n(1)}} V(s, \tau_{n(1)}|\tau_{n(1)}),$$

$$V(s, \tau_{leaf}|\tau_{n(1)}) = R_{n(1)}(s,a) + \mathbb{E}_{s' \sim T(s,a)}\left[\hat{\gamma}_{n(1)}(s')V(s', \tau_{n(1)}|\tau_{n(1)})\right].$$

In some target-framework algorithms, the execution of a subgoal lasts for a fixed number of frames $c$, while the upper-level value estimation only accounts for the value of the subgoal when it has been fully executed. It does not estimate the value of the remaining subtask transitions during execution. Therefore, the ESIF framework cannot be directly applied. To address this, the subgoal $g$ can be extended to a subgoal–elapsed-time pair $(g, \hat{t})$, with $0 \leq \hat{t} \leq c$.

## E EXPERIMENT DETAILS AND ANALYSIS

### E.1 ENVIRONMENT SETTINGS

**Randomized version of AntReacher.** A continuous task simulated in MuJoCo aims at controlling an ant robot to achieve a random goal on the map. When deploying a policy, an additional random transition event is engaged: For every 10 timestep, the ant will have a random state transition on the whole map with the probability of $10\%$, only involving coordinates $x$ and $y$.

**Fourrooms.** A GridWorld environment of four rooms. The cells of the grid correspond to the states of the environment. The agent must start from a random starting cell and reach a fixed-end cell. In any state, the agent is feasible to perform one of four actions, $up, down, left, right$. And there is a random transition after the action is selected. With the probability of $\frac{2}{3}$, the agent moves one cell in the chosen direction. With the probability of $\frac{1}{3}$, the agent moves one cell in one of the other three directions, each with the chance of $\frac{1}{9}$. In either case, if the movement takes the agent onto a wall, the agent remains in the same cell.

### E.2 ALGORITHMS USED

**HAC.** This is a state-of-the-art goal-based HRL algorithm. For the goal-based HRL algorithms, the higher level of the hierarchy interacts with an SMDP induced by the environment dynamics and the behavior of the lower level. During training, the lower-level policy constantly changes with updates, making the higher-level interaction non-stationary. HAC proposed the hindsight action relabeling method to alleviate this problem for static environments by 'pretending' that the lower-level policy has already been optimal. In addition, HAC also proposed hindsight goal transitions and subgoal testing to accelerate the learning of sparse-reward tasks.

---

[11] this computation is irrelevant for policy learning and can be omitted in practice.

**OC.** This is a classic option-based HRL algorithm. This algorithm learns the entire hierarchical policy tree in an end-to-end manner. The higher-level policy and all the options (i.e., intra-option policies and termination functions) are trained using policy gradient derived using the root task rewards (i.e., the original environmental reward). A limitation of OC is that it requires predefining the number of options. Moreover, the policy gradients are heavily dependent on the root task reward. Thus, OC is likely to perform poorly on the sparse-reward tasks.

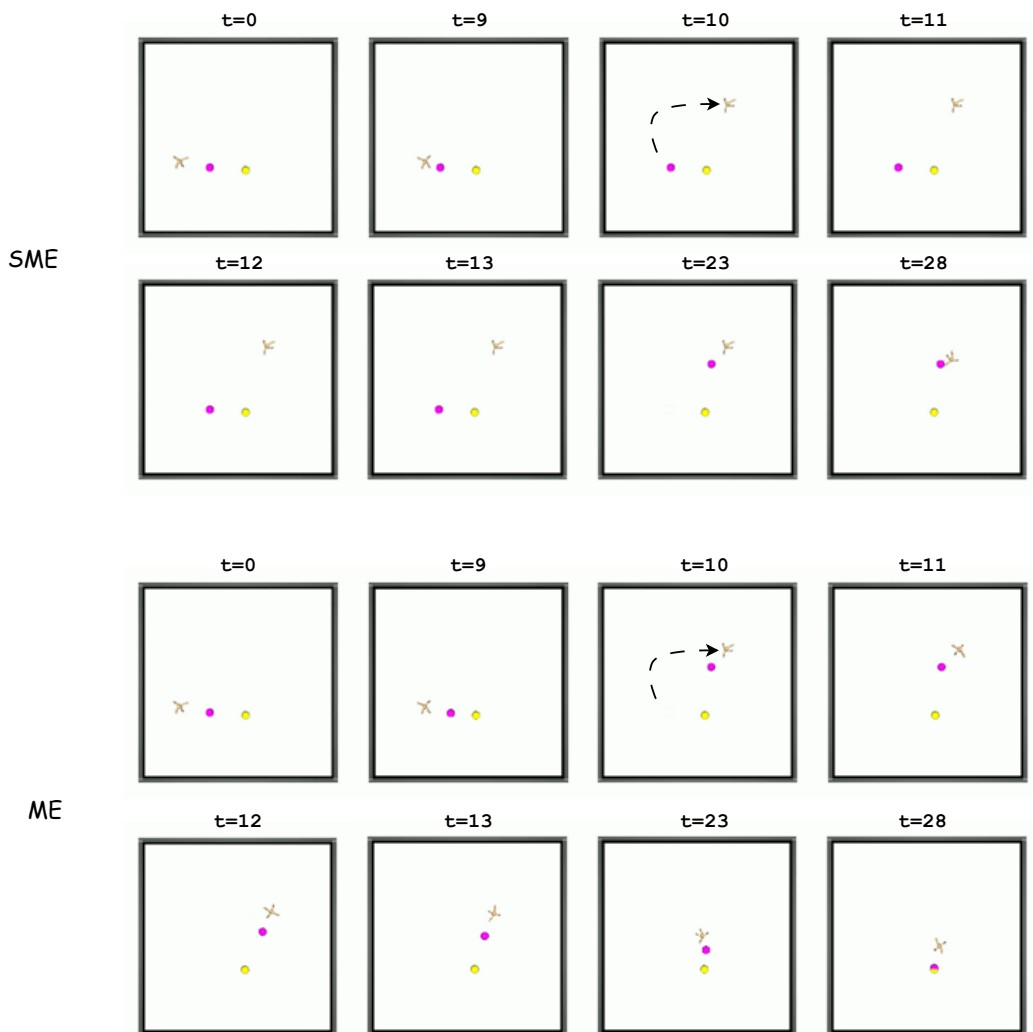

Figure 13: The ant robot navigates from a random starting position to a random ending position in the AntReacher environment. The yellow sphere on the map refers to the ending position, the red sphere refers to the subgoal proposed by the agent's higher-level policy, and the lower-level policy directly controls the ant's joints for moving towards this subgoal.

To explore the performance difference of the same policy under SME and ME in the environment without random transition events, we deploy the HAC algorithm in the original environments of its paper: UR5, AntReacher, and AntFourrooms. And the results are shown in Fig. 14. We observe that the performance of ME is as least as good as SME's. The opposite conclusion may be obtained in other algorithms. Not all algorithms can improve performance after switching the execution mode from SME to ME at no cost due to their possible defects, especially when the higher-level policy is suboptimal and has not yet converged to the optimum. At this point, the conditions required by the Optimal Execution Theorem are not satisfied. Instead of directly adopting ME based on the theorem, one should design a new algorithm according to the Execution Improvement Theorem to learn the optimal interruption function for the deployment policy. Our framework provides a

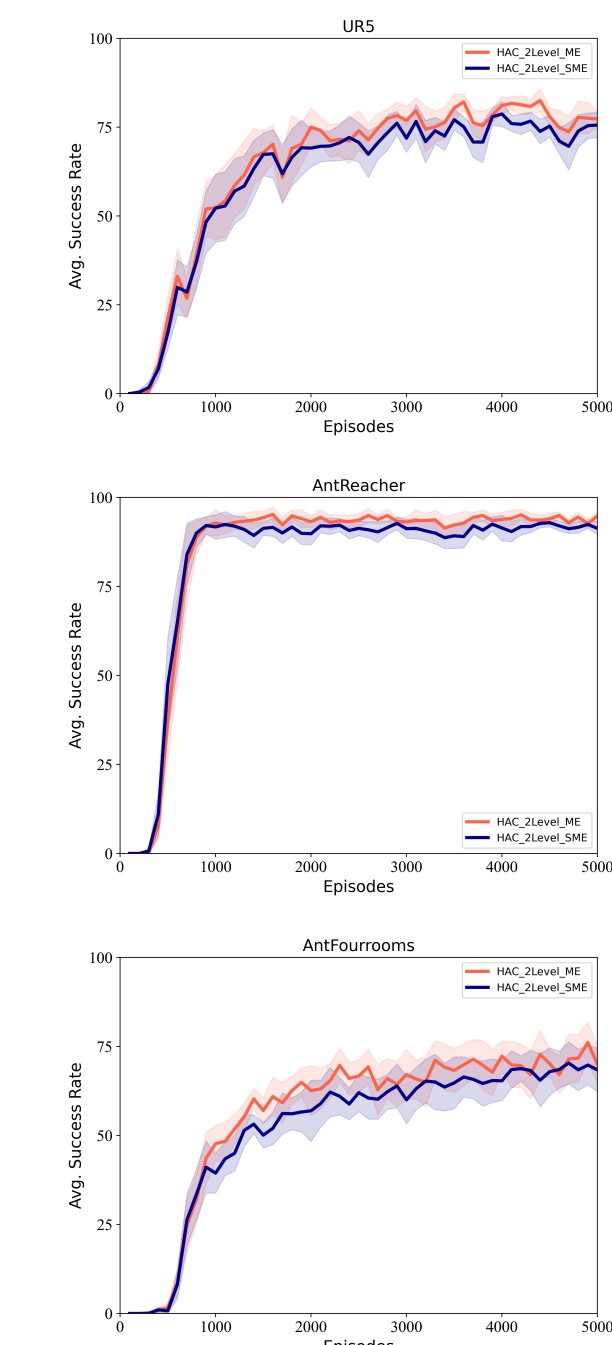

Figure 14: Comparisons for HAC under SME and ME.

principled foundation for further potential improvements: the execution modes of the target policy, behavior policy, and deployment policy are mutually independent.

### E.3 HYPERPARAMETERS

For HAC, we follow the open-source implementation of the author, which is available at
https://github.com/andrew-j-levy/Hierarchical-Actor-Critc-HAC-. All
the experimental results are averaged using 10 random seeds.

For OC, we provide the code in the supplementary material, and all the experimental results are averaged using 350 random seeds. The key training hyperparameters are as follows:

- the discount factor $\gamma = 0.99$.
- the learning rate of termination function $\alpha_{term} = 0.25$.
- the learning rate of intra-option policy $\alpha_{intra} = 0.25$.
- the learning rate of critic $\alpha_{critic} = 0.5$.
- the learning rate of higher-level policy $\alpha_q = 0.2$.
- the number of options $\omega = \{4, 8\}$.

### E.4 EXTRA RESULTS AND LIMITATIONS

We visualize the results of deploying the HAC algorithm in the AntReacher environment under SME and ME in Fig. 13 and found evident disadvantages of SME. When a random transition event occurs at $t = 10$, the ant shows *stagnant* until $t = 23$ under SME. This result may be because the large-scale coordinate offset prevents the lower-level policy from making effective actions according to the current state and subgoal. However, under ME, the higher-level policy adjusts the subgoal in time so that the task of the lower-level policy is still moving towards the attainable subgoal.

## F USE OF LLMs

We employed the Large Language Model to polish writing.

