# OpenReview forum: "On the Suboptimality of Semi-Markov Decision Process in Hierarchical Reinforcement Learning"
_ICLR.cc/2026/Conference — Submitted to ICLR 2026_

### Official Review · Reviewer_vbRc · 2025-10-26

**Soundness:** 3
**Presentation:** 2
**Contribution:** 3
**Rating:** 6
**Confidence:** 3

**Summary:**

In this paper, the authors aimed at: first, identify a "fundamental design flaw" of hierarchical reinforcement learning(HRL) through theoretical analysis; second, propose new framework for improvement; and finally, conduct numerical experiments to demonstrate the effectiveness of the proposed framework.

**Strengths:**

I feel that the basic approach in this paper is sound. The authors demonstrated the order of expected returns of different execution and policy combinations through a thorough analysis. These theoretical results also allow them to propose new improvement frameworks to achieve effective outcomes for HRL. The numerical results, while limited, demonstrate that the proposed frameworks indeed improved the performance as expected.

**Weaknesses:**

The implications of the theoretical results on the Markov decision processes and SemiMarkov decision processes and fundamental design flaws of HRL is not convincing demonstrated.
     a. First, the HRL is not formally defined and described, while RL and MDP are very well-known subjects, HRL and SMDP are not, the authors should produce formal definitions of key concepts, such as subtask.
     c. Related, the difference between MDP and SMDP as presented is not convincing, the authors need to use precise  language to describe their differences, in addition to illustration by figures;
     b. While the fundamental flaw of HRL regarding "SMDP is simultaneously adopted in both the target and behavior policies" is prominently stated in the abstract, it is not adequately and explicitly discussed in the main paper.

**Questions:**

1. A detailed discussion on the similarities and differences of the MDP and SMDP concepts used in this paper is needed;
2. An explicit discussion on the implication of Theorem 4.1 on design flaw of HRL would be helpful.

---

> ### Author Response · Authors · 2025-11-22
> **Response to Reviewer vbRc (1/3)**
>
> We sincerely thank Reviewer vbRc for the careful reading and positive evaluation of our paper. We are very glad that you find our overall methodology reasonable. Our goal is indeed to develop theoretical analyses that can *reveal fundamental issues and guide algorithm design*, rather than retrofitting theory to justify existing algorithms. Below, we respond to your main concerns point by point.
>
> ---
>
> ### 1. On the formal definition and description of HRL and subtasks
>
> > “HRL lacks formal definition and description. Reinforcement learning (RL) and Markov Decision Processes (MDPs) are well-established research areas, whereas HRL and Semi-Markov Decision Processes (SMDPs) are not. The authors should formally define key concepts such as subtask.”
>
> In the current version, we do provide several relevant definitions in Section 2, but they are scattered across different paragraphs and are not prominently highlighted. This makes it difficult for readers to recognize them as a coherent set of *formal definitions*.
>
> More specifically:
>
> - In **Section 2.1**, we provide a formal description of a *task*.
> - In **the first paragraph of Section 2.2**, we build on the notion of task and give a formal definition of a *subtask*, thereby extending from RL to HRL: "We refer to this stochastic process with variable-duration transitions as a Semi-Markov Decision Process (SMDP), and the tasks that can be called under $\tau$ are called the subtasks of $\tau$.".
> - In **the second paragraph of Section 2.2**, we give an SMDP-based formal HRL model: "... define the SMDP of task $\tau$ as $(\mathcal{S},\mathcal{T}' _{\tau},\bar T _{\tau},\bar R _{\tau},\alpha _{\tau},\beta _{\tau})$. Here, $\bar T _{\tau}: \mathcal{S} \times \mathcal{T}' _{\tau} \to \Delta \mathcal{S}$ is the subtask transition function, and $\bar R _{\tau}: \mathcal{S} \times \mathcal{T}' _{\tau} \to \mathbb{R}$ is the subtask reward function.","... comprises a state space $\mathcal S$, a action space $\mathcal A$, a transition kernel $T: \mathcal S\times\mathcal A\to\Delta(\mathcal S)$ (where $\Delta(\mathcal S)$ denotes probability distributions over $\mathcal S$), a reward function $R _\tau:\mathcal S\times\mathcal A\to\mathbb R$, a task  initial-state distribution $\alpha _\tau\in\Delta(\mathcal S)$, and a termination function $\beta _\tau:\mathcal S\to[0,1]$ which means the probability of exiting task $\tau$ upon visiting a state $s$.".
>
> In the revision, we will **boldface these key concepts** (task, subtask, etc.) and make them more explicit, so that readers can form a clearer understanding of the HRL model used in our paper.
>
> ---

---

> > ### Author Response · Authors · 2025-11-22
> > **Response to Reviewer vbRc (2/3)**
> >
> > ### 2. On the differences between MDP and SMDP, and the relation to ME and SME
> >
> > > “The differences between MDP and SMDP are not convincingly explained. The authors need to use more precise language to describe their differences, supported by figures. A detailed discussion of the similarities and differences between the MDP and SMDP concepts used in our paper is needed.”
> >
> > We fully understand this concern: if the difference between MDP and SMDP is not clearly explained, it becomes difficult for readers to fully grasp our later theoretical analysis on HRL execution modes (ME vs. SME).
> >
> > A one-sentence summary of **MDP vs. SMDP** in our setting is:
> >
> > - In an **MDP**, the agent (standard RL) calls *primitive actions* that last exactly one time step.
> > - In an **SMDP**, the agent (HRL at a higher level) calls *subtasks/options* that last for a random number of time steps.
> >
> > A one-sentence summary of **ME vs. SME** in our HRL setting is:
> >
> > - Under **ME (Markov Execution)**, subtasks are *executed only for a single time step* before the higher-level policy can re-decide.
> > - Under **SME (Semi-Markov Execution)**, subtasks are *executed for a random duration* (until completion), and the higher-level policy only decides again when the subtask terminates.
> >
> > In **Section 3.1**, we formally define the difference between ME and SME, which are precisely the two execution modes that our paper focuses on. Their differences, both in definition and in concrete interaction patterns, are described through **Definition 3.1**, **Definition 3.2**, and the examples in **Figure 2** and **Figure 3**.
> >
> > In addition, **Section 2.2** introduces the precise formal definitions of MDP and SMDP used in this work. However, we agree that the contrast between MDP and SMDP is not currently emphasized as clearly as it could be. The main focus of the paper is on the execution modes ME and SME *given a hierarchical decomposition*, rather than on the general MDP/SMDP theory itself. That said, in the revision, we are happy to **add a short explanatory paragraph (and, if space permits, a small illustrative figure)** to more intuitively highlight the difference between MDP and SMDP for readers.
> >
> > ---

---

> ### Author Response · Authors · 2025-11-22
> **Response to Reviewer vbRc (3/3)**
>
> ### 3. On the connection between Theorem 4.1 and the “design flaw” of HRL
>
> > “Although the abstract emphasizes a fundamental flaw of HRL—‘both target and behavior policies adopt SMDP’—this is not sufficiently and explicitly discussed in the main text. A clear discussion of the impact of Theorem 4.1 on HRL design flaws would be helpful.”
>
> This is a very important comment, and it made us realize that while the current version does present **Theorem 4.1**, it does not “explicitly spell out” the causal link between this theorem and the HRL design flaw we describe in the abstract.
>
> Our intended logical structure in the paper is as follows:
>
> 1. In **Section 3.2**, we use a concrete example to demonstrate the existence of **policy suboptimality**, with the goal of giving readers an intuitive sense that:
>    - For the same set of subtasks, the *optimal policy itself* can differ under different execution modes (SME vs. ME).
>    - Therefore, if we train an “optimal” policy under SME and then deploy it under ME, there can still be a gap where the policy is not truly optimal under ME.
>
> 2. In **Section 4.1**, we formalize this phenomenon into a theoretical result and provide a proof sketch. In brief:
>    - Let $J_\mathcal{A}$ denote “the optimal value under ME” and $J_\mathcal{B}$ denote “the optimal value under SME”. We prove that $J_{\mathcal{B}} \le J_{\mathcal{A}}.$
>    - This means that, given the same set of subtasks and termination structure, if the *target policy* is optimized under SME semantics, the best return it can achieve is at most $J_{\mathcal{B}}$, whereas in principle ME can achieve the higher value $J_{\mathcal{A}}$.
>
> 3. In **Section 4.2**, we further analyze how the **execution mode of the behavior policy** affects the training process:
>    - If the behavior policy also adopts ME, many HRL algorithms (especially options/OC-style methods) can suffer from serious off-policy issues and learning difficulties for lower-level tasks.
>    - In contrast, using SME for the behavior policy helps maintain a more stable data distribution within subtasks, which is beneficial for training lower-level subtasks.
>
> Based on the above analysis, in **Section 4.3** we draw the following design conclusions:
>
> - To avoid **policy suboptimality**, the **target policy should use ME** (i.e., we should optimize the policy under ME semantics so that it approaches $J_{\mathcal{A}}$ instead of being limited by $J_{\mathcal{B}}$).
> - To ensure stable sampling and learnability of lower-level subtasks during training, the **behavior policy should still use SME**.
> - Overall, the common HRL design in which **“both target and behavior policies use SME”** is exactly the **“fundamental design flaw”** we refer to in the abstract.
>
> We fully agree with your suggestion that this logical chain should be written *explicitly* in the main text, rather than leaving readers to reconstruct it solely from theorems and examples. We believe such an explicit explanation will greatly help readers understand the direct impact of Theorem 4.1 on HRL design.
>
> ---
>
> Once again, we thank you for your positive assessment of our approach and contributions, and for pointing out the weaknesses in our formalization and in the explanation of the theorem’s implications. We will incorporate the above clarifications in the revised version to improve the clarity and readability of the paper.

---

### Official Review · Reviewer_q9eR · 2025-10-28

**Soundness:** 2
**Presentation:** 2
**Contribution:** 2
**Rating:** 2
**Confidence:** 3

**Summary:**

The authors explore the suboptimality of semi-MDPs (SMDPs) and propose a novel framework based on optimal execution modes that addresses this suboptimality.

**Strengths:**

The problem setting is well-motivated. Exploring the suboptimality of SMDPs seems like a worthwhile topic to explore. The analysis done on exploring the suboptimality was done reasonably well.

**Weaknesses:**

My biggest concern with this paper is that it lacks a critical and necessary discussion point. In particular, the authors arrive at the conclusion that the so-called Markov Execution (ME) mode, in which a subtask is only active for a single frame, should be utilized during deployment. Yet, there is no discussion as to how this differs from learning single-frame actions via a regular (non-semi) MDP. Perhaps even more importantly, there is no discussion that addresses the motivation for ME. More specifically, if the subtasks are only being called for a single frame during execution, then what is the point of learning them at all, when one could just learn one-frame action policies via a regular MDP, at what I would imagine is a lower computational cost?

In my view, the authors’ fixation on execution modes works against them, and I would question whether it is even necessary to include it in the paper. The premise of exploring the suboptimality of SMDPs is, in itself, quite interesting, and the work that the authors performed in this regard was done reasonably well. But then the discussion on execution modes obscures the narrative of the paper, introduces an excessive number of acronyms, and ultimately makes the paper hard to follow.

One potential idea that the authors could explore if they insist on exploring this notion of execution modes, is that rather than having the binary ME/SME modes, it would be much more insightful to perform some sort of analysis that looks at how far away from optimality the agent is based on how long the subtask is executed. For example, perhaps in the first few frames of execution the gap is insignificant, but then after some threshold amount of frames, the gap begins to significantly increase. Accordingly, the authors could propose a way to find this ‘optimal stopping time’ for the subtasks, such that the optimality gap is negligible, but the agent still benefits from the benefits of temporal abstraction.

Overall, while the work done on analyzing the suboptimality of SMDPs is done reasonably well, the notion of execution modes, in my view, needs to be heavily revised.

Moreover, there are several aspects related to the presentation of the paper that need to be significantly improved. Aside from Figure 1, all the figures are hard to follow and could benefit from annotations and perhaps partitioning into a), b), etc. subfigures. For example, it is not clear in Figure 2 when the legend ends, when SME/ME begins, etc. In Figure 3, there is no legend, and ultimately, the figure as a whole is confusing and does not communicate to the reader what is happening in a sufficiently-well manner. In Figure 5a) right, one of the baselines is cropped out of the plot. From a writing perspective, similar issues occur. For example, the proof sketch of theorem 1 introduces too many acronyms and concepts without explaining them, thereby making it hard to understand.

**Questions:**

N/A

---

> ### Author Response · Authors · 2025-11-22
> **Response to Reviewer q9eR (1/5)**
>
> We sincerely thank the reviewer for the positive assessment that our analysis of SMDP suboptimality is “done reasonably well” – this has been exactly the core goal we have been working toward.
>
> Below we address your main concerns regarding (1) the role and motivation of Markov Execution (ME), and (2) the narrative burden introduced by execution modes and related notation.
>
> ---
>
> ### 1. Questions about ME and its relation to flat MDPs
>
> > “The authors conclude that the so-called Markov Execution (ME) mode, in which a subtask is only active for a single frame, should be utilized during deployment. However, they do not discuss how this differs from learning single-frame actions via a regular (non-semi) MDP. … After all, learning one-frame action policies via a regular MDP should be cheaper.”
>
> #### 1.1 ME at deployment: executing a learned hierarchical policy, not relearning a flat one
>
> As described in Section 2.3, our focus at **deployment** is how to execute an **already learned** hierarchical policy.
> In this phase, ME is *not* “learning a flat policy from scratch.” Instead, it is:
>
> **Executing a trained hierarchical policy in a Markov way, in a manner that resembles a flat MDP in terms of decision frequency, while keeping the hierarchical structure and abstractions.**
>
> At deployment time, we do not modify or train the policy; we only care about the **maximum return that a fixed policy can achieve under different execution semantics**. Our notion of **execution suboptimality** formalizes exactly this: for a given policy, different execution modes (e.g., SME vs ME) can yield different returns, and in our framework ME is necessary to fully realize the policy’s potential return at deployment.
>
> #### 1.2 ME during training vs flat RL: why hierarchical ME is still different
>
> Your question also touches on the **training phase**: in what sense is training hierarchical policies under ME different from training a single-layer flat RL policy that directly outputs primitive actions?
>
> During training, we set the **target policy** to use ME as its execution mode. Superficially this resembles standard RL, since both are searching for an optimal policy under Markov execution. However, in a two-level HRL setting there are several important theoretical and practical differences:
>
> 1. **Action abstraction and reduced branching factor**
>    The high-level policy chooses from a relatively small set of subtasks (options/goals), rather than directly producing high-dimensional primitive actions. This is particularly important in continuous control or large action spaces.
>
> 2. **Representation and credit sharing**
>    Subtasks (options/goals) are reusable modular representations. They can be reused across many states and even across different tasks. A flat policy often has to “re-learn” similar behaviors in many contexts, whereas ME-based HRL can reuse the same subtask policy.
>
> 3. **Reduced planning complexity per level**
>    When the overall mapping is decomposed into multiple stages via well-chosen subtasks, the planning complexity at each stage is lower than that of the original task (cf. Wen et al., 2020). This is one of the classical motivations for HRL.
>
> Empirically, in the ablation study of Figure 5(a), we explicitly include **flat baselines**: HAC\_1level (DDPG) and OC\_1OP (AC). In our sparse-reward, long-horizon tasks, these flat baselines respectively obtain zero success rate or always reach the maximum episode length without success.
>
> This suggests that, at least in our settings, **relying purely on flat RL to discover the ME-optimal policy is not effective in practice**, especially under sparse rewards and long-term planning requirements. Our point is that, in the tasks we study, hierarchical structure plus ME execution is crucial for making progress, while flat RL fails.
>
> Finally, an important conceptual clarification:
>
> Our theoretical analysis is **not** trying to show that “HRL+ME always dominates flat RL.” Instead, we focus on a more specific question:
>
> **Given that we have already adopted a hierarchical architecture, what extra suboptimality is introduced purely by the SMDP/SME execution semantics, and how can we remove it by using a more suitable execution mode such as ME?**
>
> We will emphasize this scope more clearly in the revised paper.
>
> ---

---

> > ### Author Response · Authors · 2025-11-22
> > **Response to Reviewer q9eR (2/5)**
> >
> > ### 2. Motivation for ME and the role of execution modes
> >
> > > “More importantly, there is no discussion that addresses the motivation for ME. If the subtasks are only being called for a single frame during execution, then what is the point of learning them at all?”
> >
> > #### 2.1 Motivation of ME in our framework
> >
> > In our framework, the motivation for ME can be summarized as follows:
> >
> > 1. **At deployment:**
> >    For a fixed hierarchical policy, using ME as the execution mode **maximizes the achievable return** of that policy, and removes execution suboptimality.
> >
> > 2. **During training (target policy):**
> >    If the target policy is optimized under SME, it may converge to the *SME-optimal* policy, which is not necessarily optimal under ME. Using ME for the target policy allows it to learn the *ME-optimal* policy and avoids policy suboptimality.
> >
> > In addition, as discussed in Section 4.3, during **data collection** we actually use SME for the **behavior policy**:
> >
> > - The behavior policy calls subtasks in the standard SME manner and executes them until (stochastic) termination, which allows the agent to reach “meaningful” states that are difficult to visit with purely single-step behavior.
> > - This combines the exploration benefits of temporally extended execution (SME) with the optimization advantages of ME as the target and deployment execution mode.
> >
> > We are happy to expand this discussion in the revised version, so that the distinct roles of ME and SME (for target policy, behavior policy, and deployment) are clearer to the reader.
> >
> > ---

---

> > > ### Author Response · Authors · 2025-11-22
> > > **Response to Reviewer q9eR (3/5)**
> > >
> > > ### 3. Execution modes, notation burden, and the “optimal stopping time” idea
> > >
> > > > “In my view, the authors’ fixation on execution modes works against them… The discussion on execution modes obscures the narrative, introduces an excessive number of acronyms, and ultimately makes the paper hard to follow.”
> > >
> > > > “If the authors insist on exploring this notion of execution modes, … it would be much more insightful to analyze how far away from optimality the agent is based on how long the subtask is executed… and find an ‘optimal stopping time’ for the subtasks…”
> > >
> > > #### 3.1 Why execution modes are necessary for SMDP suboptimality
> > >
> > > We fully understand this concern: in the current version, the notation and acronyms related to execution modes are indeed heavy and may obscure the main message. This is a weakness in our presentation, and we will address it.
> > >
> > > On the other hand, our analysis indicates that **both the root cause of SMDP suboptimality and the natural remedies for it ultimately lie in the execution semantics**:
> > >
> > > - SMDP models are defined not only by *what* subtasks are available, but also by **how long they persist and when control returns to the parent**.
> > > - In other words, to talk about SMDP suboptimality, we must formalize the execution mode; otherwise we cannot even state what “suboptimality of SMDPs” means.
> > >
> > > The primary goal of our paper is to clarify “what SMDP suboptimality is” in hierarchical RL. To define and distinguish **execution suboptimality** and **policy suboptimality**, it is hard to avoid at least two canonical execution modes:
> > >
> > > - **SME** (the standard mode used in existing HRL implementations), and
> > > - **ME** (the mode we analyze in depth).
> > >
> > > Our key observation is that most existing HRL methods **tie the duration of a subtask call directly to the time needed to complete that subtask**, for all three roles:
> > >
> > > - behavior policy during data collection,
> > > - target policy during optimization, and
> > > - deployment policy at test time.
> > >
> > > We show that this unified SME-style treatment of duration is exactly what leads to **two distinct types of suboptimality**:
> > >
> > > 1. **Execution suboptimality:**
> > >    Even after the hierarchical policies at all levels have been learned, using SME at deployment can be strictly worse than executing the same policies under ME.
> > >
> > > 2. **Policy suboptimality:**
> > >    When the target policy is optimized under SME, it may converge to an SME-optimal policy that is no longer optimal when executed under ME at deployment.
> > >
> > > Our theoretical and algorithmic proposals can be summarized as:
> > >
> > > 1. **Deployment policy:** use ME to remove execution suboptimality and maximize the return of the learned hierarchical policy.
> > > 2. **Behavior policy (for data collection):** use SME so that subtasks are executed fully and can drive the agent to more meaningful states, while still being compatible with efficient learning.
> > > 3. **Target policy:** use ME so that training directly optimizes the policy for the deployment execution mode, eliminating policy suboptimality.

---

> > > > ### Author Response · Authors · 2025-11-22
> > > > **Response to Reviewer q9eR (4/5)**
> > > >
> > > > #### 3.2 Why ME vs SME instead of directly searching for an “optimal stopping time”
> > > >
> > > > You suggested a very interesting direction: instead of focusing on the binary ME/SME distinction, one could analyze how the optimality gap depends on subtask execution length and search for an “optimal stopping time” that keeps the gap negligible while preserving the benefits of temporal abstraction.
> > > >
> > > > This is exactly the kind of question our formalism is designed to support. In our framework:
> > > >
> > > > - The **execution mode** is controlled by a decision-interruption function $\hat\beta$, which can encode various termination / interruption rules;
> > > > - SME and ME correspond to two extreme settings of $\hat\beta$;
> > > > - Intermediate choices of $\hat\beta$ correspond to subtask execution lengths between these extremes, and can be interpreted as different “stopping rules.”
> > > >
> > > > One of the main theoretical results in our appendix is an **execution optimality theorem**, which (under Markov assumptions and within our class of execution modes) shows that, for a node whose policy is optimal under its execution mode:
> > > >
> > > > - ME not only dominates SME, but also has the strongest return guarantee within the execution-mode family considered in our framework (no other execution mode strictly outperforms it under the same optimal policy assumption).
> > > >
> > > > We acknowledge that this theorem has limitations:
> > > >
> > > > - In multi-level hierarchies, additional conditions are needed (e.g., Corollary C.1 in the appendix).
> > > > - The comparison is made under the assumption that the node’s policy is optimal for the given execution mode. If the policy is not optimal, we cannot guarantee that:
> > > >   (1) switching to ME will always improve its performance, or
> > > >   (2) no other execution mode would yield a higher return.
> > > >
> > > > From our perspective, your idea of searching for an “optimal stopping time” is very meaningful! However, according to the core viewpoint of our paper, this “optimal” choice must be considered separately for **three different roles**: the target policy, the behavior policy, and the deployment policy:
> > > >
> > > > 1. **Behavior policy (exploration and sampling efficiency):**
> > > >    Use methods such as bottleneck-state discovery or unsupervised skill discovery to find subtasks that enable more efficient exploration, and then execute them under SME so that tasks at all levels remain learnable (Section 4.2).
> > > >
> > > > 2. **Target policy (exploitation and learnability):**
> > > >    On one hand, we can learn an execution mode that maximizes the return of the **current** policy (e.g., via the termination gradient theorem in OC). Our framework instead chooses ME, which is equivalent to directly using the execution mode that maximizes the return of the **optimal** policy, rather than learning an execution mode that maximizes the return of the **current** policy. On the other hand, we must avoid being too far off-policy from the behavior execution mode to maintain learning efficiency.
> > > >
> > > > 3. **Deployment policy (return maximization):**
> > > >    Use the execution mode that maximizes the return of the **current** learned policy at test time.
> > > >
> > > > In existing HRL frameworks, the execution modes of the three policies are *tied together*, so there is effectively only a **single** notion of optimal stopping time. In our framework, these three execution modes are **decoupled**, leading to three different notions of “optimal,” which allows us to optimize the stopping time in a more principled way. This is the key difference between our work and conventional HRL frameworks, and your suggestion is precisely an important direction for our future work.
> > > >
> > > >
> > > >
> > > > ---

---

> > > > > ### Author Response · Authors · 2025-11-22
> > > > > **Response to Reviewer q9eR (5/5)**
> > > > >
> > > > > ### 4. Presentation issues: figures and the proof sketch
> > > > >
> > > > > > “Aside from Figure 1, all the figures are hard to follow… Figure 2 is unclear regarding where the legend ends and where SME/ME begins. Figure 3 has no legend… In Figure 5a) right, one of the baselines is cropped out… The proof sketch of Theorem 1 introduces too many acronyms and concepts without explanation, making it hard to understand.”
> > > > >
> > > > > We very much appreciate these concrete suggestions. We will revise the presentation as follows:
> > > > >
> > > > > - **Figures**
> > > > >   - Add annotations and, where appropriate, split complex plots into subfigures (a), (b), etc., each with a clear title.
> > > > >   - Clarify in Figure 2 where the legend ends and where SME/ME markers begin, and add a proper legend for Figure 3.
> > > > >   - Add instructions on cropping the RL experiment.
> > > > >
> > > > > - **Proof sketch of Theorem 1 (Theorem 4.1 in the main text)**
> > > > >   - Explicitly indicate at the beginning that this is a proof sketch for Theorem 4.1.
> > > > >   - Bring the most important theorems from the appendix into the main text with brief explanations, so that the sketch reads as a clear proof roadmap: new concepts → formalization → key lemmas → final theorem.
> > > > >
> > > > > Our goal is to make the figures and the proof sketch easy to follow for readers in the broader community.
> > > > >
> > > > > ---
> > > > >
> > > > > If any part of our response regarding execution suboptimality or policy suboptimality remains unclear, we would be very happy to further clarify the definitions or the proofs. Once again, we thank you for your thoughtful and detailed comments, which have helped us significantly improve both the clarity and the focus of our work.

---

> > > > > > ### Comment · Reviewer_q9eR · 2025-11-24
> > > > > > **Response to Author Rebuttals**
> > > > > >
> > > > > > I thank the authors for their thorough response to my review.
> > > > > >
> > > > > > For now, let us just focus on my primary concern: the motivation and intuition behind ME is not clear.
> > > > > >
> > > > > > In particular, my understanding is that the authors claim that, to maximize the return, any trained hierarchical policy is best executed by only executing it for a single time step (i.e. Markov Execution (ME)). This has several counterintuitive ramifications that the paper, in my view, does not address adequately.
> > > > > >
> > > > > > First, by definition, ME only executes single-step primitive actions, so to call this a hierarchical method seems imprecise. Rather, what seems to be happening is that ME effectively reduces a hierarchical policy to an 'equivalent' flat policy. There is nothing wrong with this, but to frame it as a hierarchical method is a bit misleading and counterintuitive when a decision must be made at every single time step.
> > > > > >
> > > > > > Next, if indeed ME is the optimal way to execute a hierarchical policy, then I question (from an intuitive perspective) the validity of the training. That is, if the training was done appropriately, would the learned subtasks not capture this? I see this as two cases: 1) the agent had the flexibility to learn when to stop executing the subtask during training, in which case I wonder why the agent would not learn that it is optimal to stop after a single time step; or 2) the agent did not have the flexibility to learn when to stop the subtask, in which case I think that a fair comparison was not made between training and execution which would greatly weaken the claims made about ME.
> > > > > >
> > > > > > Third, I think the work could benefit from a more thorough exploration of the findings to yield more insights. In particular, let us assume that the claims made about ME are correct as stated in the paper. Given that the executed policy effectively acts as a flat policy, why does it perform so much better than a regular flat policy? I know that the authors provide theoretical results in this matter, but the stark difference in the empirical results (vs a regular flat policy) is counterintuitive; I would expect there to be a difference in performance, but not that big of a difference. It seems like effectively what is happening is that the subtasks act as a sort of 'filter' for the primitive actions. That is, instead of having to choose from the entire action space at execution, the agent only needs to pick from the primitive actions that would be executed first in the subtasks. This 'action filtering via subtask learning' could be a more insightful way to frame the paper than what is presented.
> > > > > >
> > > > > > All in all, I do not question the validity of the results presented by the authors. However, these results imply several counterintuitive ramifications that need to be addressed in the paper prior to publication.

---

> > > > > > > ### Author Response · Authors · 2025-12-01
> > > > > > > **Response to Reviewer q9eR (1/5)**
> > > > > > >
> > > > > > > We sincerely thank you for your timely and careful response. You have articulated those “counterintuitive” consequences very clearly, which greatly helps us clarify the standpoint and contributions of this paper. Our work focuses on the following question:
> > > > > > >
> > > > > > > **Given that a hierarchical structure has already been adopted**, how do different execution modes (SME/ME) affect and induce suboptimality for the three roles of **target policy, behavior policy, and deployment policy**, rather than proposing an entirely new “hierarchical method”?
> > > > > > >
> > > > > > > Below we respond to your points one by one.
> > > > > > >
> > > > > > > ---
> > > > > > >
> > > > > > > >My understanding is that the authors claim: in order to maximize return, any already trained hierarchical policy is best executed by only executing it for a single time step (i.e., Markov Execution, ME).
> > > > > > >
> > > > > > > **On the precise meaning of “well trained”:**
> > > > > > > Yes, your understanding is correct. Our strict definition of “well trained” is:
> > > > > > >
> > > > > > > **Under a given execution mode, each node’s policy is optimal in all states.**
> > > > > > >
> > > > > > > ---
> > > > > > >
> > > > > > > >From the definition, ME only executes single-step primitive actions, so calling it a hierarchical method does not seem entirely accurate. It looks more like ME reduces a hierarchical policy to an “equivalent” flat policy. There is nothing wrong with doing so, but describing it as a “hierarchical method” when a decision has to be made at every step feels somewhat misleading and counterintuitive to me.
> > > > > > >
> > > > > > > **The relationship between ME and “hierarchical methods”:**
> > > > > > > We believe this is essentially a matter of how one **defines** the notion of “hierarchy”.
> > > > > > >
> > > > > > > - **Definition 1 (structural hierarchy):**
> > > > > > >   In standard RL, given the current state $s$, the agent makes a one-step decision $ \tau_0 \to a$ to select an action.
> > > > > > >   After introducing hierarchy, this becomes a multi-step mapping
> > > > > > >   $$
> > > > > > >   \tau_0 \to \tau_1 \to \dots \to \tau_l \to a.
> > > > > > >   $$
> > > > > > >   Since the policy mapping is composed across multiple task levels, we may call this “hierarchical” in the sense of being **non-flat in task space**.
> > > > > > >
> > > > > > > - **Definition 2 (temporal hierarchy):**
> > > > > > >   Based on Definition 1, we further require that different levels have different decision frequencies. For example,
> > > > > > >   - $\tau_0$ makes a decision once every 100 frames,
> > > > > > >   - $\tau_1$ makes a decision once every 10 frames,
> > > > > > >   - $\tau_2$ (the lowest level) makes a decision at every frame to choose $a$.
> > > > > > >
> > > > > > >   This corresponds to the “non-flat in time” style of hierarchy used in existing HRL via SME (SMDP-style hierarchical execution semantics).
> > > > > > >
> > > > > > > In fact, there is no real contradiction between these two viewpoints. We apologize for any confusion in our discussion that might have given the impression that ME itself is a new “hierarchical method”.
> > > > > > >
> > > > > > > In the paper, we do **not** present ME as a standalone hierarchical method. Rather, we aim to replace certain components in existing HRL frameworks with a more reasonable, **temporally flat** (but **structurally hierarchical**) execution scheme, in order to resolve the SMDP-induced suboptimality that current “hierarchical methods” suffer from.
> > > > > > >
> > > > > > > Thus, ME is a **replaceable design component** inside existing hierarchical methods, not a hierarchical method per se.
> > > > > > >
> > > > > > > That said, we can clarify:
> > > > > > >
> > > > > > > - Under **Definition 1 (structural hierarchy)**, an algorithm improved with ME still belongs to “hierarchical methods” in that sense.
> > > > > > > - Under **Definition 2 (temporal hierarchy)**, an algorithm improved with ME is no longer “hierarchical” in the SMDP temporal sense.
> > > > > > >
> > > > > > > **Why the definition feels counterintuitive:**
> > > > > > > We believe the discomfort may come from a natural (but not our) assumption that in our framework the **target policy, behavior policy, and deployment policy all use ME**. If that were the case, then the subtasks would indeed act merely as “filters”, as you suggested, which is far from enough to handle sparse-reward problems.
> > > > > > >
> > > > > > > However, this is **not** what we do in the paper. In our design:
> > > > > > >
> > > > > > > - the **behavior policy** uses SME, and
> > > > > > > - only the **target policy** and the **deployment policy** use ME.
> > > > > > >
> > > > > > > The key difference between our approach and existing work is:
> > > > > > >
> > > > > > > Given an existing “hierarchical method”, we argue that one should **independently decide** whether to use the temporally flat ME structure for each of the three roles:
> > > > > > > **target policy**, **behavior policy**, and **deployment policy**.
> > > > > > > These are three different design questions, *not* a single forced tradeoff between SME and ME (different termination times)!
> > > > > > >
> > > > > > > We strongly believe this separation is very important for the future development of HRL.
> > > > > > >
> > > > > > > In Section 4, we analyze in detail why **the target and deployment policies should use ME** (where existing frameworks use SME), and why **the behavior policy should not use ME** (and should remain SME as in existing frameworks). Later, we also provide simple examples to help make this intuition more immediate.

---

> > > > > > > > ### Author Response · Authors · 2025-12-01
> > > > > > > > **Response to Reviewer q9eR (2/5)**
> > > > > > > >
> > > > > > > > >Next, if ME is indeed the optimal way to execute a hierarchical policy, I would intuitively question the reasonableness of the training process. That is, if training is “appropriate”, shouldn’t the learned subtasks already capture this? I would consider two cases:
> > > > > > > > >1) If the agent has the flexibility to learn “when to stop subtasks” during training, then why wouldn’t it learn that “stopping after a single step” is optimal?
> > > > > > > > >2) If the agent does not have such flexibility during training, then it seems training and execution are not being compared “fairly”, which would greatly weaken the claims about ME.
> > > > > > > >
> > > > > > > > **Regarding the first question: why wouldn’t the agent learn that “stop after one step” is optimal?**
> > > > > > > > This is a very important question.
> > > > > > > >
> > > > > > > > In the OC framework, the agent uses the “termination gradient theorem” to learn when to stop options. The parameters of the termination function are optimized precisely to **maximize upper-level return**. However, this optimization target leads to an “undesirable” effect: many OC-related papers have reported that Option-Critic tends to **shorten options** [1], even collapsing them to length 1, effectively resulting in temporally flat decisions. This is exactly the phenomenon that a large body of subsequent OC work tries to mitigate.
> > > > > > > >
> > > > > > > > On the one hand, optimizing termination purely for upper-level return naturally pushes options toward ME. This is fully consistent with our conclusion: **ME execution maximizes upper-level return**. The reason why options do not completely degenerate into single-step execution (pure ME) is:
> > > > > > > >
> > > > > > > > When the current option remains optimal for the next few steps, “terminating and re-selecting that option” and “continuing the current option” yield the same return. In this case, the gradient is zero; whether or not we terminate the option makes no difference to the return.
> > > > > > > >
> > > > > > > > Under optimal high-level policy and optimal termination, executing under pure ME and executing under the learned SME termination function yield the same return—they are effectively equivalent to “stopping after one step”.
> > > > > > > >
> > > > > > > > However, once the high-level policy is optimal but the learned termination function is **not** optimal (e.g., many later OC variants explicitly discourage single-step termination and encourage options to persist longer; neural approximation and sampling noise also prevent the termination from being truly optimal [2][3][4]), then the learned SME policy becomes **weaker** than the theoretically optimal ME execution. This is precisely what our ESIF comparison experiments verify.
> > > > > > > >
> > > > > > > > From this perspective, **a termination condition optimized solely for return maximization does not actually need to be learned**—there exists a simple theoretical optimum, namely ME.

---

> > > > > > > > > ### Author Response · Authors · 2025-12-01
> > > > > > > > > **Response to Reviewer q9eR (3/5)**
> > > > > > > > >
> > > > > > > > > On the other hand, why do we still say that “temporally flat ME is bad” in practice?
> > > > > > > > > The reason is that in our experiments, training with temporally flat ME (using ME for both behavior policy and target policy) behaves very poorly: on tasks where HRL excels, ME training performs almost as badly as plain RL. Unlike previous work, however, we do not simply take ME as “bad” by default. Instead, we carefully analyze the role that ME plays during training by **separating the target policy from the behavior policy**.
> > > > > > > > >
> > > > > > > > > Our finding is:
> > > > > > > > >
> > > > > > > > > For the **target policy**, using ME is actually reasonable.
> > > > > > > > > The real culprit is using ME for the **behavior policy**.
> > > > > > > > >
> > > > > > > > > 1. With SME behavior, the agent can efficiently reach and sample **bottleneck states**. Intuitively, in a multi-room navigation task, SME allows us to call a subtask “go from door A to door B” once and execute it to completion, moving directly between bottlenecks.
> > > > > > > > >    Under ME behavior, in contrast, at each frame inside the room the agent must reconsider which door (B, C, or D) to head toward. This makes it very unlikely to sample trajectories that successfully reach the correct next doorway.
> > > > > > > > >
> > > > > > > > > 2. As analyzed in Section 4.2, ME behavior causes lower-level subtasks to be frequently interrupted, making their policies much harder to learn (off-policy instability).
> > > > > > > > >
> > > > > > > > > This actually **confirms your intuition**: using ME for the behavior policy is indeed problematic—**the issue is with the behavior policy, not with ME as the target and deployment execution mode**.
> > > > > > > > >
> > > > > > > > > Once we have collected “good” samples using SME for the behavior policy, the target policy does **not** need to plan the SME-optimal solution (which still suffers from SMDP suboptimality). Instead, it can directly plan the truly optimal solution under ME (via our hierarchical policy improvement theorem).
> > > > > > > > >
> > > > > > > > > It is precisely this careful analysis that leads to the core insights of this paper.
> > > > > > > > >
> > > > > > > > > Therefore, OC and its variants within existing HRL frameworks face an inherent tension:
> > > > > > > > >
> > > > > > > > > - If we want strong exploration, we use long-duration options that can efficiently reach bottleneck states—but this produces SMDP-induced suboptimality in the final solution.
> > > > > > > > > - If we want to maximize return under pure value semantics, termination will tend to degenerate toward per-step termination (ME), which hurts exploration and may even make training ineffective.
> > > > > > > > >
> > > > > > > > > A natural resolution of this tension is exactly what we propose:
> > > > > > > > >
> > > > > > > > > - The component that should use long-duration, bottleneck-reaching options is the **behavior policy**. The behavior policy should use SME and learn termination times that are good for exploration.
> > > > > > > > > - The component that should use ME (to maximize return) is the **target policy**. The termination gradient theorem, when used purely for return maximization, should conceptually apply only to the target policy. In fact, under the conditions of our theorem, the termination time for the target policy does not even need to be learned: ME is already optimal. When those conditions are not strictly satisfied, learning termination (via a generalized execution improvement theorem, as we provide) may still be beneficial.
> > > > > > > > >
> > > > > > > > > From our perspective, this tension **should not exist in the first place**, and resolving it in this way is important for the future of HRL.
> > > > > > > > >
> > > > > > > > > **Regarding the second question:**
> > > > > > > > > In our experiments, the OC baseline used for theoretical validation is precisely a scheme with such “flexibility” (learnable termination). However, its termination function is only an approximate neural solution; It did not learn the optimal termination function (i.e., ME) to maximize the return, so it still suffers from SMDP suboptimality. This is why our ESIF/PSIF frameworks can further improve the return.
> > > > > > > > >
> > > > > > > > > In the goal-based HAC setting, subtask horizons are fixed and the original algorithm does **not** learn termination times. Our experiments show that using the “optimal stopping time” (i.e., ME) can achieve higher return, which in turn illustrates that **having such flexibility is actually important**.

---

> > > > > > > > > > ### Author Response · Authors · 2025-12-01
> > > > > > > > > > **Response to Reviewer q9eR (4/5)**
> > > > > > > > > >
> > > > > > > > > > >Third, I believe this work would be more valuable if it mined these findings more deeply and provided more intuitive insights. Specifically, let us assume the claims about ME are correct. Since the executed policy behaves almost like a flat policy, **why does it perform so much better than a regular flat policy?** I understand that the authors provide theoretical results, but the large empirical performance gap (vs. flat policy) is still counterintuitive. I would expect a difference, but not such a huge one.
> > > > > > > > > >
> > > > > > > > > > Please note that our training process is **very different** from standard flat RL. During training, only the **target policy** uses ME; the **behavior policy** uses SME, **not** ME.
> > > > > > > > > >
> > > > > > > > > > This is crucial, because it allows us—just like standard HRL—to sample “good” bottleneck states.
> > > > > > > > > >
> > > > > > > > > > If the behavior policy during training also used ME, then subtasks would indeed only play the role of filters, as you suggested. Returning to the multi-room example: even after we have trained subtasks, at each frame inside the room we still must decide which door among B, C, and D to move toward. Intuitively, if we need 10 steps to cross the room, we might need to try up to $3^{10}$ different trajectories (HRL, but behavior policy and target policy both use ME). Although this is much smaller than $|A|^{10}$, it is still unacceptable in practice (flat RL). This is also why, intuitively, ME feels “bad” when used for behavior.
> > > > > > > > > >
> > > > > > > > > > However, when the behavior policy uses SME (as in our method), sampling can be done by **fully executing** a subtask to directly reach one of the doors B, C, or D.
> > > > > > > > > >
> > > > > > > > > > As a consequence, in our experiments:
> > > > > > > > > >
> > > > > > > > > > - Both our method and the original HRL algorithms (OC, HAC) can successfully explore to the goal by calling subtasks under SME.
> > > > > > > > > > - By contrast, the flat RL baselines in our sparse-reward environments fail to successfully reach the goal within the given budget, and thus cannot really talk about “planning an optimal strategy” at all.
> > > > > > > > > >
> > > > > > > > > > Therefore, the **large performance gap** we see in experiments is primarily the gap between:
> > > > > > > > > >
> > > > > > > > > > **Existing HRL frameworks vs. flat RL** on sparse-reward problems (for a fixed training budget, HRL is nearly optimal, while RL has not yet explored to the goal / has too few successful trajectories to plan correctly, as also observed numerically in OC’s original four-room-with-teleport experiments),
> > > > > > > > > >
> > > > > > > > > > rather than the gap between:
> > > > > > > > > >
> > > > > > > > > > **Our improved method vs. existing HRL frameworks**.
> > > > > > > > > >
> > > > > > > > > > The reason our method is so much better than flat RL is that we **stand on the shoulders of HRL**: we inherit the strong exploration capability from HRL (through SME behavior), and on top of that we further optimize away the SMDP-induced suboptimality.
> > > > > > > > > >
> > > > > > > > > >
> > > > > > > > > > It seems that, in actual execution, subtasks behave like a kind of “filter” for primitive actions. That is, during execution, the agent no longer needs to select from the entire action space, but only from the small set of actions that appear as the *first-step actions* of each subtask. Interpreting this as “action filtering via subtask learning” may provide more insight than the current presentation in the paper.
> > > > > > > > > >
> > > > > > > > > > We fully agree that:
> > > > > > > > > >
> > > > > > > > > > - “action filtering via subtask learning” (non-flat in task space)
> > > > > > > > > >
> > > > > > > > > > Even further “discovering subtasks that better reach bottleneck states” (non-flat in decision time, one of the mainstream research directions of HRL),
> > > > > > > > > > are both extremely meaningful and are indeed central goals of HRL. Our work aims to provide a more principled **framework setting** for these research directions, by emphasizing that:
> > > > > > > > > >
> > > > > > > > > > When thinking about such issues, we must treat the **target policy**, **behavior policy**, and **deployment policy** separately.
> > > > > > > > > >
> > > > > > > > > > In our view, hierarchy (maybe action filtering, SME, or others tricks) should primarily benefit the **behavior policy**, rather than the target policy.
> > > > > > > > > >
> > > > > > > > > > To use a simple analogy: among 10,000 numbers, only 10 are very large. We want to find the largest number. If we are allowed to sample only 10 times, the hard part is **how to efficiently sample those 10 large numbers** (sampling), not how to compare the 10 sampled numbers (planning).
> > > > > > > > > >
> > > > > > > > > > - To efficiently sample those 10 “good” numbers (sampling: behavior policy), we use some tricks (subtasks, hierarchy).
> > > > > > > > > > - Once we have obtained these 10 number samples, we no longer need these tricks to compare which one is largest (planning: target policy).
> > > > > > > > > >
> > > > > > > > > > This is exactly the central idea of our work.
> > > > > > > > > >
> > > > > > > > > > Our contribution is not in competition with these existing research directions. Instead, we hope to provide a stronger foundation for them by clarifying:
> > > > > > > > > >
> > > > > > > > > > In hierarchical RL, the execution modes for **behavior**, **target**, and **deployment** should be designed separately.
> > > > > > > > > >
> > > > > > > > > > We hope that this discussion can resolve your concerns and help further highlight the theoretical importance of our results.

---

> ### Author Response · Authors · 2025-12-01
> **Response to Reviewer q9eR (5/5)**
>
> Reference:
>
> [1] Harb J, Bacon P L, Klissarov M, et al. When waiting is not an option: Learning options with a deliberation cost[C]//Proceedings of the AAAI Conference on Artificial Intelligence. 2018, 32(1).
>
> [2] Harutyunyan A, Vrancx P, Bacon P L, et al. Learning with options that terminate off-policy[C]//Proceedings of the AAAI Conference on Artificial Intelligence. 2018, 32(1).
>
> [3] Khetarpal K, Precup D. Learning options with interest functions[C]//Proceedings of the AAAI Conference on Artificial Intelligence. 2019, 33(01): 9955-9956.
>
> [4] Zhu X, Zhao L, Zhu W. Salience Interest Option: Temporal abstraction with salience interest functions[J]. Neural Networks, 2024, 176: 106342.

---

### Official Review · Reviewer_YuW2 · 2025-10-29

**Soundness:** 3
**Presentation:** 3
**Contribution:** 3
**Rating:** 6
**Confidence:** 3

**Summary:**

The paper identifies a structural flaw in SMDP-based HRL: committing to a subtask until termination limits adaptability and optimality. It decomposes this limitation into execution and policy suboptimality, formally proving both through the Hierarchical Policy Improvement and Optimal Execution Theorems. By introducing Task Trees and Execution Trees, the authors decouple task decomposition from policy scheduling, enabling flexible analysis of Markov vs. Semi-Markov execution. They develop a Unified Value Function for HRL (UVFH) and a Generalized Hierarchical Bellman Equation (GHBE) to compute multi-level values under arbitrary execution modes. Building on this, three frameworks—ESIF, PSIF-1S, and PSIF-2S—address the identified suboptimalities by separating target and behavior policies. Experiments across HRL benchmarks demonstrate improved adaptability, efficiency, and returns under the proposed frameworks

**Strengths:**

1.	Interesting to note the issues arising from the two distinct components: execution suboptimality and policy suboptimality, which deserves attention from RL community.
2.	Rigorous derivation of Generalized Hierarch Bellman Equations: Introduce the Task Tree (defining optimization objectives) and the Execution Tree (defining scheduling/interruption), leading to the Generalized Hierarchical Bellman Equation (GHBE) that provides a unified framework for analyzing any execution mode

**Weaknesses:**

Overall, the paper is hard to follow. For example, Fiture 3 is hard to understand even after reading related texts multiple times. If the author made a connection between mathematical notation (such as \pi^{-} and \pi^*) to whatever in the figure (such as paths of certain color), it would have been easier to follow.

The paper is full of new concepts, numerous defintions, etc. I would rather see a more abstract version of the paper with all the technical details in Appendix. The space could have been better used for exposition of the key ideas, the workings of proposed algorithms, and limitations of the work.

Off-policy problem due to the discrepancy between the behavior execution mode (SME used to collect data) and the target execution mode (ME used in value backups). The issues seems outstanding and hence deserves numerical analysis, which is not done in the current version of the paper.

**Questions:**

In option discovery, the termination function can be gradually optimized, allowing more exploration early in the training and later more on exploitation. Would more sensible option discovery address the issues of SME vs ME?

---

> ### Author Response · Authors · 2025-11-22
> **Response to Reviewer YuW2 (1/5)**
>
> We sincerely thank Reviewer YuW2 for the thorough reading and constructive feedback.
>
> ---
>
> ### On the overall readability and Figure 3
>
> > “Overall, the paper is hard to follow. For example, Figure 3 is hard to understand even after reading related texts multiple times. If the author made a connection between mathematical notation (such as $\pi^{-}$ and $\pi^*$) to whatever in the figure (such as paths of certain color), it would have been easier to follow.”
>
> Thank you very much for pointing this out. We completely agree that the current version has room for improvement in terms of readability. In the revision, we will make **Figure 3** much clearer by explicitly aligning **mathematical notation** (e.g., $\pi^{-}$, $\pi^*$, different value functions) with the **colored paths, labels, and regions** in the figure, so that readers can immediately see:
>
> - which path corresponds to the SME-optimal policy $\pi^{-}$,
> - which path corresponds to the ME-optimal policy $\pi^*$, and
> - how the two types of suboptimality (execution vs. policy) are illustrated.
>
> We have already explored this direction in more detail in **Appendix B**, where **Figures 6, 7, and 8** provide more elaborate legends, annotations, and explanations. In the revision, we will bring this level of clarity into the main text figure and its caption, so that readers can quickly grasp the intuitive meaning of the two suboptimalities from Figure 3 alone.
>
> ---

---

> > ### Author Response · Authors · 2025-11-22
> > **Response to Reviewer YuW2 (2/5)**
> >
> > ### On too many concepts and the balance between abstraction and technical detail
> >
> > > “The paper is full of new concepts, numerous definitions, etc. I would rather see a more abstract version of the paper with all the technical details in Appendix. The space could have been better used for exposition of the key ideas, the workings of proposed algorithms, and limitations of the work.”
> >
> > We very much agree with your perspective. Our intention has also been to use the limited main-text space to emphasize **key ideas**, **algorithmic insights**, and **limitations**, rather than overwhelming readers with technical minutiae.
> >
> > The current structure was designed with the following goals:
> >
> > - In **Section 3**, we aim to answer the question:
> >   **“What exactly is SMDP suboptimality?”**
> >   and to decompose it into **execution suboptimality** and **policy suboptimality**.
> >
> > - In **Section 4**, our goal is to present a **core theoretical result**—the “SMDP suboptimality theory”—and to provide a proof sketch that shows the main reasoning steps:
> >   - introducing more appropriate concepts for this problem (Task Trees, Execution Trees),
> >   - building new formal tools (unified value functions and generalized Bellman equations),
> >   - and proving the key lemmas and theorems.
> >   Many of the detailed lemmas and technical results are indeed moved to the **appendix**, and only the main theorem is kept in the main text.
> >
> > - In **Sections 4.1–4.3**, our analysis is meant to convey the **central design insight**:
> >   current HRL execution modes have a structural issue.
> >   Specifically, for both training and deployment, existing HRL systems implicitly **bind**
> >   - “the duration for which a subtask is executed after being called”
> >   with
> >   - “the time it takes for the subtask to complete its own local goal”.
> >   We show that this particular binding of execution semantics is what causes both **execution suboptimality** and **policy suboptimality**.
> >
> >   Concretely:
> >   1. To fix **execution suboptimality** once we have already learned a good hierarchical policy (e.g., in a 2-level setting), we should use **ME at deployment**, so that we always choose the best action at each step and maximize the deployed return.
> >   2. During training, to reach “semantically meaningful” states via full subtask execution while avoiding making lower-level learning too difficult, the **behavior policy** should use **SME**.
> >   3. To fix **policy suboptimality** during training, the **target policy** should use **ME**, so that the optimization target matches the best achievable performance under ME.
> >
> > Decoupling the execution modes of **three different policies** (behavior, target, deployment) is exactly the core design idea behind our proposed frameworks for addressing SMDP suboptimality. The performance and convergence of these frameworks are backed by the theoretical results (such as the Optimal Execution Theorem and the decomposition theorem) whose proof sketches appear in Section 4, with the detailed proofs in the appendix.
> >
> > As you noted, the paper is close to **40 pages in total**, with **14 pages of proofs** in the appendix. This indeed reflects the fact that we have already moved a significant amount of technical detail out of the main text. After thoroughly working through the problem, we found that:
> >
> > - the **formal** resolution of the issue is technically involved, and
> > - making readers truly understand *“what the core problem is”* and *“where the existing framework is structurally flawed”* is even more challenging.
> >
> > That said, we strongly agree with your suggestion that the main text can still be **more abstract and more focused on the core ideas**.
> >
> > We would also like to mention that **another reviewer (ZE4K)** suggested the opposite direction—namely, to move more key theorems from the appendix back into the main text. In the revision, we will strive to **balance** these two perspectives:
> >
> > - Moving some **key theoretical results** (e.g., GHBE, the Optimal Execution Theorem, and the decomposition theorem) from the appendix into the main text;
> > - At the same time, using **clear figures, examples, and high-level explanations** in the main text to convey the core ideas, while keeping detailed technical proofs in the appendix.
> >
> > ---

---

> > > ### Author Response · Authors · 2025-11-22
> > > **Response to Reviewer YuW2 (3/5)**
> > >
> > > ### On the off-policy issue caused by SME behavior and ME target execution modes
> > >
> > > > “Off-policy problem due to the discrepancy between the behavior execution mode (SME used to collect data) and the target execution mode (ME used in value bootstrap). The issues seems outstanding and hence deserves numerical analysis, which is not done in the current version of the paper.”
> > >
> > > You have raised a very important point. In fact, this is a central aspect we intentionally confront in the design of the **PSIF frameworks**: using **SME** for the behavior execution mode and **ME** for the target execution mode is essentially a form of **off-policy learning across execution modes**.
> > >
> > > Briefly, the OC implementation used in our experiments performs single-step bootstrapping, similar to Q-learning, and can correctly update values in an off-policy setting.
> > >
> > > More deeply, within our theoretical framework, off-policy learning is reasonable and convergent mainly due to two key factors:
> > >
> > > 1. **Unified value definition and frame-wise GHBE bootstraps**
> > >    - Through UVFH and GHBE, we rewrite multi-level values under arbitrary execution modes into a form that **always bootstraps from the next frame**, rather than from the (random) subtask-termination time.
> > >    - Under this definition, each per-timestep transition collected by the SME behavior policy can be treated as a valid training sample for the ME target policy, as long as sampling coverage is ensured so that the value can be learned correctly (analogous to standard off-policy RL, where Q-learning uses $\epsilon$-greedy behavior samples to update a greedy target policy).
> > >
> > > 2. **$\epsilon$-SME guarantees state–level support coverage**
> > >    - In both our theoretical analysis and the OC framework, the “SME” we adopt gives a non-zero probability at each frame (i.e., $\hat\beta$ is never identically 0) of interrupting the current subtask and returning control to the upper level, ensuring that over a long time scale the behavior policy can visit the state–level pairs that are reachable under ME.
> > >    - This is reflected in the conditions of Lemma C.1 in the appendix, which ensure that the behavior distribution provides sufficient support for the target distribution, thereby making off-policy updates possible.
> > >
> > > From an empirical standpoint, **PSIF-1S and PSIF-2S** exhibit **stable training curves and consistently outperform the baselines** across all benchmarks. This provides empirical evidence that cross-execution-mode off-policy updates do *not* cause severe instability in our setting.
> > >
> > > We fully agree with you that such cross-mode off-policy phenomena are themselves an interesting research topic. In future work, we plan to study this issue more deeply, and also explore whether there are alternative approaches beyond the one proposed in our paper.
> > >
> > > ---

---

> > > > ### Author Response · Authors · 2025-11-22
> > > > **Response to Reviewer YuW2 (4/5)**
> > > >
> > > > ### On option discovery and whether better termination functions can resolve ME vs. SME
> > > >
> > > > > “In option discovery, the termination function can be gradually optimized, allowing more exploration early in the training and later more on exploitation. Would more sensible option discovery address the issues of SME vs ME?”
> > > >
> > > > This is an excellent and thought-provoking question. We also believe that **learning/optimizing termination functions** is closely related to our analysis of execution modes.
> > > >
> > > > Our view can be summarized as follows:
> > > >
> > > > 1. **Better termination functions may alleviate some issues, but cannot remove the structural suboptimality inherent to SMDP semantics**
> > > >
> > > >    - In most option discovery work, the termination function $\beta$ simultaneously determines **(i)** when a subtask is considered “completed” (i.e., its *definition*), and **(ii)** when control returns to the higher level (i.e., its *execution semantics*). This remains within the classic SMDP framework: once a subtask is chosen, it is executed under SME semantics until it terminates.
> > > >    - The execution and policy suboptimalities we prove via the Optimal Execution Theorem and the decomposition theorem hold **for any given $\beta$**. As long as the execution mode is tied to subtask termination (i.e., $\hat{\beta} = \beta$), there is a structural disadvantage compared to ME.
> > > >    - In other words, even if option discovery learns a “better” termination function, it is still operating within SMDP semantics and only adjusts “when a subtask is considered finished”, without decoupling **“task termination”** from **“when decisions are allowed to happen”**.
> > > >
> > > > 2. **Our framework’s contribution is to explicitly decouple “task termination” and “execution interruption”**
> > > >
> > > >    - We use the **Task Tree** to model **when a task truly terminates and how value bootstraps upward** (optimization objective), and use the **Execution Tree** with $\hat{\beta}$ to model **which level makes decisions at each time step** (execution scheduling).
> > > >    - Within this framework, even if $\beta$ is learned via option discovery, as long as behavior/target/deployment policies still enforce $\hat{\beta} = \beta$ (standard SMDP/SME semantics), they remain subject to the execution and policy suboptimalities we identify.
> > > >    - Only when we explicitly allow **$\hat{\beta} \neq \beta$**, for example using ME (frequent interruption) in deployment and for target policies, while using SME for behavior policies, can we theoretically escape these structural suboptimalities.

---

> > > > > ### Author Response · Authors · 2025-11-22
> > > > > **Response to Reviewer YuW2 (5/5)**
> > > > >
> > > > > 3. **What is a “sensible” option discovery?**
> > > > >
> > > > >     - One of our main theoretical results (the execution optimality theorem in the appendix), under the Markov assumption and within the class of execution modes we consider, shows the following: assuming that the policy at a node is already optimal **under its own execution mode**,
> > > > >
> > > > >     - ME not only dominates SME, but from the perspective of return, ME has the strongest performance guarantee **within the family of execution modes considered in our framework**—there is no other execution mode that is strictly better than ME, under the assumption that the policy is optimal for that mode.
> > > > >
> > > > >     - We also acknowledge that this conclusion has certain limitations:
> > > > >
> > > > >       - In multi-level hierarchical structures, additional conditions are needed (e.g., Corollary C.1 in the appendix);
> > > > >       - The comparison between execution modes is based on the assumption that “the policy is already optimal under that execution mode.” If the policy is still far from optimal, we cannot guarantee that: (1) simply switching to ME will always improve performance; or (2) there is no other execution mode that yields higher return for the **current** policy.
> > > > >
> > > > >     - From our perspective, your idea of “finding more sensible option discovery methods” is very meaningful! However, according to the core viewpoint of our paper, what counts as “reasonable” must be considered separately for three different roles: the target policy, the behavior policy, and the deployment policy:
> > > > >
> > > > >       1. **Behavior policy (exploration and sampling efficiency):**
> > > > >       One can construct subtasks that favor efficient exploration via methods such as bottleneck-state discovery or unsupervised skill discovery, and then invoke them with SME during training, so that all levels receive sufficient data for learning (see Section 4.2).
> > > > >
> > > > >       2. **Target policy (exploitation and learnability):**
> > > > >       On the one hand, one may learn an execution mode that maximizes the return of the **current** policy, e.g., via methods similar to the termination gradient theorem in OC. Our framework instead chooses to directly use ME, which can be viewed as directly using the execution mode that maximizes the return of the **optimal** policy, rather than learning an execution mode that maximizes the return of the *current* policy. On the other hand, we must also avoid deviating too much from the behavior policy’s execution mode, in order to maintain learning efficiency.
> > > > >
> > > > >       3. **Deployment policy (return maximization):**
> > > > >       At test time, we should use the execution mode that maximizes the return of the **currently learned** policy.
> > > > >
> > > > >     - In existing HRL frameworks, the execution modes of these three policies are usually tied together, so there is essentially only a **single** notion of what is “most sensible.” In our framework, these three execution modes are **decoupled**, leading to three different notions of “sensible” which allows us to optimize options in a more principled way. This is one of the key differences between our work and traditional HRL frameworks, and your suggestion is precisely an important direction for our future work.
> > > > >
> > > > > ---
> > > > >
> > > > > Once again, we thank you for these very insightful questions and comments. They not only helped us better clarify the current work, but also pointed to meaningful directions for future research.

---

### Official Review · Reviewer_ZE4K · 2025-10-30

**Soundness:** 2
**Presentation:** 1
**Contribution:** 2
**Rating:** 2
**Confidence:** 4

**Summary:**

The paper investigates different execution models for hierarchical reinforcement learning. In semi-Markov execution (SME) each subtask continues until termination before selecting a new subtask, while in Markov execution (ME) subtask selection is performed at each time step. The authors demonstrate that the expected return is higher under ME, though subtask policies are harder to train under ME. For this reason the authors propose several new learning frameworks that are then tested in experiments.

**Strengths:**

Determining the best way to train and execute subtasks policies in hierarchical reinforcement learning is an important research question, and the two execution modalities proposed by the authors seem like reasonable choices.

**Weaknesses:**

The problem definitions and learning setup are not clearly explained. The authors do not seem to use a discount factor, in which case the value function is only well-defined if all policies eventually terminate with probability 1. The value functions V^ME and V^SME are never formally defined. The definition of a training phase lacks details of exactly how training is performed.

I understand what the authors mean by execution suboptimality, but without formal definitions of V^ME and V^SME I am not sure what policy suboptimality means. It seems to me that the optimal policy of a task is fully determined by the definition of a task on page 2. Either we learn this optimal policy during training, or we learn a suboptimal policy. My best understanding of policy suboptimality is that during training we have not been able to learn the optimal subtask policy.

A curious choice of the authors is to *not* include key theoretical results in the main text. I believe all theorems have to be included in the main text even if the proofs are deferred to the appendix. Otherwise it is impossible for a casual reader to fully appreciate the theoretical contribution. On page 5 there is a supposed proof sketch but it is not clear which theorem or theoretical result is being proven.

On page 5 the authors state that the value computation under different execution modes cannot be directly described by the Bellman equation. I believe that there is a well-defined Bellman equation at each level of the hierarchy, so I am not sure what the authors mean.

For the training framework ESIF, the authors claim that ME execution can achieve expected return J_B. However, when training under SME it is possible that the agent *never makes subtask choices* in some states. For example, if the subtasks traverse rooms between hallways, the SMDP policy may only have been trained to make subtask choices in hallway states. In this case ME execution will fail in room states since the SMDP policy does not know which subtask to choose. It seems that the same thing can happen in PSIF-2S if some subtask choices are never made during Phase 1. Hence I am not convinced that the agent can achieve the optimal return in these training frameworks.

In the description of the training frameworks on page 7, there is suddenly a discount factor. I do not understand why the update rule returns two values V(s_t,w_t) and V(s_t,w_t,a_t). I believe you have to formally state the Bellman equations for subtasks at different levels of the hierarchy.

**Questions:**

Do you train the policies of all subtasks in parallel? This is known to be a non-stationary problem since the subtask policies determine the dynamics at the SMDP level. How are the local rewards of subtasks used?

What does the update rule mean that returns two values V(s_t,w_t) and V(s_t,w_t,a_t)?

---

> ### Author Response · Authors · 2025-11-22
> **Response to Reviewer ZE4K (1/5)**
>
> We sincerely thank Reviewer ZE4K for the valuable comments.
> Our paper introduces a relatively detailed formalism and a large amount of notation, which indeed makes it hard to read on the first pass. Below, we respond to your concerns **point by point**, and we will clarify how we plan to improve the paper.
>
> ---
>
> ### 1. Problem definition, discount factor, and training setup
>
> > “The problem definitions and learning setup are not clearly explained. The authors do not seem to use a discount factor, in which case the value function is only well-defined if all policies eventually terminate with probability 1. … In the description of the training frameworks on page 7, there is suddenly a discount factor.”
>
> #### Clarifying the discount factor
>
> This is an important point. We **do** use a discount factor in our work, but we express it via a termination function $\beta$, and we extend its role in the hierarchical setting. Using termination functions is a standard and convenient choice in HRL. In fact, a large part of our theoretical work can be viewed as studying equivalent formulations and extensions of discounting/termination in hierarchical structures.
>
> 1. **Equivalence between discounting and termination**
>
> In single-layer RL, the discount factor $\gamma$ can be interpreted as follows: after executing one action, the agent moves to an absorbing terminal state $\bar s$ with probability $1-\gamma$ (where $V(\bar s)=0$), and otherwise transitions according to the environment dynamics $T$ with probability $\gamma$. That is,
>
> $V(s)=\mathbb E_{a \sim \pi}[R(s,a)]+\gamma \mathbb E_{a \sim \pi,s' \sim T}[V(s')]$
>
> $= \mathbb E_{a \sim \pi}[R(s,a)]+(1-\gamma) \cdot 0+\gamma \mathbb E_{a \sim \pi,s' \sim T}[V(s')]$
>
> $= \mathbb E_{a \sim \pi}[R(s,a)]+(1-\gamma) \cdot V(\bar s)+\gamma \mathbb E_{a \sim \pi,s' \sim T}[V(s')]$ (1)
>
> If we set $\beta = 1-\gamma$, then “terminating with probability $\beta$” is numerically equivalent to discounting with $\gamma$. For infinite-horizon RL, the usual assumption $\gamma \le 1-\epsilon$ ($\epsilon>0$) becomes $\beta \ge \epsilon>0$. This termination-based view is widely used in options-based HRL, and is the starting point of our formalism.
>
> 2. **Extending discounting in the hierarchical setting**
>
> In single-layer RL, when the task terminates, the episode ends: there are only two cases, “task terminated” vs “task not terminated,” corresponding to the two bootstrap terms in (1). In HRL, by contrast, when a **subtask** terminates, the overall episode typically does *not* end; instead, the parent task continues and may call other subtasks. Thus, a single notion of “termination” is insufficient to capture all inter-level transitions.
>
> In our framework, discounting/termination is naturally extended in two directions, leading to two different roles:
>
> 2.1 **Task termination $\beta_\tau$ (task tree)**
>    This controls, in value computation, what happens *after* a task $\tau$ terminates:
>    - it may return to a global absorbing state with value $0$, or
>    - it may return to its parent task, which continues and whose value we bootstrap. (in OC)
>    This directly determines the optimization objective of each task and which value appears in its Bellman equation.
>
> 2.2 **Decision interruption $\hat\beta_\tau$ (execution tree)**
>    This controls, during execution, **which level makes decisions at each time step**. Even if a task has not terminated in the task-tree sense, the execution tree may return control to its parent with some probability and re-select subtasks. Different choices of $\hat\beta_\tau$ correspond to different execution modes (ME, SME, or more general hybrids).
>
> For example, in two-level Option-Critic, let $\beta_0=1-\gamma$ be the top-level termination and $\beta_1$ be the option termination. After one time step:
>
> - with probability $\beta_0$: the whole task terminates and we jump to the absorbing state;
> - with probability $(1-\beta_0)\beta_1$: the option terminates and the high level chooses a new option;
> - with probability $(1-\beta_0)(1-\beta_1)$: the option continues.
>
> Many existing HRL works implicitly assume $\hat\beta = \beta$, i.e., **target policy, behavior policy, and deployment policy** all share the same SME execution mode. Our paper analyzes the theoretical suboptimality of this “unified SME” choice, and argues that execution modes should be chosen differently depending on the role: in our framework, for instance, we use ME/SME/ME for (target / behavior / deployment) policies. Appendix C provides the formal details.
>
> ---

---

> > ### Author Response · Authors · 2025-11-22
> > **Response to Reviewer ZE4K (2/5)**
> >
> > ### 2. Formal definitions of $V ^{ME}$ and $V ^{SME}$
> >
> > > “The value functions $V ^{ME}$ and $V ^{SME}$ are never formally defined.”
> >
> > In the current version, we do not explicitly write down $V ^{ME}$ and $V ^{SME}$ in the main text; instead, we focus on the verbal description of ME and SME. The unified formal definitions are given in Appendix C (Definition C.12), which unfortunately makes it hard to see what these value functions are.
> >
> > At an intuitive level, one can simply define
> >
> > $V ^{ME}(s)
> > = \mathbb{E}\Big[\sum _{t=0} ^{\infty} \gamma ^t r _t \,\Big|\, \text{starting from } s \text{ under ME execution}\Big],$
> >
> > $V ^{SME}(s)
> > = \mathbb{E}\Big[\sum _{t=0} ^{\infty} \gamma ^t r _t \,\Big|\, \text{starting from } s \text{ under SME execution}\Big].$
> >
> > Formally, for **any** execution mode (including ME and SME), the value function is defined in Definition C.12, and the corresponding Bellman-style recursion in Definition C.13. We place the unified definition there because, no matter which execution mode we use, value can always be written as a sum of rewards under that mode; this cumulative form alone does not make the differences between execution modes explicit. The generalized hierarchical Bellman equation (GHBE) in Definition C.13 is what exposes how the execution tree affects value computation.
> >
> > In Appendix C.3.1, we specify the execution trees for ME and SME. Plugging them into Definition C.13 yields recursive forms of $V ^{ME}$ and $V ^{SME}$ in a multi-level hierarchy. In the revision, we will add a simple two-level OC example to the main text, so that $V ^{ME}$ and $V ^{SME}$ have concrete symbolic definitions already in the main body.
> >
> > Concretely, for two-level OC, the high-level value under SME is
> > $V ^{SME}(s,\omega,a)=R(s,a)+\beta _0 \cdot 0
> > +(1-\beta _0)\beta _1 V ^{SME}(s')
> > +(1-\beta _0)(1-\beta _1)V ^{SME}(s',\omega),$
> >
> > while under ME we have
> > $V ^{ME}(s,\omega,a)=R(s,a)+\beta _0 \cdot 0
> > +(1-\beta _0)\cdot 1 \cdot V ^{ME}(s')
> > +(1-\beta _0)\cdot 0\cdot V ^{ME}(s',\omega).$
> >
> > Here:
> >
> > - $V(s)$ is the value before making a high-level decision at state $s$;
> > - $V(s,\omega)$ is the value after choosing option $\omega$ at state $s$;
> > - $V(s,\omega,a)$ is the value after choosing both option $\omega$ and action $a$ at state $s$.
> >
> > They satisfy
> > $V(s) = \mathbb E _{\omega \sim \pi _{hi}}[V(s,\omega)]
> >      =\mathbb E _{\omega \sim \pi _{hi}}\big[\mathbb E _{a \sim \pi _{lo}}[V(s,\omega,a)]\big].$
> >
> > We will add such concrete definitions (and pointers to Definition C.12/C.13) in the revised main text.
> >
> > ---
> >
> > ### 3. What “policy suboptimality” means
> >
> > > “I understand what the authors mean by execution suboptimality, but … I am not sure what policy suboptimality means.”
> >
> > We appreciate that the reviewer already understands **execution suboptimality**: even when given the high- and low-level policies (e.g., the two-level toy example), switching from SME to ME can change the induced trajectories and returns.
> >
> > We now clarify **policy suboptimality** more explicitly.
> >
> > Intuitively, suppose we fix a rich policy class (containing all relevant hierarchical policies). Then:
> >
> > - under SME execution, there is a set of SME-optimal policies;
> > - under ME execution, there is a set of ME-optimal policies.
> >
> > If we *train* under SME and obtain a policy that is optimal **for SME**, but *deploy* under ME, then even if we fully exploit execution suboptimality (by switching from SME to ME at deployment), our policy may still not be optimal for ME. The gap to the truly ME-optimal policy is what we call **policy suboptimality**.
> >
> > Consider a simple example. The policy class contains only two policies:
> >
> > - Policy 1 yields returns (SME, ME) = (1, 4);
> > - Policy 2 yields returns (SME, ME) = (2, 3).
> >
> > Under SME, policy 2 is better (2 > 1). Under ME, policy 1 is better (4 > 3).
> >
> > If we train under SME and end up with policy 2, and then deploy it under ME, the return improves from 2 to 3. This removes **execution suboptimality** (switching execution mode from SME to ME improves performance). However, policy 1 would have achieved 4 under ME, so the gap $4 - 3$ is **policy suboptimality**.
> >
> > Formally, in the paper we introduce the following decomposition. Let
> >
> > - $\pi^-$ denote the policy that is optimal **for SME** (i.e., trained and evaluated under SME);
> > - $\pi^*$ denote the policy that is optimal **for ME**.
> >
> > Then:
> > $\Delta ^{execution}=V ^{ME} _{\pi ^-}-V ^{SME} _{\pi ^-},$
> >
> > $\Delta ^{policy}=V ^{ME} _{\pi ^*}-V ^{ME} _{\pi ^-},$
> >
> > $\Delta ^{SMDP}=V ^{ME} _{\pi ^\ast}-V ^{SME} _{\pi ^-}=(V ^{ME} _{\pi ^-}-V ^{SME} _{\pi ^-})+(V ^{ME} _{\pi ^\ast}-V ^{ME} _{\pi ^-})=\Delta ^{execution}+\Delta ^{policy}.$
> >
> > One of the main theoretical messages of our paper is that, under our formalism (task trees, execution trees, GHBE), this decomposition and the associated partial order are not special cases but hold in general.
> >
> > ---

---

> > > ### Author Response · Authors · 2025-11-22
> > > **Response to Reviewer ZE4K (3/5)**
> > >
> > > ### 4. Why key theoretical results are not in the main text and what the proof sketch is proving
> > >
> > > > “A curious choice of the authors is to not include key theoretical results in the main text… On page 5 there is a supposed proof sketch but it is not clear which theorem or theoretical result is being proven.”
> > >
> > > We thank the reviewer for carefully reading Appendix C and noticing that, in the course of proving Theorem 4.1, we actually derived several more general theoretical results: a generalized hierarchical Bellman equation (GHBE), hierarchical policy improvement results, and a partial order among execution modes.
> > >
> > > In the current version, the main text focuses on **explaining the SMDP suboptimality phenomenon itself** and presenting Theorem 4.1 as the core conclusion. The more general formalism (task trees, execution trees, GHBE, improvement theorems) is placed in Appendix C. This choice indeed makes it hard for readers to see the full scope of the theoretical contribution from the main body alone.
> > >
> > > In the revision, we will:
> > >
> > > - promote some key theorems from the appendix into Sections 4.1;
> > > - add clearer cross-references between the main text and Appendix C, so that readers can see that the appendix is not only a proof appendix but also a reusable formal toolkit for HRL.
> > >
> > > ---
> > >
> > > ### 5. On Bellman equations at different levels and our GHBE
> > >
> > > > “I believe that there is a well-defined Bellman equation at each level of the hierarchy, so I am not sure what the authors mean.”
> > >
> > > We apologize for the confusing wording in the current version. In Section 2, when introducing SMDPs, we wrote the value recursion
> > > $V_{\pi}(s_t|\tau)=\mathbb E_{\tau',n,s_{t+n}}[R_\tau(s_t,\tau')+\gamma^n V_{\pi}(s_{t+n}|\tau)]$ (2)
> > >
> > > where $s'$ is the state reached after executing subtask $\tau'$ for $n$ steps. This is indeed a valid per-level SMDP Bellman equation.
> > >
> > > However, from the viewpoint of **local task value** and **cross-mode comparison** that we focus on, Equation (2) has two serious limitations:
> > >
> > > 1. If we focus on the local cumulative reward of task $\tau$, Equation (2) can only describe the value of task $\tau$ when it chooses a single subtask $\tau'$. For a 3-level example, suppose at time step $t=5$, task $\tau$ (level 0) selects a subtask $\tau' _5$ (level 1) according to $\hat\pi _\tau$, and then $\tau' _5$ further selects its own subtask $\tau'' _{5}$ (level 2). If at time $t=6$ neither $\tau$ nor $\tau' _5$ has terminated, then at time $t=6$ the next decision might
> > >    - be made at the level of $\tau' _5$ (first selecting a new $\tau'' _{6}$ and then selecting action $a$), or
> > >
> > >    - be made directly at the level of $\tau'' _{5}$ (using $\tau'' _{5}$ to select action $a$).
> > >
> > >    Under the current Bellman equation and value definitions $V _{\pi}(s _{6} \mid \tau)$ and $V _{\pi}(s _{6}, \tau' \mid \tau)$, we cannot distinguish or correctly compute the subsequent value (in terms of the local cumulative reward of $\tau$) in these two different cases.
> > >
> > > 2. More importantly, Equation (2) only relates adjacent levels and hides the effect of the **execution mode** (ME/SME/others) in the random duration $n$. Under SME, the bootstrap happens at a random time $t+n$; under ME, the option duration is always 1 and the bootstrap is at time $t+1$. Because the bootstrap time index is not unified, it is extremely hard to compare values under different execution modes within a single operator framework.
> > >
> > > The GHBE we propose in Definition C.13 addresses this by:
> > >
> > > - unifying all bootstraps at the **next** time step;
> > > - augmenting the state to explicitly record which task currently has control, e.g., $(s_{t+1},\tau)$, $(s_{t+1},\tau,\tau')$, $(s_{t+1},\tau,\tau',\tau'')$, etc.
> > >
> > > Intuitively, if we cannot control “how many steps later” the value is bootstrapped when we stick to a fixed level, we instead fix the time (always at $t+1$) and record **which level** we return to. This one-step GHBE is crucial for our theoretical analysis of execution modes and for deriving the partial order among the four returns in Section 4.
> > >
> > > Note that $V_{\pi}(s_{t+1},\tau,\tau',\tau'' \mid \tau)$ **cannot** be written as $V_{\pi}(s_{t+1},\tau,\tau'' \mid \tau)$. When $\tau''$ is selected from two different parent subtasks $\tau'$ and $\tilde{\tau}'$, the policies are indeed identical **during** the execution of $\tau''$, but they differ **after** $\tau''$ terminates, which leads to different values. This is exactly why our unified construction of value functions and Bellman equations is necessary.
> > >
> > >
> > > ---

---

> > > > ### Author Response · Authors · 2025-11-22
> > > > **Response to Reviewer ZE4K (4/5)**
> > > >
> > > > ### 6. On SME training and states where no subtask choice is made
> > > >
> > > > > “… when training under SME it is possible that the agent never makes subtask choices in some states…”
> > > >
> > > > This is a very sharp observation. The issue can be described more precisely as follows: in a two-level HRL setup, if we train with SME interaction (e.g., the high level acts only every $n$ steps), then during training the high level only makes decisions at a subset of time steps. As a result, there may be states at which the agent has **never** made any high-level decision, and thus has never learned how to act at that level.
> > > >
> > > > We have analyzed this issue in some depth; if space allows, we will add a dedicated discussion in the appendix. The issue has two aspects:
> > > >
> > > > 1. **Sampling coverage.**
> > > >    In the worst case, some states may be unreachable under strict SME training. That is, not only has the high level never acted there, the state is simply never visited (e.g., entering a room in the middle of a hallway may require interrupting a “go-to-the-end” option in the middle).
> > > >
> > > > 2. **Learning coverage.**
> > > >    Even if a state is visited, the high level might never be updated there. Let $d ^{SME}(s)$ be the state distribution during deployment, and $d ^{SME} _{hi}(s)$ be the distribution of states where the high level actually makes decisions during SME training. They can differ significantly; for some states with $d ^{SME}(s) > 0$, we may have $d ^{SME} _{hi}(s) = 0$, so learning on $d ^{SME} _{hi}$ alone requires strong out-of-distribution generalization.
> > > >
> > > > For the **sampling coverage** issue: in both our theoretical analysis (Lemma C.1) and our OC-based experiments (original OC, PSIF-2S, PSIF-1S), the SME behavior policy is not a strict, fixed-interval SME. Instead, in each time step, there is a non-zero probability $\hat\beta$ of interrupting the current option and returning control to the high level. This means that every step has a small probability of “breaking” the current subtask (e.g., in the middle of a hallway) and re-choosing a new one that can enter a room. This “$\epsilon$-SME” behavior allows the agent to, in principle, explore the same state space as ME, just as $\epsilon$-greedy can explore the entire state space while greedy cannot.
> > > >
> > > > For the **learning coverage** issue: even if we only collect SME interaction data, we do **not** restrict value updates to the high-level decision points. While Equation (2) suggests updating only at those points, our GHBE (Definition C.13) reveals that high-level values can be updated at any state along the trajectory. In particular:
> > > >
> > > > - For OC-type algorithms (or Goal-based methods with state-dependent termination), if an option $\omega_{t_0}$ is chosen at time $t_0$, then for any later time $t>t_0$ while $\omega_{t_0}$ is active, “continuing $\omega_{t_0}$” is equivalent to “re-selecting $\omega_{t_0}$” at $s_t$. Thus, we can treat these states as if the high level chose $\omega_{t_0}$ at time $t$ and update the corresponding value. This is in fact what the original OC algorithm does.
> > > >
> > > > - For Goal-based algorithms with fixed duration $n$, we can extend the goal space to (goal, remaining time). If goal $g_{t_0}$ is chosen at time $t_0$ with duration $n$, then at time $t_0+n > t \geq t_0$ we can still learn values for “goal $g_{t_0}$ with remaining duration $n-(t-t_0)$,” and later use these to compare goals with different remaining durations in ME execution.
> > > >
> > > > Overall, these two aspects are **algorithm design** issues about sampling and value updates, and are orthogonal to the conceptual notion of SMDP suboptimality (which concerns the execution mode of the behavior policy, the Bellman equation for the target policy, and the deployment execution mode). For our OC-based experiments, the above design choices make these coverage issues much less problematic, so OC is well suited to validating our theoretical claims about $J_\mathcal{B}$ and $J_\mathcal{A}$.
> > > >
> > > > For the original HAC algorithm used in our experiments, our goal is to test the **robustness** of our conclusions when the algorithm does not fully satisfy the theoretical assumptions. In this setting, **both issues indeed occur** and affect its final performance.Theoretically, decisions at intermediate states rely on **out-of-distribution generalization**: the optimal policy must have a certain degree of smoothness to maintain good performance. In the environments we construct in our paper—where random jumps are added and the agent is required to flexibly “change plans” on the fly—although these two factors still have an impact, the effect of **execution suboptimality** remains dominant: ME get higher return than SME   .
> > > >
> > > >
> > > > ---

---

> > > > > ### Author Response · Authors · 2025-11-22
> > > > > **Response to Reviewer ZE4K (5/5)**
> > > > >
> > > > > ### 7. Why the update rule returns both $V(s_t,w_t)$ and $V(s_t,w_t,a_t)$
> > > > >
> > > > > > “What does the update rule mean that returns two values $V(s_t,w_t)$ and $V(s_t,w_t,a_t)$? I believe you have to formally state the Bellman equations for subtasks at different levels of the hierarchy. The definition of a training phase lacks details of exactly how training is performed.”
> > > > >
> > > > > #### Meaning of the values and training-time execution details
> > > > >
> > > > > In single-level RL, the quantity $R + V(s')$ can be used to update both $V(s)$ and $V(s,a)$. The update distributions differ:
> > > > >
> > > > > - $V(s,a)$ is only updated when the specific pair $(s,a)$ is sampled;
> > > > > - $V(s)$ can be updated whenever the state $s$ is visited (regardless of which action was chosen).
> > > > >
> > > > > Here, $V(s)$ denotes the value “starting from state $s$,” while $V(s,a)$ denotes the value “starting from $s$ and taking action $a$.”
> > > > >
> > > > > In our hierarchical setting, we may analogously maintain three kinds of values: $V(s)$, $V(s,\omega)$, and $V(s,\omega,a)$:
> > > > >
> > > > > - $V(s,\omega)$ is updated when option $\omega$ is chosen at $s$;
> > > > > - $V(s,\omega,a)$ is updated when both $\omega$ and $a$ are chosen at $s$;
> > > > > - $V(s)$ is not needed for some lower-level updates, so in practice we may not explicitly maintain it.
> > > > >
> > > > > Here, $V(s,\omega)$ denotes the value “starting from state $s$ and choosing option $\omega$,” and $V(s,\omega,a)$ denotes the value “starting from $s$, choosing option $\omega$, and then choosing action $a$.”
> > > > >
> > > > > In PSIF-1S for OC, although both high-level and low-level receive environment rewards, their **target execution modes differ**, so their value functions and Bellman equations are not the same:
> > > > >
> > > > > - To optimize the high-level policy, we use environment rewards and the high-level target execution mode (ME) to compute $V_{hi}(s)$, $V_{hi}(s,\omega)$, and $V_{hi}(s,\omega,a)$; in practice, only $V_{hi}(s,\omega)$ is needed.
> > > > > - To optimize the low-level option policy, we use environment rewards and the low-level target execution mode (ME inside options, SME outside) to compute $V_{lo}(s)$, $V_{lo}(s,\omega)$, and $V_{lo}(s,\omega,a)$; in practice, we only use $V_{lo}(s,\omega,a)$ for learning the option policy, while $V_{lo}(s,\omega)$ is used to learn the option termination.
> > > > >
> > > > > In our paper:
> > > > >
> > > > > - Definition C.13 gives the unified hierarchical Bellman equation;
> > > > > - Appendix D.2 derives the two-level OC and two-level Goal-based Bellman equations (under SME) as special cases;
> > > > > - Appendix D.1 provides pseudocode for PSIF-2S and PSIF-1S, and Table 6 summarizes the behavior/target execution modes during training.
> > > > >
> > > > > We will add pointers from the main text to these appendix sections to make the training-time execution details clearer.
> > > > >
> > > > > ---
> > > > >
> > > > > ### 8. Parallel training, non-stationarity, and local rewards
> > > > >
> > > > > > “Do you train the policies of all subtasks in parallel? This is known to be a non-stationary problem since the subtask policies determine the dynamics at the SMDP level. How are the local rewards of subtasks used?”
> > > > >
> > > > > #### Parallel training and non-stationarity
> > > > >
> > > > > Yes, we train all levels in parallel. The non-stationarity arises because the transitions and rewards induced by subtasks change during training as the lower-level policies are updated.
> > > > >
> > > > > For OC and its variants, the value bootstraps to the **next** state. The environment itself is stationary, so this is similar in spirit to off-policy Q-learning with a changing behavior policy. Our theoretical analysis explicitly compares values under **fixed** policies, not the transient training process, so it does not require strict stationarity at every intermediate step.
> > > > >
> > > > > For HAC and HAC-ESIF, the high-level backup is over a variable number of time steps; the per-step subgoal reward is always $-1$ (stationary), but the distribution of the resulting transition states changes as the low-level policy changes. This is a known challenge in HAC; we follow the original HAC paper and use hindsight techniques to mitigate it.
> > > > >
> > > > > #### Local rewards for subtasks
> > > > >
> > > > > In OC and our ESIF/PSIF variants, we do **not** design separate local rewards: both high-level and low-level directly use the environment reward. The difference between levels comes from their respective **target execution modes**.
> > > > >
> > > > > In contrast, HAC is feudal: lower-level subgoals receive a local reward of $-1$ per step until the subgoal is reached or a fixed horizon is exceeded, and do not directly optimize the global task reward. The high level, instead, is responsible for optimizing the final task return. This can be naturally expressed in our task-tree view as a feudal structure.
> > > > >
> > > > > ---
> > > > >
> > > > > Some of our concepts, notation, and theorems are admittedly complex on first sight, but we believe that the underlying structure is quite unified and useful. We greatly appreciate your detailed questions and suggestions; they have helped us improve the clarity of our presentation and make our assumptions and goals more transparent. We would be very happy to discuss any further questions you may have.

---

### Author Response · Authors · 2025-11-22
**Summary of Key Revisions to the Manuscript (Nov 22)**

We have revised the manuscript in response to the reviewers’ comments. The main changes are as follows:

1. Added formal definitions of $V^{ME}$ and $V^{SME}$ (page 3); moved the explanation of the equivalence between the discount factor and termination from a footnote into the main text (page 2); and rewrote the updates for $V(s,a)$ and $V(s,\omega,a)$ (page 8), splitting them into two clearer update equations.

2. For the abbreviated theorem names used in the proof sketch in Section 4, we now include their full statements. The key theorems — the **Hierarchical Policy Improvement Theorem**, **Hierarchical Execution Improvement Theorem**, and **Optimal Execution Theorem** — have been moved from the appendix into Section 4.1 of the main text (page 6). We also added explicit references to the appendix for the main formal contributions (page 5).

3. In the caption of Figure 3, we now provide a more detailed explanation of the figure contents, so that trajectories, colors, policies, and execution modes correspond clearly (page 4). The detailed illustrative subfigures obtained by decomposing Figure 3 have been moved to the appendix (page 14–16). We also improved the legends of Figures 2 and 4 (page 4, 7) and clarified the cropping description for Figure 5 (page 9).

4. In Sections 4.1, 4.2, and 4.3, we added more detailed analysis and logical organization of the target policy, behavior policy, and deployment policy, making explicit the logical chain between the theoretical results in Section 4 and the resulting choices of execution modes (page 6–7). We also explicitly state that the proof sketch is for Theorem 4.1 (page 5).

5. We boldfaced key concepts such as task (page 2), and added a concise summary of the differences between MDPs and SMDPs (page 3).

We again thank you for your careful review of our theoretical work.

---

### Author Response · Authors · 2025-12-02
**Rebuttal Summary and Appreciation**

# Rebuttal Summary and Appreciation

Dear Reviewers, ACs, SACs, and PCs,

We sincerely thank you for your time, expertise, and thoughtful engagement throughout the review and rebuttal process. Your comments have significantly improved both the clarity and the scope of our work on suboptimality in SMDP-based HRL. We have greatly benefited from the detailed discussions with all four reviewers (**ZE4K**, **YuW2**, **q9eR**, **vbRc**).

---

Below we briefly summarize how the rebuttal process refined the paper and our presentation:

* **Clarifying the problem setting and formalism (mainly to ZE4K and vbRc)**

  * We clarified that our focus is: *given an existing hierarchical structure*, how different execution modes (SME/ME) for **target, behavior, and deployment** policies induce two kinds of suboptimality, rather than proposing a new “hierarchical method”.
  * We added **formal definitions** of tasks, subtasks, and HRL/SMDP, and provided a clearer, concise **comparison between MDPs and SMDPs**, as requested by **vbRc**.
  * We made the notion of “well-trained” explicit (“each node’s policy is optimal under a given execution mode”) and gave concrete symbolic definitions of $V^{\text{ME}}$ and $V^{\text{SME}}$, along with their Bellman-style recursions, addressing **ZE4K**’s concerns on value definitions and training setup.

* **Task tree, execution tree, and GHBE: making the theory more visible**

  * Several key theoretical tools originally buried in Appendix C—**Task Trees**, **Execution Trees**, **Unified Value Function (UVFH)**, and the **Generalized Hierarchical Bellman Equation (GHBE)**—are now more clearly linked to the main text.
  * We promoted and highlighted core results such as the **Hierarchical Policy Improvement Theorem**, **Hierarchical Execution Improvement Theorem**, and **Optimal Execution Theorem**, and clarified how **Theorem 4.1** formally underpins the “fundamental design flaw” (“SMDP/SME simultaneously in target and behavior policies”) emphasized in the abstract.

* **Execution modes and ME vs SME (mainly to q9eR and YuW2)**

  * In response to **q9eR**’s central concern, we clarified that:

    * ME is **not** proposed as a new hierarchical method, but as a *replaceable execution component* within an existing hierarchical architecture.
    * Our analysis advocates **ME for target & deployment** and **SME for behavior**, and stresses that these three roles must be designed **separately**, not tied to a single SME execution mode.
  * We distinguished two notions of “hierarchy”: structural (multi-level mapping) vs temporal (different decision frequencies), and explained why ME can be temporally flat yet still structurally hierarchical.
  * We used OC and its variants to explain why optimizing termination purely for upper-level return tends to shorten options toward ME, while many follow-up OC works explicitly discourage single-step termination—this supports our claim that “ME is return-optimal, but SME is exploration-friendly”.

* **Training validity, off-policy issues (mainly to ZE4K, YuW2 and q9eR)**

  * To **ZE4K** and **YuW2**, we explained how UVFH+GHBE rewrite multi-level values so that **frame-wise off-policy updates across execution modes** are well-defined, and how $\epsilon$-SME behavior guarantees sufficient coverage for ME-style targets.
  * To **q9eR**, we clarified why hierarchical ME at deployment can significantly outperform flat RL:

    * Training uses **SME behavior** to reach bottleneck states via temporally extended subtasks.
    * The **target policy** is optimized under ME to remove SMDP-induced suboptimality.
    * Flat RL baselines in our sparse-reward environments often fail to even discover successful trajectories within the fixed budget, while HRL (and our frameworks) do.

* **Readability, figures, and notation burden (mainly to YuW2 and vbRc)**

  * We substantially improved **Figure 3** and other plots (legends, captions, and subfigures), aligning colors/paths with symbols such as $\pi^{-}$, $\pi^*$, and clearly marking SME/ME trajectories.
  * We clarified the proof sketch of Theorem 4.1, reduced unnecessary acronyms in the main text, and strengthened cross-references between the main body and the appendices.
  * We boldfaced key concepts (e.g., **task**, **subtask**, **target policy**, **behavior policy**, **deployment policy**) and added a short high-level summary of our theoretical pipeline to help readers navigate the numerous definitions.

---

We are deeply grateful to **ZE4K**, **YuW2**, **q9eR**, and **vbRc** for their careful reading, critical questions, and constructive suggestions. Regardless of the final decision, this has been an immensely valuable experience for us, and we will carry these insights into future work.

Best regards,
**All authors**

---

### Meta-Review · Area_Chair_7ciX · 2026-01-09

**Summary:**

The authors identify a structural limitation in Hierarchical Reinforcement Learning (HRL) wherein Semi-Markov Decision Processes (SMDP) lock agents into fixed subtasks, limiting adaptability. To address this, they propose a framework decoupling execution modes: using Semi-Markov Execution (SME) for behavior policies (to aid exploration) and Markov Execution (ME) for target/deployment policies (to maximize returns). They provide theoretical analysis and experimental verifications.

The reviews were mixed (Scores: 2, 2, 6, 6). However, there was a consensus across all reviewers—including those leaning positive—that the paper suffers from significant readability issues. Critics cited undefined concepts in the main text, key theorems buried in the appendix, and confusing figures (particularly Figure 3). Additionally, there were concerns regarding the conceptual motivation of the proposed execution modes and theoretical clarity.

Post-rebuttal interactions were limited. Only one of the reviewers engaged with the authors' response, and while they acknowledged the validity of the results, they remained unconvinced regarding the motivation of the method and the "counterintuitive ramifications" of the execution modes. Given the outstanding conceptual concerns and the need for major structural revisions to improve clarity (e.g., moving theorems, formalizing definitions), the paper is not ready for publication in its current form. Accordingly, the recommendation is to Reject.

**Reviewer Concerns:**

- The authors clarified the relationship between termination functions and discount factors (responding to ZE4K), and conceptually distinguished their approach from standard Option Discovery/termination optimization (responding to YuW2).
- The decomposition of suboptimality into "execution" and "policy" components was clarified, helping reviewers (like YuW2 and vbRc) better understand the theoretical contribution regarding the structural limitations of SMDPs.

Outstanding Concerns:
- Reviewer q9eR remained concerned that using ME at deployment effectively reduces the hierarchy to a flat policy acting as an "action filter." The reviewer found the ramifications of this "counterintuitive" and was not fully convinced by the distinction between using ME for targets/deployment versus SME for behavior/exploration.
- A consensus concern across all reviewers was that the paper is difficult to follow.
- Reviewer ZE4K questioned the sampling coverage for states that might not be visited under SME behavior, and the off-policy stability issues raised by YuW2 were addressed theoretically but lacked sufficient empirical reassurance for the reviewers.

**Reviewer Scores:**

- Reviewer q9eR (Score: 2): Was the only reviewer to respond to the rebuttal. They maintained that the results imply "counterintuitive ramifications that need to be addressed in the paper prior to publication." They did not offer to raise their score.

- Reviewer ZE4K (Score: 2): Did not respond to the rebuttal.
- Reviewer YuW2 (Score: 6): Did not respond to the rebuttal.
- Reviewer vbRc (Score: 6): Did not respond to the rebuttal.

---

### Decision · Program_Chairs · 2026-01-26

Reject